# DISTRIBUTIONAL ASSOCIATIONS VS IN-CONTEXT REASONING: A STUDY OF FEED-FORWARD AND ATTENTION LAYERS

**Lei Chen**[*] **& Joan Bruna**
Courant Institute of Mathematical Sciences
Center for Data Science
New York University

**Alberto Bietti**
Flatiron Institute

## ABSTRACT

Large language models have been successful at tasks involving basic forms of in-context reasoning, such as generating coherent language, as well as storing vast amounts of knowledge. At the core of the Transformer architecture behind such models are feed-forward and attention layers, which are often associated to knowledge and reasoning, respectively. In this paper, we study this distinction empirically and theoretically in a controlled synthetic setting where certain next-token predictions involve both distributional and in-context information. We find that feed-forward layers tend to learn simple distributional associations such as bigrams, while attention layers focus on in-context reasoning. Our theoretical analysis identifies the noise in the gradients as a key factor behind this discrepancy. Finally, we illustrate how similar disparities emerge in pre-trained models through ablations on the Pythia model family on simple reasoning tasks.

## 1 INTRODUCTION

Large language models (LLMs) have shown impressive capabilities on a variety of tasks, from generating coherent and grammatically correct text, to language understanding and basic mathematical reasoning (Brown et al., 2020; Touvron et al., 2023). At the heart of this success is the Transformer architecture (Vaswani et al., 2017), which relies on a sequence of self-attention and feed-forward layers to efficiently combine information from the input context and patterns learned from training data. Despite recent progress on interpreting the mechanisms learned by different layers (Meng et al., 2022; Wang et al., 2022), these models remain largely black boxes. A better understanding of the role of Transformer layers and how they are affected by the training process could enable new monitoring and editing techniques, better training data, and ultimately more reliable LLMs.

The task of next-token prediction in language modeling inherently involves different subtasks that may be at odds with each other, as shown in Figure 1. For instance, given the context "John gave a book to", the word "the" is a natural and grammatically correct next word to predict, and relying on global bigram statistics might be enough to predict it given the last word "to". Nonetheless, if another character is present in the context, say Mary, then the name "Mary" may be a better prediction, and this would require a more involved form of reasoning over the context to retrieve this name. In the context of Transformer language models, previous work on interpretability has found that circuits of attention heads seem responsible for such in-context predictions (Wang et al., 2022), while feed-forward layers may be storing more general statistics such as the bigram "to the" or factual knowledge (Geva et al., 2021; Meng et al., 2022; Bietti et al., 2023). To further strengthen this observation, the recent work (Sharma et al., 2023) found that selectively replacing certain layer weights to their low-rank approximation, particularly late feed-forward layers, may improve performance on various reasoning benchmarks, and observed that the truncated components were often responsible for predicting "generic" tokens such as the word "the".

In this paper, we provide a finer understanding of these phenomena by studying how such mechanisms arise during training, in particular how simple *distributional associations*, such as the bigram "to the",

---

[*]The implementation is at `https://github.com/leichen2018/ATTN_vs_MLP`

Figure 1: **Distributional association *v.s.* in-context reasoning.** In this work, we decompose tasks of next-token prediction into the distributional and the in-context ones, finding that MLPs learn distributional associations before attention develops in-context reasoning capabilities. Furthermore, truncating MLPs promotes in-context reasoning by weakening distributional associations. See Figure 5 for an example of this on the Pythia model (Biderman et al., 2023).

tend to be localized in feed-forward layers, while attention focuses on in-context reasoning. We first provide a fine-grained study of training dynamics on a synthetic task with two-layer transformers exhibiting similar properties, where the task is in-context recall (Bietti et al., 2023) with additional noise on in-context tokens consisting of a fixed "generic" token:

- In a two-layer model with feed-forward layers (FF), we show that the generic noise token is mainly learned in FF and the attention attends towards correct in-context targets. Removing the feed-forward layers then leads to clean in-context predictions. We provide some theoretical justification through early training steps.
- In a model without FF, we show that the generic noise can be identified in a rank-one subspace of the value matrix in attention block. When the noise level is small, low-rank truncation can filter it out and predict clean outputs.

We then investigate such a separation between distributional association and in-context reasoning on pre-trained language models, namely the Pythia family, which has checkpoints available at different training steps (Biderman et al., 2023). Overall, we provide a useful description of how distributional associations and in-context reasoning mechanisms are learned during training, and tend to be disentangled in different parts of the model, such that selectively removing certain components may lead to better predictions in reasoning tasks.

**Related work.** Sharma et al. (2023) recently empirically observed that a low-rank approximation of some weights in some pre-trained LLMs can improve reasoning capabilities. Several interpretability works have looked at the role of attention versus feed-forward layers for different tasks. The prominence of feed-forward/MLP layers for storing "global" or "persistent" associations or facts has been observed in (Sukhbaatar et al., 2019; Geva et al., 2021; Meng et al., 2022; Geva et al., 2023). In contrast, several works have investigated the role of attention heads for "reasoning" or computation over the context, *e.g.*, for simple copying mechanisms with so-called induction heads (Elhage et al., 2021; Olsson et al., 2022; Bietti et al., 2023), or for more complex tasks (Merrill et al., 2022; Wang et al., 2022; Zhang et al., 2022; Liu et al., 2023; Sanford et al., 2024b).

Training dynamics of transformers and attention have been studied in various works (Snell et al., 2021; Jelassi et al., 2022; Li et al., 2023; Oymak et al., 2023; Tian et al., 2023; Bietti et al., 2023; Reddy, 2024; Tian et al., 2024; Zhang et al., 2024; Nichani et al., 2024; Edelman et al., 2024). In particular, the two-layer model and copy task we consider are similar to Bietti et al. (2023), yet their data model does not involve noise on in-context predictions, and they do not study learning of global associations. Chan et al. (2022); Reddy (2024) study in-context vs. in-weights learning empirically, on different tasks than ours. Cabannes et al. (2024) study training dynamics of linear associative memories, but focuses on deterministic data while our setup has generic noise. Training dynamics were also studied empirically for interpretability (Olsson et al., 2022; Nanda et al., 2023; Quirke et al., 2023; Chen et al., 2024). Edelman et al. (2022); Bai et al. (2023); Abernethy et al. (2024) studied sample complexity of self-attention and in-context learning, but did not consider training dynamics.

## 2 PRELIMINARIES

In this section, we provide some background and motivation on reasoning tasks, and describe the weight truncation technique which we use for ablating weights.

### 2.1 REASONING FROM CONTEXT

Recent LLMs have shown promising results in more complex "reasoning" tasks which may involve multiple steps of logical or computational processing from context or prompt (Srivastava et al., 2022; Wei et al., 2022; Bubeck et al., 2023; Dziri et al., 2024), as opposed to simple pattern matching or memorization of training data, for instance using learned n-gram predictions.

While it is difficult to clearly separate reasoning from memorization, in this work we will make the simplifying distinction that **in-context reasoning** involves dependencies between *multiple tokens* potentially far away in the context, while we consider **distributional associations** as simpler predictions that only depend on the *last token*, e.g., through a bigram model. Thus, due to the residual structure of Transformers, reasoning will typically require using attention operations in Transformers over context, while feed-forward layers should suffice for learning distributional associations. We note that our assumption of distributional associations depending only on the last token is mainly for convenience of our analysis, and could be extended to depend on the last token's *residual stream* (Elhage et al., 2021), which may contain additional information from the context. For instance, this could include previous tokens thanks to position-based attention heads (Voita et al., 2019; Elhage et al., 2021; Akyürek et al., 2024), which allows capturing n-grams instead of just bigrams.

Under this definition, we list a few simple examples of reasoning that we will consider below:

- *In-context recall*: when the last token is a, we'd like to copy the token that follows previous occurrences of a in the context. This `[.. a b .. a] → b` pattern typically requires a two-layer *induction head* mechanism (Elhage et al., 2021; Bietti et al., 2023; Sanford et al., 2024a);
- *Indirect object identification (IOI)*: we consider contexts of the form "When Mary and John went to the store, John gave the ice cream to" where the prediction should be "Mary" (IO, the indirect object), instead of "John" (S, the subject). Wang et al. (2022) found a circuit of several attention heads that perform this task by copying the name which only occurs once in the context;
- *Factual recall*: sentences of the form "Paul Citroen is a native speaker of" with target "Dutch" as in (Sharma et al., 2023). While this may be seen as retrieving a distributional association, we will treat it here as reasoning since it involves combining the subject and relation from the context, while a bigram model that only depends on the last token "of" might instead predict the generic word "the".

### 2.2 TRUNCATING WEIGHTS WITH LASER (SHARMA ET AL., 2023)

In order to assess the importance of different weight components for certain predictions, we use the weight truncation technique introduced by Sharma et al. (2023). They observed that reducing the rank of MLP matrices in certain layers of LLMs effectively brings better performance on several reasoning benchmarks. Their proposed method, Layer-Selective Rank Reduction (LASER), replaces any matrix in the full model by its low-rank approximation with fraction $\rho$, *i.e.*, a matrix $\mathbf{W} \in \mathbb{R}^{d_{in}, d_{out}}$ would be replaced by its rank-$\lfloor \rho \cdot \min\{d_{in}, d_{out}\} \rfloor$ approximation via Singular Value Decomposition (SVD). After searching for the best parameters of different models on different datasets, Sharma et al. (2023) found that applying their method to weight matrices of MLPs on relatively deep layers can enhance in-context reasoning performance on various benchmarks, consistent with our findings. The optimal $\rho$ is smaller than $0.2$ for many datasets.

Another observation from Sharma et al. (2023) is that, when LASER improves the model's prediction on some samples, the full model often predicts "generic" words while the improved model is able to predict the ground-truth answer. For instance, given an input "Madrid is located in", the full model predicts "the" while the truncated model predicts the target "Spain" in Table 1. Here, the generic word is consistent with our definition of distributional associations in Section 2.1, as it may naturally follow from a bigram distribution conditioned on "in", while the factual answer is more akin to reasoning from context. Thus, we would like to better understand how such a modification of

feed-forward layers improves the model from predicting generic words to inferring the answer from context, and how such a gap appears during training.

## 3   TWO-LAYER TRANSFORMER ON NOISY IN-CONTEXT RECALL

In this section, we consider simple one- or two-layer transformers on an in-context recall task with added generic token noise, which allows us to study the trade-offs between MLPs and attention layers for storing in-context versus distributional associations, in a controlled setting. We empirically show how transformers solve this task by storing the generic noise token in feed-forward layers, while attention implements the in-context mechanism. We then provide theory showing that feed-forward layers are more likely to store the distributional association (generic token) while attention learns to attend to in-context targets. Finally, we show that when the model has no feed-forward layers, the value matrix in attention stores both in-context and distributional information, in different subspaces.

**Data and task.** The data model we consider is similar to Bietti et al. (2023), with additional noise. Consider a vocabulary $\mathcal{V} = \{1, 2, \ldots, N, N+1\}$. The token $\tau \triangleq N+1$ is the generic noise token. We fix a *trigger* token $q \in [N]$, which governs in-context recall, and a context length $T$. Each sequence of tokens $z_{1:T} = [z_1, z_2, \ldots, z_T]$ is generated as follows:

i. Sample a correct *output* token $\bar{y}$ uniformly in $[N]$.

ii. Sample $z_{1:T-1}$ according to the following Markov process ($\pi_u, \pi_b$ are distributions on $[N]$ defined later): $z_1 \sim \pi_u(\cdot)$, and

$$z_{t+1}|z_t \sim \begin{cases} \pi_b(\cdot|z_t), & \text{if } z_t \neq q, \\ p_{\alpha,\bar{y}}(\cdot), & \text{otherwise,} \end{cases} \qquad p_{\alpha,\bar{y}}(x) = \begin{cases} 1-\alpha, & \text{if } x = \bar{y}, \\ \alpha, & \text{if } x = \tau, \\ 0, & \text{otherwise.} \end{cases}$$

iii. Set $z_T = q$, and sample the final output $y = z_{T+1} \sim p_{\alpha,\bar{y}}(\cdot)$.

Note that the true $\bar{y}$ varies across sequences, so that the model needs to infer it from context, e.g., using an induction head as in (Bietti et al., 2023). Predicting $\bar{y}$ may thus be seen as a basic "reasoning" task, yet when training with $\alpha > 0$, the noisy output also requires the model to learn a distributional trigger-noise association, similar to the "of/in the" bigram discussed in Section 2. We also consider using multiple trigger tokens in Appendix B.4 and Figure 9.

**Two-layer transformer.** We consider a simplified two-layer transformer formulated below. The input is a sequence of tokens $z_{1:T} = [z_1, \ldots, z_T] \in [N+1]^T$, and the output is $\xi$. The embedding matrix $\mathbf{W}_E \in \mathbb{R}^{(N+1)\times d}$ and un-embedding matrix $\mathbf{W}_U \in \mathbb{R}^{(N+1)\times d}$ are fixed at random initialization. The two attention layers have learnable weights $\mathbf{W}_{QK}^1, \mathbf{W}_V^1, \mathbf{W}_{QK}^2, \mathbf{W}_V^2 \in \mathbb{R}^{d \times d}$ with $\sigma(\cdot)$ the softmax on a vector. The two feed-forward layers $F_1, F_2$ are also learnable, and typically we set them as two-layer MLPs with ReLU activation. We will discuss different architectural choices of $F_1, F_2$ in Appendix B.5. We use the cross-entropy loss to predict $y = z_{T+1}$ from the logits $\xi_T \in \mathbb{R}^{N+1}$.

$$
\begin{aligned}
x_t &\triangleq \mathbf{W}_E(z_t) + p_t, \\
h_t^1 &\triangleq \sum_{s \leq t} \left[ \sigma(x_t^\top \mathbf{W}_{QK}^1 x_{1:t}) \right]_s \cdot \mathbf{W}_V^1 x_s, \\
x_t^1 &\triangleq x_t + h_t^1 + F_1(x_t + h_t^1), \\
h_t^2 &\triangleq \sum_{s \leq t} \left[ \sigma(x_t^{1\top} \mathbf{W}_{QK}^2 x_{1:t}^1) \right]_s \cdot \mathbf{W}_V^2 x_s^1, \\
x_t^2 &\triangleq x_t^1 + h_t^2 + F_2(x_t^1 + h_t^2), \\
\xi_t &\triangleq \mathbf{W}_U x_t^2.
\end{aligned}
\tag{1}
$$

Table 1: Probabilities of the top-5 next-tokens in Pythia-1B before and after LASER. The input prompt is "Madrid is located in". Probabilities of two generic words, *i.e.*, "the" and "a", drop sharply after LASER, while probabilities of meaningful words increase, especially the target "Spain".

|  | "the" | "Spain" | "a" | "southern" | "northern" |
|---|---|---|---|---|---|
| Full | **0.499** | 0.079 | 0.069 | 0.023 | 0.021 |
| LASER | 0.027 | **0.300** | 0.002 | 0.044 | 0.046 |

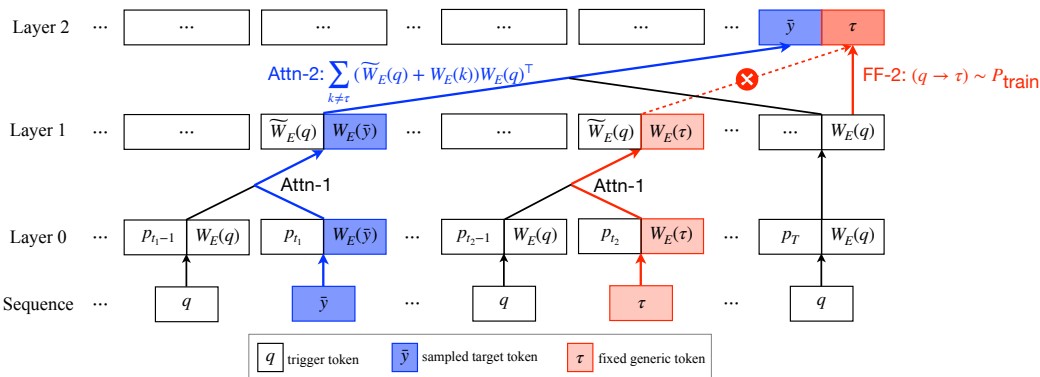

Figure 2: **Noisy in-context recall**. *Purpose of design*: understand mechanisms of attention and feed-forward layers for tasks with **in-context reasoning** (predict $\bar{y}$) and **distributional association** (predict $\tau$). *Task*: predict tokens $\bar{y}$ *v.s.* $\tau$ from a sentence $[\dots, q, \bar{y}, \dots, q, \tau, \dots, q]$ where $q$ is trigger, $\bar{y}$ is sampled target token for a sentence, and $\tau$ is a fixed generic token across sentences. *Our findings*: in a two-layer transformer, the second-layer attention (Attn-2) only attends towards target tuples $[q, \bar{y}]$ while the feed-forward layer (FF-2) learns to predict $\tau$.

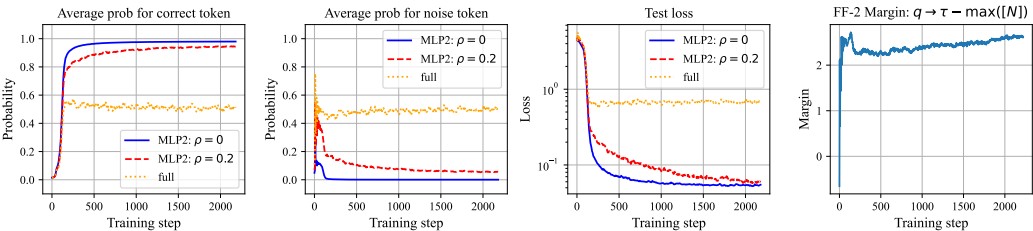

Figure 3: **Left three**: Average probability of predicting correct and noise tokens, and test loss on clean data ($\alpha = 0$), with different fractions $\rho$ of preserved rank in $U_{in}$ of the second-layer MLP $F_2$. The full model learns to predict noise with probability around $\alpha = 0.5$, as expected from training data. When $F_2$ is dropped ($\rho = 0$), the model predicts the correct token $\bar{y}$ with probability $\approx 0.98$. **Rightmost**: the FF-2 margin of $\tau$ *v.s.* all the other tokens with input as $q$, *i.e.*, $[\mathbf{W}_U F_2(\mathbf{W}_E(q))]_\tau - \max_{k \leq N}[\mathbf{W}_U F_2(\mathbf{W}_E(q))]_k$. It reveals that FF-2 learns trigger-noise association in early steps.

**Experimental observations.** Following Bietti et al. (2023), we take $\pi_u$ and $\pi_b$ to be the unigram and brigram character-level distributions estimated from the tiny Shakespeare dataset with $N = 65$. The model setup includes $d = 256$ and two-layer MLPs with ReLU for both $F_1, F_2$. The training setup includes batch size as $512$ and the context length $T = 256$. When evaluating trained models, we consider LASER on the input weight $U_{in}$ of $F_2$. We consider a noise level $\alpha = 0.5$ for training data (though any other constant value would lead to similar observations). During test time, we set $\alpha = 0$ to compute the test loss, aiming to measure how likely the (full or after-truncation) model predicts the ground-truth $\bar{y}$.

Experimental results are reported in Figure 3 and 8. The full model predicts noise with probability close to $\alpha$, which is expected since it is trained to predict the noise token w.p. $\alpha$. However, when dropping the second-layer MLP $F_2$, the truncated model predicts the ground-truth $\bar{y}$ with an almost perfect probability $\approx 0.98$. This suggests that $F_2$ is responsible for storing the distributional association "[trigger] + [noise]". Another observation is that the full model first learns to predict the noise with high probability in very early steps, after which it starts learning to predict the correct $\bar{y}$, which resembles the dynamics observed for learning the "to/in the" bigram in Pythia models in Figure 5. This suggests that learning the (distributional) trigger-noise association is easier than predicting $\bar{y}$, and we will study this theoretically in Section 3.1.

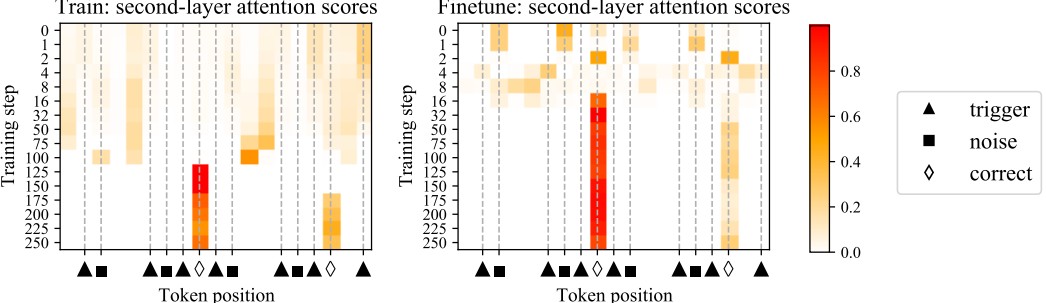

Figure 4: The second-layer attention scores of models trained with noise (left), fine-tuned with noise (right, initialized as a model pre-trained without noise), given the same input. It turns out both models learn to attend to the informative structure "[trigger]+$\bar{y}$" instead of "[trigger]+noise". This implies that the attention in these models is only responsible to predict $\bar{y}$, although the training input and output have noise with probability $\alpha = \Theta(1)$. The fine-tuning setting is in Appendix B.1.

After the distributional noise association is learned, we observe a slower learning of an induction head mechanism, with similar dynamics to Bietti et al. (2023). Compared to Bietti et al. (2023), we notice that the induction head (i.e., the second layer attention head) filters out the noise tokens and only attends to non-noisy output tokens following the trigger, corresponding to the correct $\bar{y}$, as shown in Figure 4. We present theoretical understanding for this mechanism in Section 3.2. Figure 2 and Appendix B.2 summarize the roles of all components of the two-layer transformer in this task.

**Simplified architecture and data for theoretical analysis.** Understanding the full dynamics of the model used in our experiments is out of the scope of the present paper, due to the many moving parts and the complexity of non-linear MLPs. Instead, we focus on a simpler model involving one linear feed-forward layer and one attention layer, and look at the gradient dynamics near initialization. We consider the following simplified 1-layer model. The input $x_t \in \mathbb{R}^d$ at position $t$ is defined as $x_t \triangleq \mathbf{W}_E(z_t) + \widetilde{\mathbf{W}}_E(z_{t-1})$, where $z_t \in [N+1]$ is the token at position $t$, $\mathbf{W}_E(z_t)$ is its embedding and $\widetilde{\mathbf{W}}_E(z_{t-1})$ is a different embedding of the previous token to a different direction, as in the *previous token head* construction of Bietti et al. (2023), where the value matrix remaps the previous token to a different subspace. We assume all embeddings to be orthogonal (Assumption D.1), which requires large enough $d$, and holds in the infinite-width limit with random embeddings. This model allows us to simplify our analysis by considering a single attention layer with no positional embeddings, while capturing the difficulty of long-range interactions. We note that such a simplification is standard in the in-context learning literature (e.g., Akyürek et al., 2023; Mahankali et al., 2024; Zhang et al., 2024), For data generation, $\pi_u$ and $\pi_b$ are uniform distributions on $[N]$. Given a sequence of inputs, $x_{1:T} \in \mathbb{R}^{T \times d}$, the output of model is $\xi \triangleq \xi_{\text{attn}} + \xi_{\text{ff}}$ as

$$x_t \triangleq \mathbf{W}_E(z_t) + \widetilde{\mathbf{W}}_E(z_{t-1}) \in \mathbb{R}^d,$$
$$\phi(x_T, x_{1:T}) \triangleq \sum_{t \leq T} \left[ \sigma \left( x_T^\top \mathbf{W}_{QK} x_{1:T} \right) \right]_t \cdot \mathbf{W}_V x_t \in \mathbb{R}^d,$$
$$\xi_{\text{attn}}(x_{1:T}) \triangleq \mathbf{W}_U \phi(x_T, x_{1:T}) \in \mathbb{R}^{N+1}, \quad (2)$$
$$\xi_{\text{ff}}(x_{1:T}) \triangleq \mathbf{W}_U F(x_T) = \mathbf{W}_U \mathbf{W}_F x_T \in \mathbb{R}^{N+1},$$

where $\mathbf{W}_U \in \mathbb{R}^{(N+1) \times d}$ is the unembedding matrix, $\phi(s, t)$ is the attention module with query $s$ and context $t$, and $F(\cdot)$ is a linear feed-forward layer. This architecture is similar to a one-layer transformer, but already highlights the difference between feed-forward and attention layers in a way that we expect to still hold for more layers. In the above parametrization, the learnable matrices are $\mathbf{W}_{QK}, \mathbf{W}_F, \mathbf{W}_V \in \mathbb{R}^{d \times d}$. At initialization, we set $\mathbf{W}_{QK}, \mathbf{W}_F, \mathbf{W}_V = 0$, noting that random initialization in high dimension would lead to similar behaviors thanks to near-orthogonality.

### 3.1 FEED-FORWARD LAYERS STORE THE GENERIC NOISE

As we saw in Figure 3 and 8, the model very quickly learns to predict the noise token after a few steps. Then the gap between $\rho = 0$ and 1 in Figure 3 suggests that the feed-forward layer $F_2$ is responsible for storing the distributional association about noise, which is verified in Figure 7 (middle). We now provide theoretical justification for this behavior. In particular, we will show that, at initialization, the gradients over the feed-forward parameters are much more informative than the attention gradient, which is dominated by noise unless the sample size is very large. This shows that the feed-forward layer is much more likely to capture the distributional association.

We now look at the first gradient step from initialization, which has commonly been used to understand feature learning and sample complexity in neural networks (Damian et al., 2022; Ba et al., 2022; Dandi et al., 2023; Oymak et al., 2023; Bietti et al., 2023). Note that $\mathbf{W}_{QK}$ has no gradient at initialization, so that the gradient of $\mathbf{W}_V$ is most relevant initially (see also Snell et al., 2021; Li et al., 2023; Oymak et al., 2023; Bietti et al., 2023).

**Theorem 1** (Logits after one gradient step). *Assume $N, T \gg 1, \alpha = \Theta(1)$. For the model in Eq(2), consider one gradient step update from zero-initialization on $m$ i.i.d. samples of $z_{1:T}$ with separate learning rates $\eta_f$ for $\mathbf{W}_F$ and $\eta_v$ for $\mathbf{W}_V$ (note that the gradient on $\mathbf{W}_{QK}$ is zero). With probability $1 - \delta$, the resulting logits for the feed-forward and attention blocks satisfy, for any test sequence $z_{1:T}$,*

$$|\Delta(\xi_{ff}(x_{1:T})) - \eta_f \cdot \alpha| \leq \eta_f \cdot O\left(\sqrt{\frac{\ln \frac{2(N+1)}{\delta}}{m}}\right),$$

$$\left|\Delta(\xi_{attn}(x_{1:T})) - \frac{\eta_v}{N} \cdot \hat{\alpha}\right| \leq \eta_v \cdot O\left(\sqrt{\frac{\left(\frac{1}{TN} + \frac{1}{N^2}\right)\ln \frac{2(N+1)}{\delta}}{m}} + \frac{\ln \frac{2(N+1)}{\delta}}{m}\right),$$

*where $\Delta(\xi) = \xi_{N+1} - \max_{j \in [N]} \xi_j$ is the margin of predicting the generic noise token and $\hat{\alpha} = (\alpha^2 \hat{q} + \alpha(1 - \hat{q}))$, where $\hat{q} = \frac{1}{T}\sum_{t \leq T} \mathbb{1}\{z_t = N + 1\}$ is the fraction of noise tokens in $z_{1:T}$.*

The margin $\Delta(\xi)$ reflects how much signal there is in the logits for predicting the noise token, and the theorem provides concentration bounds on the contributions of the updates on $\mathbf{W}_F$ and $\mathbf{W}_V$ to the margin. Note that $\hat{q} \ll 1$ w.h.p. for large $N, T$, so $\hat{\alpha} \approx \alpha$. We make the following observations:

  i. When $m = \tilde{\Omega}(1)$, there is enough signal in $\mathbf{W}_F$ to predict the noise, say with $\eta_f = 1$, and a choice of $\eta_v = O(1)$ will lead to a small but controlled contribution to the prediction from $\mathbf{W}_V$.

  ii. When $m = \tilde{\Omega}(N)$, $\mathbf{W}_V$ can also reliably predict the noise by setting $\eta_v = \Theta(N)$ (i.e., with small deviation on the r.h.s.), at the cost of many more samples.

Our result shows that in the initial phase of training, feed-forward layers are more likely to pick up the noise token, leading to a structure of the form $\mathbf{W}_F \approx \mathbf{W}_U(N + 1)\mathbf{W}_E(q)^\top$, while attention will be slower due to additional noise and possibly smaller step-sizes. We may then expect the attention layers to focus instead on in-context reasoning, as we observe empirically and discuss next.

### 3.2 ATTENTION ATTENDS TO IN-CONTEXT TARGETS AND AVOIDS NOISE

When the feed-forward weight learns to predict the noise as shown in Theorem 1, Figure 4 reveals that the second-layer attention in the two-layer model attends only towards the correct tokens. In contrast, a model pre-trained without noise has second-layer attention attend towards all tokens just after the triggers (Bietti et al., 2023), as observed in the attention pattern at the first step in Figure 4(right). Then, after being fine-tuned on noise data, the attention becomes only focused on the correct tokens. Understanding this mechanism requires the analysis of the dynamics of $\mathbf{W}_{QK}$.

Following the simplified model and data distribution in Eq(2), we take a step towards understanding how attention "avoids" the noise tokens. Concretely, this mechanism appears because, after the initial training phase when FF learns noise association much faster than the attention, $\mathbf{W}_V$ has a structure of $\sum_{k \leq N+1} \mathbf{W}_U(k)(\mathbf{W}_E(k) + \widetilde{\mathbf{W}}_E(k))^\top$, similar to the non-noise setting in Bietti et al. (2023). After such a $\mathbf{W}_V$ is learned, the trigger-label association provides a stronger gradient signal on $\mathbf{W}_{QK}$ than the trigger-noise association. We show this in the following theorem.

**Theorem 2** (Attention attends to in-context targets). *Assume $N, T \gg 1$ and Assumption F.1 hold. Consider the simplified model in Eq(2) with infinite samples as $m \to \infty$. After $\mathbf{W}_F$ learns the noise association as in Theorem 1, in one step the attention weight $\mathbf{W}_{QK}$ learns to attend to positions $t \in [T]$ where the correct label follows a trigger word, i.e., $z_{t-1} = q, z_t = \bar{y}$.*

*More concretely, $\mathbf{W}_{QK}$ has the following structure*

$$\xi_{q \to j} - \xi_{k \to l} = \Omega(N^{-3}) > 0, \quad \text{if } k \neq q, \ \forall j, l, \tag{3}$$

$$\xi_{q \to j} - \xi_{q \to N+1} = \Omega(N^{-4}) > 0, \quad \forall j \leq N, \tag{4}$$

*where $\xi_{i \to j} \triangleq \mathbf{W}_E(q)^\top \mathbf{W}_{QK}(\widetilde{\mathbf{W}}_E(i) + \mathbf{W}_E(j))$ denotes the attention logit for different combinations of $z_{t-1} = i, z_t = j$, with $i, j \leq N + 1$.*

Note that a set of logits induces a probability distribution via differences between them as $\exp(\xi_i) / \sum_j \exp(\xi_j) = 1 / \sum_j \exp(\xi_j - \xi_i)$. Therefore, the above theorem reveals that the attention has two patterns: Eq. (3) shows that $\mathbf{W}_{QK}$ prefers attending to locations just after a trigger $q$, *i.e.*, such that $z_{t-1} = q$, similar to Bietti et al. (2023), and Eq. (4) shows that among all positions that follow a trigger $q$, $\mathbf{W}_{QK}$ places less attention on the noise token, *i.e.*, $z_t = N + 1$, compared to correct tokens $z_t = \bar{y} \leq N$. Such a key difference for attention between noisy and non-noise tasks verifies our experimental observations in Figure 4.

### 3.3 NO FEED-FORWARD LAYERS: VALUE MATRIX STORES GENERIC NOISE ASSOCIATION

In the above discussion, we've seen separate roles of attention and feed-forward layers play to conduct noisy in-context learning. A natural question is, when there is *no feed-forward layer*, how the attention layer stores both in-context and distributional information. Figure 13 indicates that the value matrix stores the noise association in subspace with smaller singular values. In this section, we propose a setting of *linear associative memory with noise* to understand this mechanism.

Unlike Theorem 1 and 2 showing the separate roles of attention and FF, the attention in a non-FF model has to handle both noise and in-context information once the model is sufficiently trained to reach a global minimum. Due to symmetry from uniformly random sampling $\bar{y}$ from $N$ tokens, we consider passing the output $x \in \mathbb{R}^d$ of the attention to the value matrix $\mathbf{W}_V$ and output matrix $\mathbf{W}_U$ to predict next-token probability $y \in \mathbb{R}^{N+1}$ given $z_{1:T} \in [N+1]^T$ with noise probability of $\alpha$ as follows

$$x | \bar{y}, z_{1:T} \triangleq \mathbf{W}_E(\bar{y}) + \overline{\mathbf{W}}(z_{1:T}) \in \mathbb{R}^d, \quad \xi \triangleq \mathbf{W}_U \mathbf{W}_V x \in \mathbb{R}^{N+1},$$
$$p_\alpha(y | \bar{y}) = (1 - \alpha) \cdot \mathbb{1}\{y = \bar{y}\} + \alpha \cdot \mathbb{1}\{y = N + 1\}, \tag{5}$$

where $\overline{\mathbf{W}}(z_{1:T})$ is an aggregate embedding independent of $\bar{y}$. When $T \to \infty$, $\overline{\mathbf{W}}(z_{1:T})$ converges to a fixed embedding $\overline{\mathbf{W}}$ independent of $\bar{y}$, so that we may consider a simplified model $x | \bar{y} \triangleq \mathbf{W}_E(\bar{y}), \xi \triangleq \mathbf{W} x \in \mathbb{R}^{N+1}$ with $\mathbf{W} \in \mathbb{R}^{(N+1) \times d}$, since $\overline{\mathbf{W}}$ only contributes a fixed offset in all logits that can be easily canceled in the softmax predictions. Therefore, we investigate the following *linear associative memory with noise*.

**Model and data.** Consider a learnable weight matrix $\mathbf{W} \in \mathbb{R}^{d \times d}$ with $d > N$. Consider embeddings for $N$ input tokens as $\{e_i\}_{i=1}^N \subset \mathbb{R}^d$ and embeddings for $(N + 1)$ output tokens as $\{u_i\}_{i=1}^{N+1} \subset \mathbb{R}^d$. Given any pair of input and output tokens, the associative memory model takes the form $f(i, j; \mathbf{W}) \triangleq \langle u_j, \mathbf{W} e_i \rangle, \ \forall i, j \in [N] \times [N + 1]$, as logits to approximate $p_\alpha(\cdot | i)$ in (5). When $k \leq d$, we denote the rank-$k$ approximation of $f$ as $f^{(k)}$ by replacing $\mathbf{W}$ with $\mathbf{W}^{(k)}$, where $\mathbf{W}^{(k)}$ is its rank-$k$ approximation.

**Experiments.** During training, the dataset $\mathcal{D}_\alpha$ is generated with non-zero noise probability $\alpha > 0$. At test time, the dataset $\mathcal{D}_0$ is without noise as $\alpha = 0$, so the computed loss is called **pure-label** loss. The *full* model is trained with Gradient Descent (GD) subjected to cross-entropy loss. The results are reported in Figure 18, with more discussions in Appendix G.1.

**Low-rank subspace stores noise.** In Figure 18, the rank-1 subspace corresponding to the smallest non-zero singular value is responsible to store the noise. We prove this mechanism as follows. Note that, here $N = 2$ is for simplicity, which is easy to extend to any $N > 2$.

**Theorem 3.** *Assume Assumptions G.1 and G.2 hold, considering $N = 2$ and $\alpha \in (0.2, 0.4)$, we train the full model $f(\cdot, \cdot; \mathbf{W})$ with gradient flow. Denote $P(i, j; \mathbf{W})$ as the model's predicted probability*

*for output $j$ conditioned on input $i$. Then, for $t \to \infty$ and $i \in \{1, 2\}$, we have*

$$P(i, j; \mathbf{W}) = (1 - \alpha) \cdot \mathbb{1}\{j = i\} + \alpha \cdot \mathbb{1}\{j = N + 1\},$$
$$P(i, j; \mathbf{W}^{(1)}) = (1 - \Theta(t^{-1/2})) \cdot \mathbb{1}\{j = i\} + \Theta(t^{-1/2}) \cdot \mathbb{1}\{j = N + 1\}.$$

The above theorem implies, the full model always predicts noise w.p. $\alpha$, while the rank-1 model eventually predicts correctly without noise, although training is only on the full model with noise. Actually when $N > 2$, the noise is stored in rank-1 subspace and the correct correspondence is stored in rank-$(N - 1)$ space. Therefore, this explains how the value matrix stores both in-context and noise information when the model is without FF.

## 4 EXPERIMENTS

In this section, we empirically investigate how LLMs process distributional vs in-context associations, and how this evolves during training. Meanwhile, we provide numerical results of how much low-rank truncation improves complex reasoning on a real-world reasoning benchmark, GSM8K. Appendix J provides another synethetic IOI dataset that requires counting tokens.

### 4.1 AN INVESTIGATION ON GPT-2 SMALL AND PYTHIA MODELS

We consider GPT-2 small and Pythia models on the indirect object identification (IOI) and factual recall tasks described in Section 2.1.

**Quick demonstration: IOI on GPT2 Small.** Different from Wang et al. (2022), we would like to consider whether a model proposes an output beyond the input $x$. A quick demonstration is to consider the IOI task with input $x =$ "When Mary and John went to a store, John gave a drink to"[1]. The top 4 predicted tokens for GPT-2 Small (Radford et al., 2019) on $x$ are ["Mary", "them", "the", "John"]. Although GPT-2 Small successfully predicts Mary (the IO target) instead of John (S), the other two top candidate tokens, *i.e.*, "them" and "the", do not even appear in the context. This prominence of such "generic" words is similar to the factual recall example from Section 2.2, and plausibly follows from a distributional associative mechanism conditioned on the preposition "to".

**Comprehensive experiment: IOI on Pythia-1B.** Now we would like to verify this observation on more models and, more comprehensively, track the behavior of these models along training. We choose to conduct the IOI experiments on Pythia (Biderman et al., 2023), a family of models ranging in sizes from 14M to 12B trained on web data, with hundreds of training checkpoints for each size. We generate an IOI dataset of 100 sentences with random names for [IO] and [S] in each sample. Figure 5 reports the test results of Pythia-1B along training. Here LASER is conducted on MLP weights, with parameters given in Appendix C.2. LASER boosts the probability ratio of [IO] over "the" from $2.3\times$ to $12.3\times$ at 14K steps.

**Factual recall on Pythia-1B.** As in Table 1, we verify factual recall with input as "Madrid is located in". The full model of Pythia-1B generates "Madrid is located in the north of Spain", while the model after LASER generates "Madrid is located in Spain". We track the probability of predicting "Spain" and "the" along training in Figure 5. LASER turns out to boost the probability ratio of "Spain" over "the" from $0.16\times$ to $11.3\times$ at 14K steps. We note that better prompting could avoid the need for LASER in this case (e.g., "Madrid is located in the country of" predicts "Spain"), but increases the context length and thus the inference cost, though this is outside the scope of this paper.

**Training dynamics on Pythia.** The behavior of the Pythia models on the IOI and factual recall tasks during their pre-training process displays several phases, as shown in Figure 5. For IOI, we observe:

  i. Initialization: all tokens have similar logits since the weights are random initialized.
  ii. Between 10 and 1000 steps: the models consistently output "the". They cannot solve IOI task at all, as long as they have almost the same prediction for [IO] and [S]. After 500 steps, [IO] starts the growth towards one of the top predictions.
  iii. After 2000 steps: Pythia starts to be able to solve IOI task by preferring [IO] than [S] and "the". Meanwhile, the benefit of LASER appears as enhancing the leading position of [IO].

---

[1]Note that here we use "a" store instead of "the" store in the original example of Wang et al. (2022). The reason is to rule out the word "the" from the input context.

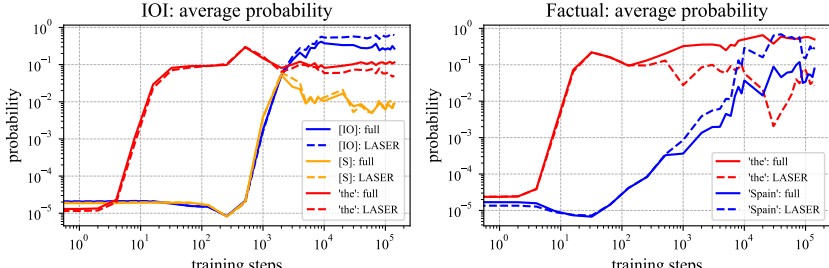

Figure 5: **Left**: average probability of tokens [IO], [S] and "the" in 100-sentence IOI task in the prediction by Pythia-1B along training. **Right**: average probability of tokens "Spain" and "the" in a factual task predicted by Pythia-1B along training, with input as "Madrid is located in". In both tasks, the full model learns to predict "the" with high probability starting from ∼10 steps, and then learns to solve the tasks. LASER boosts the probability of correct answers against "the" in both tasks: the average probability ratio of correct answers against "the" improves from 2.3× to 12.3× (in IOI) and from 0.16× to 11.3× (in factual) at 14K steps.

Therefore, the training process reveals the capacity of predicting "the" is learnt much earlier than predicting [IO]. The reason might be that predicting "the" requires a simpler grammar structure, while predicting [IO] requires a complicated architecture of attention heads of different roles across layer (Wang et al., 2022). Then we note that the IOI task always has "to" before the masked [IO], which means "to" may be an indicator for the model to predict "the" with non-negligible probability. Similarly, for factual recall we see early learning of the "generic" answer, while the factual answer is learned later. Conceptually, if LLMs are able to write natural text or have been trained sufficiently with natural texts, it is not surprising for the model to predict "the" with high probability after seeing "to". This is verified in Appendix C.1.

## 4.2 THE EFFECT OF TRUNCATING FEED-FORWARD LAYERS ON GSM8K

As our previous examples of in-context reasoning tasks are too simple for real-world reasoning, we verify whether truncating MLPs improves reasoning on the GSM8K benchmark (Cobbe et al., 2021). As shown in Table 2, LASER improves the few-shot Chain-of-Thought (Wei et al., 2022) reasoning performance on GSM8K when only using 1 or 2 shots, although the performance is worse in the standard 8-shot setting. This suggests that truncating MLPs may help promote in-context reasoning even in more complex settings, perhaps by removing spurious distributional associations.

Table 2: Few-shot accuracy (%) of pretrained and finetuned language models on GSM8K. Truncating MLPs (LASER) improves reasoning performances in few-shot CoT settings while it has worse performance in the standard 8-shot setting. The LASER hyper-parameters are in Appendix C.2.

|                                        | 1-shot | 2-shot | 4-shot | 8-shot (standard) |
|----------------------------------------|--------|--------|--------|-------------------|
| Phi-3 (Abdin et al., 2024)             | 56.0   | 72.2   | 78.2   | 82.7              |
| Phi-3 + LASER                          | **66.1** | **74.4** | 77.0 | 82.3            |
| Llama-3.1-8B (AI@Meta, 2024)           | 44.7   | 50.0   | 57.6   | 56.0              |
| Llama-3.1-8B + LASER                   | **46.1** | **50.7** | 55.9 | 53.8            |
| Llama-3.1-8B-Instruct (AI@Meta, 2024)  | 72.6   | 74.7   | 78.5   | 79.7              |
| Llama-3.1-8B-Instruct + LASER          | **73.6** | **75.6** | 77.7 | 77.0            |

## 5 DISCUSSION AND LIMITATIONS

In this paper, we studied the questions of how transformer language models learn to process distributional associations differently than in-context inputs, and how truncating specific weights or layers, particularly feed-forward layers, can help in-context reasoning. While our work provides some initial theoretical understanding of how this may arise on simple controlled settings, it would be interesting to study how these ideas may extend to more complex tasks where in-context reasoning and distributional knowledge interact in more intricate ways.

## ACKNOWLEDGEMENTS

We are grateful to Yifang Chen, Ekin Akyürek and Denny Wu for helpful discussions. This work was done in part while AB was visiting the Simons Institute for the Theory of Computing. LC and JB were supported by the Alfred P. Sloan Foundation, and awards NSF RI-1816753, NSF CAREER CIF 1845360, NSF CHS-1901091 and NSF DMS-MoDL 2134216. This work was supported in part through the NYU IT High Performance Computing resources, services, and staff expertise. We thank anonymous reviewers for feedbacks.

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

CONTENTS

## A  CONTRIBUTIONS AND IMPLICATIONS

Our contribution focuses on understanding the different roles of attention and FF weights in disentangling distributional vs in-context associations, both empirically and theoretically. The application of low-rank truncation is simply a way to verify our claims, and is consistent with the findings in the LASER paper that truncating some FF layers may improve performance on some reasoning tasks.

Nevertheless, our perspective based on distributional associations versus in-context reasoning may be helpful in thinking about how to allocate parameters to feed-forward versus attention layers: for instance, in Figure 6 on our synthetic task, we found that for a fixed total parameter budget, models with fewer MLP parameters achieve higher loss on distributional predictions (e.g., non-contextual bigrams) compared to models with more MLP parameters (and fewer attention parameters). These notions may also provide a different way to reason about circuit discovery in mechanistic interpretability from the perspective of training dynamics and properties of the training data. Finally, this disentanglement may inform more effective ways to fine-tune models, e.g., by selectively choosing which layers to fine-tune.

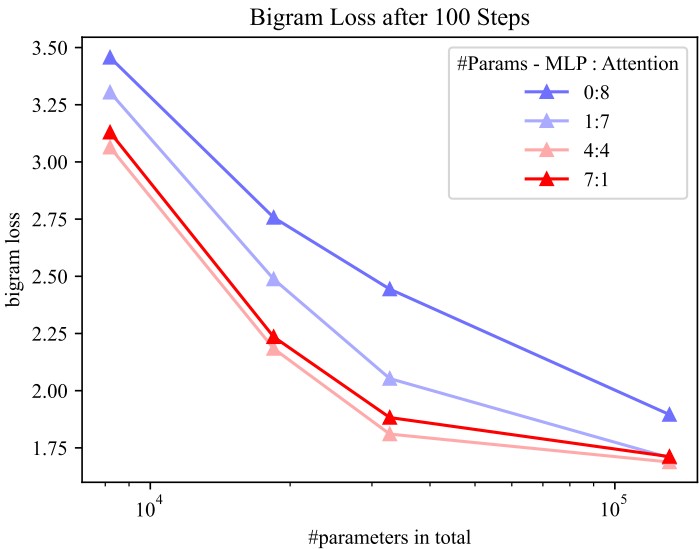

Figure 6: The training loss of approximating the global bigram $\pi_b$ with various allocations of parameters in MLP and Attentions. For each configuration of total parameters and ratios, we use the corresponding best learning rate after search to train 100 steps.

## B  HOW DOES THE TWO-LAYER MODEL SOLVE NOISY IN-CONTEXT RECALL?

### B.1  TRAINING SETTINGS

In most parts of this work, we consistently train the model with a fixed level of $\alpha > 0$. However, we also present numerical results of **fine-tuning** in Figure 8 and 4 to show the mechanism of avoiding generic noise token in the second-layer attention. The details of such a fine-tuning setting is as follows.

**Fine-tuning**: there are two phases of training as

- phase 1 (pre-training): starting from a model with random initialized weights, we train the model on data generated with $\alpha = 0$. This is exactly the same as Bietti et al. (2023). At the end of this phase, the second-layer attention is expected to attend *all tokens* after the trigger token, *i.e.*, $t \leq T$ such that $z_{t-1} = q$ no matter what $z_t$ is.

- phase 2 (fine-tuning): starting from a model after phase 1, we train all weights in the model on data generated with $\alpha > 0$. At the end of this phase, the second-layer attention learns to avoid the generic noise token, *i.e.*, $t \leq T$ such that $z_t = N_1, z_{t-1} = q$, as shown in Figure 4.

## B.2 Summarizing: roles of key components in the two-layer transformer

Recall the architecture of two-layer transformers in Section 3 as

$$x_t \triangleq \mathbf{W}_E(z_t) + p_t,$$
$$h_t^1 \triangleq \sum_{s \leq t} \left[ \sigma(x_t^\top \mathbf{W}_{QK}^1 x_{1:t}) \right]_s \cdot \mathbf{W}_V^1 x_s,$$
$$x_t^1 \triangleq x_t + h_t^1 + F_1(x_t + h_t^1),$$
$$h_t^2 \triangleq \sum_{s \leq t} \left[ \sigma(x_t^{1\top} \mathbf{W}_{QK}^2 x_{1:t}^1) \right]_s \cdot \mathbf{W}_V^2 x_s^1,$$
$$x_t^2 \triangleq x_t^1 + h_t^2 + F_2(x_t^1 + h_t^2),$$
$$\xi_t \triangleq \mathbf{W}_U x_t^2.$$

When the task is without noise, *i.e.*, $\alpha = 0$, Bietti et al. (2023) point out the first-layer attention attends to the previous token through $\mathbf{W}_{QK}^1 = \sum_{t=2}^T p_{t-1} p_t^\top$. Therefore, when $z_t = \bar{y}$ with $z_{t-1} = q$, the output of the first layer is $x_t^1 \approx \mathbf{W}_E(\bar{y}) + \mathbf{W}_V^1 \mathbf{W}_E(q)$. Then they show that the second-layer attention matches such $x_t^1$ with $z_T = q$ by $\mathbf{W}_{QK}^2 = (\mathbf{W}_V \mathbf{W}_E(q)) \mathbf{W}_E(q)^\top$, through which the information of $\bar{y}$ in $x_t^1$ is copied to last token as $h_T^2 \approx \mathbf{W}_V^2 \mathbf{W}_E(\bar{y})$. Finally $\mathbf{W}_V^2 = \sum_{z \in [N]} \mathbf{W}_U(z) \mathbf{W}_E(z)^\top$ helps output the correct label of $\bar{y}$.

In our work with noise $\alpha > 0$, the key difference is that there is a fixed probability $\alpha$ for a noise token $N + 1$ to appear after each trigger $q$. This requires $\mathbf{W}_{QK}^2$ to not only match the trigger but also avoid the noise token after trigger. Let's first summarize the whole pipeline of this model for our task.

**Roles of key components.** The first layer will be basically the same as Bietti et al. (2023), where $\mathbf{W}_{QK}^1 = \sum_{t=2}^T p_{t-1} p_t^\top$ attends to the previous token. Consider two positions $t_1, t_2$ with $z_{t_1-1} = z_{t_2-1} = q, z_{t_1} = \bar{y}, z_{t_2} = N + 1$, then outputs of the first layer at these two positions are $x_{t_1}^1 \approx \mathbf{W}_E(\bar{y}) + \mathbf{W}_V^1 \mathbf{W}_E(q)$, $x_{t_2}^1 \approx \mathbf{W}_E(N + 1) + \mathbf{W}_V^1 \mathbf{W}_E(q)$. Then the second-layer attention $\mathbf{W}_{QK} = (\mathbf{W}_V \mathbf{W}_E(q) - c \cdot \mathbf{W}_E(N + 1)) \mathbf{W}_E(q)^\top$ with some positive $c$ makes the attention attend to $t_1$ and avoid $t_2$ simultaneously, matching with the last token $z_T = q$. Therefore, the output of the second-layer attention at $T$ is basically $h_T^2 \approx \mathbf{W}_V^2 \mathbf{W}_E(\bar{y})$. Similar to the noiseless case, $\mathbf{W}_V^2 = \sum_{z \in [N]} \mathbf{W}_U(z) \mathbf{W}_E(z)^\top$ helps output the correct label of $\bar{y}$. Meanwhile, note that $x_T^1$ actually contains $\mathbf{W}_E(q)$ through $x_T$, so $F_2$ is able to predict the noise $N + 1$ when seeing a fixed $\mathbf{W}_E(q)$. As a result, combining the two streams from $h_T^2$ and $F_2(x_T^1)$, the full model is able to predict any $\bar{y}$ w.p. $1 - \alpha$ and predict the noise $N + 1$ w.p. $\alpha$.

**Evidence.** Figure 4 illustrates that the second-layer attention learns to attend to $z_{t_1} = \bar{y}$ and avoid $z_{t_2} = N + 1$, with Appendix B.3 presenting a primitive exploration on how the avoidance is learnt in a simplified setting. Figure 7 (left) shows the attention pattern from $\mathbf{W}_{QK}^1$ of attending to the previous token. Figure 7 (middle) shows the memory recall of $\mathbf{W}_U(N + 1)^\top F_2(\mathbf{W}_E(q))$ to predict the noise. Figure 7 (right) illustrates the memory recall of $\mathbf{W}_U(i)^\top \mathbf{W}_V^2 \mathbf{W}_E(i)$ to predict the correct token.

## B.3 How does attention attend less towards the noise token?

We use the same simplified model as in Section 3.1 to understand how the second-layer attention learns to avoid the noise. When using the same learning rate $\eta = \eta_v = \eta_f$, Theorem 1 implies that the feed-forward $\mathbf{W}_F$ makes the most contribution for predicting the noise after the first-step update. Denote the logits for the noise of the model at time $t$ as $\xi_t$. The arguments in this section make the following assumptions, which hold at least after the first-step update:

i. $\mathbf{W}_F$ dominates the logits $\xi_t$ of predicting the noise token, compared with $\mathbf{W}_V$.

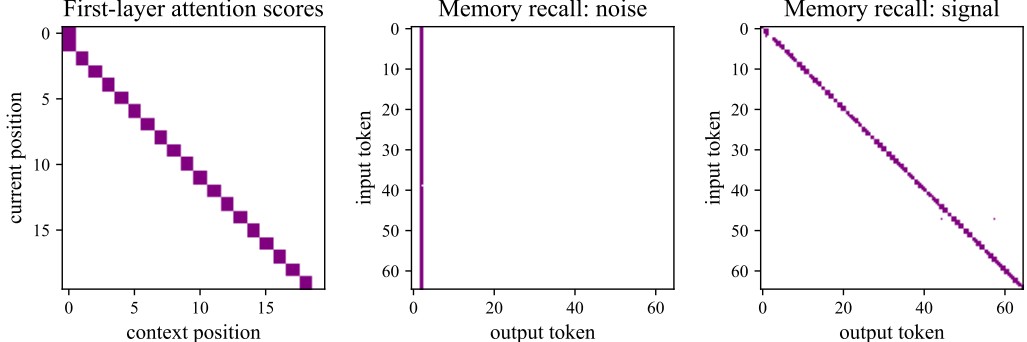

Figure 7: **Left**: first-layer attention attending to the previous token from the current token. **Middle**: logits to predict noise from $\langle F_2(\mathbf{W}_E(i)), \mathbf{W}_U(j)\rangle$ with input $i \in [N+1]$ and output $j \in [N+1]$, where the output channel 2 is set as the noise channel. It turns out, for all input $i$, the logits on output 2 are large, which matches our construction that, at least for trigger $q$ as input, the output 2 has large logits. **Right**: logits to predict singal from $\langle \mathbf{W}_V^2 \mathbf{W}_E(i), \mathbf{W}_U(j)\rangle$ for input $i \in [N+1]$ and output $j \in [N+1]$. It matches our construction that $i = j$ has large logits. Meanwhile, $i = j = 2$ does not have large logits since 2 is the noise channel.

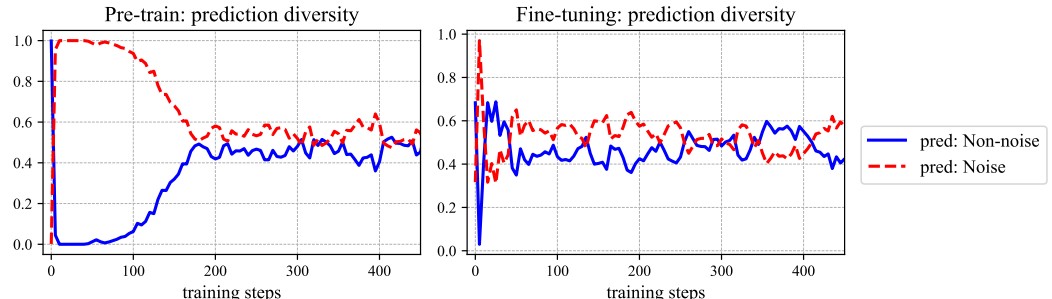

Figure 8: Fractions of predicting the noise token and the other non-noise tokens with $\alpha = 0.5$. (Left) pretraining steps on noisy data; (right) finetuning steps on noisy data, after pretraining on clean data with $\alpha = 0$. In both cases, the models learn to predict noise with probability nearly 0.5. In the first few ($\sim 5$) steps, the models quickly learn to predict noise with probability close to 1. The fine-tuning setting is in Appendix B.1.

    ii. Logits for predicting any $k \le N$ is close to 0, which means the predicted probability $p_t$ is approximately $p_t \approx \frac{\exp(\xi_t)}{N + \exp(\xi_t)}$.

   iii. The predicted probability $p_t < \alpha$.

   iv. The attention matrix $\mathbf{W}_{QK}$ is approximately 0, inducing a uniform attention.

   v. The dataset has $T, N \gg 1$ and $m \to \infty$, so the gradient is from population loss.

The first assumption holds after the first step from Theorem 1 with $\eta_f = \eta_v$.

Then, since $|\mathbf{W}_U(k)^\top (\nabla_{\mathbf{W}_F} L) \mathbf{W}_E(q)| = O(\frac{1}{N}) \cdot |\mathbf{W}_U(N+1)^\top (\nabla_{\mathbf{W}_F} L) \mathbf{W}_E(q)|$ for any $k \le N$ in Lemma D.1, the second assumption holds. Meanwhile, the projection of $\nabla_{\mathbf{W}_V} L$ onto any direction in Lemma D.2 is also smaller than $\mathbf{W}_U(N+1)^\top (\nabla_{\mathbf{W}_F} L) \mathbf{W}_E(q)$ by a factor of $O(1/N)$.

Let's check the condition of the third assumption. In the proof of Lemma D.1, the gradient of $\mathbf{W}_F$ has the form of

$$\mathbf{W}_U(N+1)^\top (-\nabla_{\mathbf{W}_F} L) \mathbf{W}_E(q) = \alpha - p_t.$$

This update induces $\xi_t$ to increase by $\eta(\alpha - p_t)$. This implies

$$\xi_t \approx \xi_{t-1} + \eta\left(\alpha - \frac{\exp(\xi_t)}{N + \exp(\xi_t)}\right), \quad \forall\, t \geq 1.$$

This sequence $\{\xi_t\}_{t\geq 1}$ has stationary point $\xi^* = \log N + \log(\frac{\alpha}{1-\alpha})$. Denoting $\hat{\xi}_t \triangleq \xi_t - \xi^*$ with $\hat{\xi}_1 = -\xi^* < 0$, the iteration becomes

$$\hat{\xi}_{t+1} \approx \hat{\xi}_t + \eta\left(\alpha - \frac{\exp(\hat{\xi}_t)}{\frac{1-\alpha}{\alpha} + \exp(\hat{\xi}_t)}\right).$$

If we would like to have $\hat{\xi}_t$ not hit the positive region by controlling $\eta$, it suffices to bound $\eta$ with any $\hat{\xi} < 0$,

$$\eta \leq \frac{\hat{\xi}}{\frac{\exp(\hat{\xi})}{\frac{1-\alpha}{\alpha} + \exp(\hat{\xi})} - \alpha},$$

where RHS is continuous and decreasing on $\xi < 0$ when $\alpha < 0.5$. Hence, we have $\eta \leq \frac{1}{\alpha(1-\alpha)}$ evaluated at $\hat{\xi} = 0$ by L'Hospital rule. This bound of $\eta$ is very strong, since $\eta = O(\log N)$ can still have $\hat{\xi} < 0$ after one step.

The fourth assumption is basically from what we will show at the end of this section, as the second observation.

Then consider the dynamics of $\mathbf{W}_V$, which is much slower than $\mathbf{W}_F$. From the proof of Lemma D.2, the gradient of $\mathbf{W}_V$ satisfies

$$\nabla_{\mathbf{W}_V} L = \mathbb{E}_x\left[\sum_{k=1}^{N+1}(p_{\mathbf{W}}(k|x) - \mathbb{1}\{y=k\})\mathbf{W}_U(k)\left(\frac{1}{T}\sum_{t=1}^{t} x_t\right)^\top\right],$$

$$\mathbf{W}_U(N+1)^\top(-\nabla_{\mathbf{W}_V} L)\mathbf{W}_E(k) \approx \frac{1}{N}\sum_{t\geq 1}(\alpha - p_t)(\mathbb{1}\{k \leq N\} + \alpha \cdot \mathbb{1}\{k = N+1\})$$

$$\triangleq c \cdot \mathbb{1}\{k \leq N\} + c \cdot \alpha \cdot \mathbb{1}\{k = N+1\} = \Theta(\frac{1}{N}),$$

(6)

where the projection on $W_E(N+1)$ is always positive and smaller than that on other directions when $p_t < \alpha$. Projections onto other directions $\mathbf{W}_U(j)\mathbf{W}_E(k)^\top, \forall\, j \leq N$, are smaller as $\Theta(\frac{1}{N^2})$.

Finally, let's consider the dynamics of $\mathbf{W}_{QK}$. At initialization, $\mathbf{W}_{QK} = 0$ and $\nabla_{\mathbf{W}_{QK}} L = 0$ due to zero initialization of $\mathbf{W}_V$. After one-step, $\mathbf{W}_V$ has such a structure in Eq.(6). Then, with $\bar{x}_{1:T} \triangleq \frac{1}{T}\sum_{1\leq t\leq T} x_t$ from uniform attention, the gradient of $\mathbf{W}_{QK}$ satisfies

$$-\nabla_{\mathbf{W}_{QK}} L = \mathbb{E}_x\left[\sum_{k=1}^{N}(\mathbb{1}\{y=k\} - p_{\mathbf{W}}(k|x))\frac{1}{T}\sum_{t=1}^{T}(\mathbf{W}_U(k)^\top\mathbf{W}_V x_t)\cdot(x_t - \bar{x}_{1:T})\mathbf{W}_E(q)^\top\right]$$

$$\approx \sum_{k=1}^{N}\left(\frac{1-\alpha}{N} - \frac{1-p_t}{N}\right)\underbrace{\mathbb{E}\left[\frac{1}{T}\sum_{t=1}^{T}\mathbf{W}_U(k)^\top\mathbf{W}_V x_t \cdot (x_t - \bar{x}_{1:T})\mathbf{W}_E(q)^\top\right]}_{\triangleq A}$$

$$+ (\alpha - p_t)\underbrace{\mathbb{E}\left[\frac{1}{T}\sum_{t=1}^{T}(\mathbf{W}_U(N+1)^\top\mathbf{W}_V x_t)\cdot(x_t - \bar{x}_{1:T})\mathbf{W}_E(q)^\top\right]}_{\triangleq B}.$$

(7)

Then, we have

$$\mathbf{W}_E(N+1)^\top B \mathbf{W}_E(q) = \mathbb{E}\left[\frac{1}{T}\sum_{t=1}^{T}(\mathbf{W}_U(N+1)^\top \mathbf{W}_V x_t) \cdot \mathbf{W}_E(N+1)^\top (x_t - \bar{x}_{1:T})\right]$$

$$\stackrel{(a)}{=} \mathbb{E}\left[\frac{1}{T}\sum_{t=1}^{T}(c + c(\alpha-1)\cdot \mathbb{1}\{z_t = N+1\})\cdot \mathbf{W}_E(N+1)^\top (x_t - \bar{x}_{1:T})\right]$$

$$\stackrel{(b)}{=} \mathbb{E}\left[\frac{1}{T}\sum_{t=1}^{T}(c(\alpha-1)\cdot \mathbb{1}\{z_t = N+1\})\cdot \mathbf{W}_E(N+1)^\top (x_t - \bar{x}_{1:T})\right]$$

$$= \frac{\alpha}{N}\cdot c(\alpha-1)(1-\frac{\alpha}{N}) = \Theta(\frac{1}{N^2}) < 0.$$

where (a) is from Eq.(6), (b) is due to $\bar{x}_{1:T} = \frac{1}{T}\sum_t x_t$ and note that $c = \Theta(\frac{1}{N})$.

Similarly, we also have

$$\mathbf{W}_E(N+1)^\top A \mathbf{W}_E(q) = \mathbb{E}\left[\frac{1}{T}\sum_{t=1}^{T}(\mathbf{W}_U(k)^\top \mathbf{W}_V x_t)\mathbf{W}_E(N+1)^\top \cdot (x_t - \bar{x}_{1:T})\right]$$

$$= \mathbb{E}\left[\frac{1}{T}\sum_{t=1}^{T}\Theta(\frac{1}{N^2})\cdot \mathbb{1}\{z_t = N+1\}\mathbf{W}_E(N+1)^\top \cdot (x_t - \bar{x}_{1:T})\right] = \Theta(\frac{1}{N^3}).$$

For any $k \le N$, we have

$$\mathbf{W}_E(k)^\top B \mathbf{W}_E(q) = \mathbb{E}\left[\frac{1}{T}\sum_{t=1}^{T}(\mathbf{W}_U(N+1)^\top \mathbf{W}_V x_t)\cdot \mathbf{W}_E(k)^\top (x_t - \bar{x}_{1:T})\right]$$

$$= \mathbb{E}\left[\frac{1}{T}\sum_{t=1}^{T}(c(\alpha-1)\cdot \mathbb{1}\{z_t = k\})\cdot \mathbf{W}_E(N+1)^\top (x_t - \bar{x}_{1:T})\right]$$

$$= \frac{\alpha}{N}\cdot c(\alpha-1)(-\frac{1}{N}) = \Theta(\frac{1}{N^3}) > 0,$$

and

$$\mathbf{W}_E(k)^\top A \mathbf{W}_E(q) = \mathbb{E}\left[\frac{1}{T}\sum_{t=1}^{T}(\mathbf{W}_U(k)^\top \mathbf{W}_V x_t)\mathbf{W}_E(k)^\top \cdot (x_t - \bar{x}_{1:T})\right]$$

$$= \mathbb{E}\left[\frac{1}{T}\sum_{t=1}^{T}\Theta(\frac{1}{N^2})\cdot \mathbb{1}\{z_t = N+1\}\mathbf{W}_E(k)^\top \cdot (x_t - \bar{x}_{1:T})\right] = \Theta(\frac{1}{N^4}).$$

Combining the above four esimation of projections of $A$ and $B$ with Eq.(7), we have

$$\mathbf{W}_E(N+1)^\top (-\nabla_{\mathbf{W}_{QK}}L)\mathbf{W}_E(q) = \Theta(\frac{1}{N^2}) < 0,$$

$$\forall\, k \le N,\ \ \mathbf{W}_E(k)^\top (-\nabla_{\mathbf{W}_{QK}}L)\mathbf{W}_E(q) = \Theta(\frac{1}{N^3}) > 0.$$

Then we have three observations

i. $\mathbf{W}_{QK}$ in this phase avoids the noise token $N+1$ and uniformly attends to all tokens $k \le N$.

ii. The update of $\mathbf{W}_{QK}$ is in $\Theta(\frac{1}{N^2})$, while the update of $\mathbf{W}_F$ is $\Theta(1)$ in Lemma D.1 and that of $\mathbf{W}_V$ is $\Theta(\frac{1}{N})$ in Lemma D.2. These three levels of updating speed also coincide with the assumptions that $\mathbf{W}_F$ dominates first and then $\mathbf{W}_V$ has a micro structure that induces the evolving of $\mathbf{W}_{QK}$.

iii. The current proof for $\mathbf{W}_{QK}$ strongly depends on the fact that the noise token appears less than other token by a factor $\alpha$ in expectation. The proof will have the opposite result if the noise token is made to appear more by manipulating the data distribution. Therefore, we leave a new proof that is robust to such an assumption in data distribution as future work.

### B.4 MULTIPLE TRIGGERS

In Section 3, we assume there is only one fixed trigger $q \in [N]$ for simplicity. Actually the case of multiple triggers has the same mechanism. As discussed by Bietti et al. (2023) and Appendix B.2, for one trigger, the second-layer attention has large logits in $\langle \mathbf{W}_V^1 \mathbf{W}_E(i)^\top, \mathbf{W}_{QK}^2 \mathbf{W}_E(j) \rangle$ only for $i = j = q$. For multiple triggers, basically $\langle \mathbf{W}_V^1 \mathbf{W}_E(i)^\top, \mathbf{W}_{QK}^2 \mathbf{W}_E(j) \rangle$ only have large values when $q \in Q$. This is verified in Figure 9.

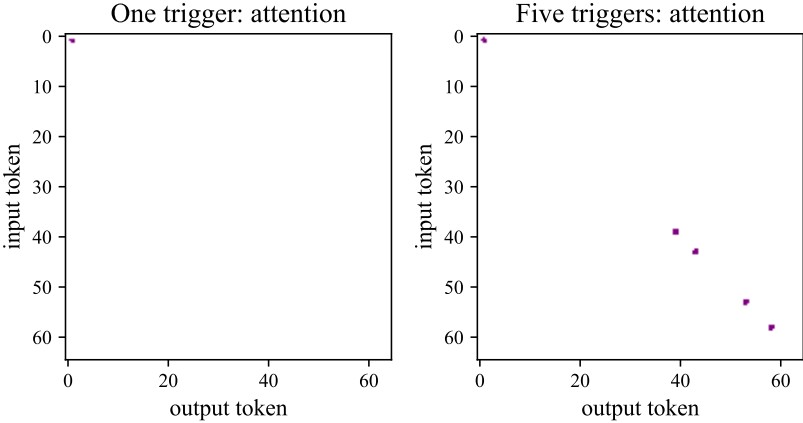

Figure 9: Logits of $\langle \mathbf{W}_V^1 \mathbf{W}_E(i)^\top, \mathbf{W}_{QK}^2 \mathbf{W}_E(j) \rangle$ for input $i$ and output $j$ when there is one trigger (left, $q = 1$) and five triggers (right, $q \in Q = \{1, 39, 43, 53, 58\}$). In both cases, the logits only have large values when $i = j = q$, verifies the matching mechanism in Appendix B.2.

### B.5 ARCHITECTURAL CHOICES

In Section 3 and Appendix B.2, we were focused on experiments with both $F_1, F_2$ being two-layer ReLU MLPs. Meanwhile, we have also tried other choices of $F_1, F_2$ and then search for the best truncation method for each architecture. In this section, we would like to summarize our experimental results for better understanding of all modules in the two-layer transformer.

Generally, the feed-forward layer can be two-layer ReLU MLPs, one-layer Linear or "None", where None stands for there is no feed-forward layer so that the value matrices in attention layers are the only weight matrices that transform features.

**Both $F_1, F_2$ are two-layer MLPs.** This is our main setting. The best truncation method is to *fully* drop $F_2$. We also try to fully drop $F_1$, as reported in Figure 10. It turns out fully dropping $F_1$ makes the model predict the noise with high probability.

**$F_1$ is MLPs and $F_2$ is Linear.** Figure 11 reports the results. Dropping $F_1$ and $F_2$ both improve the correct prediction, and dropping $F_1$ is better with lower test loss. Note that, when test accuracies are near $100\%$, lower test loss is a better measurement of the prediction quality, because accuracies are taken by argmax over the output logits while test loss are about the exactly predicted probability.

**$F_1$ is Linear and $F_2$ is MLPs.** Figure 12 reports the results. Dropping $F_2$ improves the correct prediction while dropping $F_1$ makes the model predict noise more.

**Both $F_1$ and $F_2$ are None.** Figure 13 reports the results. While there is no feed-forward layer any more, low-rank truncating a part $\mathbf{W}_O^1$ of the first-layer matrix improves the model's prediction a little. This implies that, when there is not feed-forward layers, the noise association is possible stored in the first-layer value matrix of attention. Note that the improvement of such low-rank truncation is clearly smaller than *fully* dropping one of feed-forward layers in the previous cases. Meanwhile, a smaller $\rho = 0.01$ destroys the model's performance. This implies fully dropping is not the optimal choice for low-rank truncation of the value matrix, and there is low-rank subspace in it that is useful for predicting the correct tokens. Our discussion of the role of $\mathbf{W}_V^1$ in Appendix B.2 is a possible answer to this phenomena.

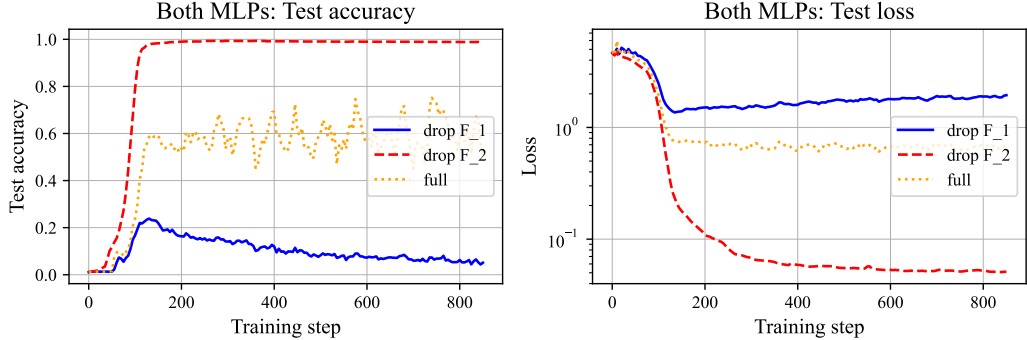

Figure 10: Test performance of fully dropping $F_1, F_2$ when both $F_1, F_2$ are two-layer MLPs. It turns out, while dropping $F_2$ makes the model predict correctly w.p. near 1, dropping $F_1$ has the model predict noise with high probability.

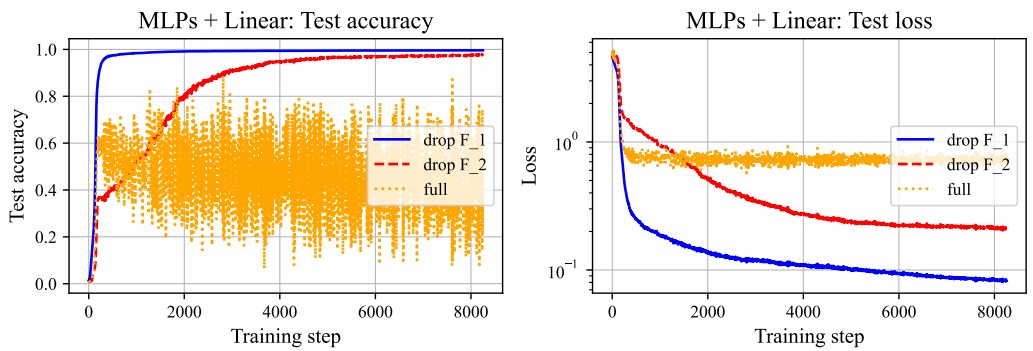

Figure 11: Test performance of fully dropping $F_1, F_2$ when both $F_1$ is MLPs and $F_2$ Linear. Both dropping methods turn out to help predict more correctly than the full model. Meanwhile, dropping the MLP $F_1$ is better with lower test loss.

### B.6   TRAINING DETAILS ABOUT EXPERIMENTS

All of the training is with SGD optimization with learning rate in $\{0.001, 0.03\}$. The batch size is 512. The dimension is 256. The context length is 256. All results in the experiments are stable for any learning rate between 0.001 and 0.03. Each run of experiments is on a single Nvidia Tesla V100 GPU. It takes 3 hours to finish each run for 2K steps, which probably can be optimized a lot since we are tracking a lot of measurement along training, not limited to hundreds of possible truncations at each test time.

## C   MORE EXPERIMENTS ON PYTHIA

### C.1   LEARNING ASSOCIATION WITH PREPOSITIONS

We would like to verify our guess about the structure of "to + the" in Pythia in Section 4.1. To make the argument generalizable than IOI dataset, we consider a structure of "[preposition] + the", where [preposition] has a pool of 30 prepositions in English, including "to". The input is a raw "[preposition]" or a random sentence ending with "[preposition]", with some examples in Appendix I.1. For both kinds of inputs, Pythia-160M/410M/1B turns out to learn the structure of "[preposition] + the" around 10 steps, as shown in Figure 14.

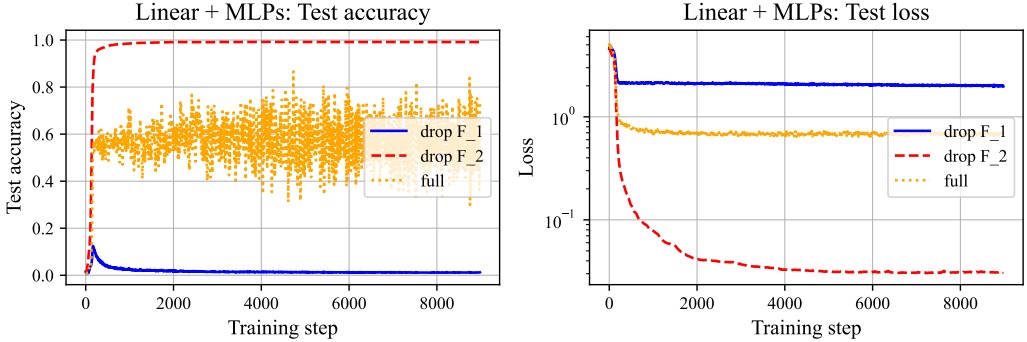

Figure 12: Test performance of fully dropping $F_1, F_2$ when both $F_1$ is Linear and $F_2$ MLPs. Only dropping $F_2$ helps predict more correctly. Dropping $F_1$ makes the model predicting noise more.

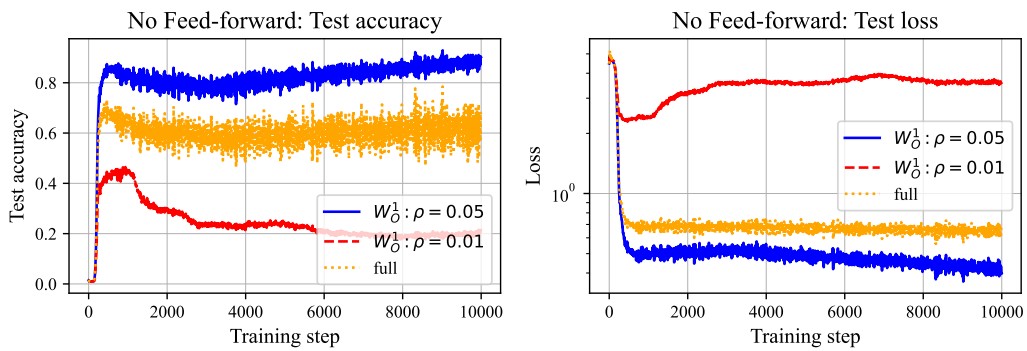

Figure 13: Test performance of low-rank truncating of $\mathbf{W}_O^1$ when there is no $F_1, F_2$. Here $\rho$ is the fraction of preserved rank of $\mathbf{W}_O^1$, where actually we re-parametrize the first-layer value matrix in attention as $\mathbf{W}_O^1 \mathbf{W}_V^1 \in \mathbb{R}^{d \times d}$. It turns out the best $\rho = 0.05$ improves the model's prediction a little. Meanwhile, a smaller $\rho$ destroys the model's performance.

## C.2 LASER PARAMETERS FOR EVALUATED LLMS

Following the definition of LASER in Section 2.2, we search for the optimal layer, $\rho$ and target weights in Pythia models and GPT-2 Small for each dataset.

**IOI on Pythia-410M.** The model has 24 layers. The truncation is on the input matrix of MLPs on the 22-th layer with $\rho = 0.02$.

**IOI on Pythia-1B.** The model has 16 layers. The truncation is on the input matrix of MLPs on the 11-th layer with $\rho = 0.008$.

**Factual recall on Pythia-1B.** The truncation is on the input matrix of MLPs on the 16-th layer with $\rho = 0.0125$.

**Factual recall on Pythia-1.4B.** The model has 24 layers. The truncation is on the input matrix of MLPs on the 24-th layer with $\rho = 0.025$.

**Factual recall on Pythia-2.8B.** The model has 32 layers. The truncation is on the input matrix of MLPs on the 32-th layer with $\rho = 0.04$.

**IOI on GPT2 Small.** Related parameters have been contained in Section 4.1.

**Phi-3 on GSM8K.** The model has 32 layers. The truncation is on the output matrix of MLPs on the 28-th layer with $\rho = 0.02$.

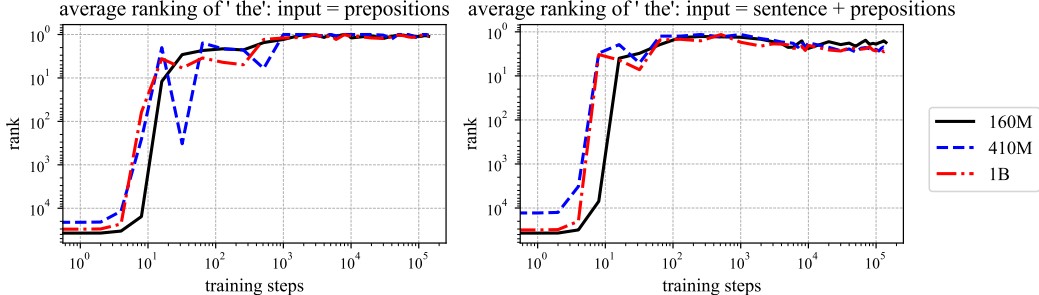

Figure 14: Average ranking of tokens "the" in the prediction by Pythia-160M/410M/1B along training. The inputs are 30 preposition words (left) and 40 sentences ending with prepositions. It turns out "the" becomes one of top predictions around 10 steps.

**Llama3.1-8B(-instruct) on GSM8K.** The models have 32 layers. The truncation is on the output of MLPs on the 27-th layer with $\rho = 0.02$.

### C.3 Other Pythia models on IOI and More Examples of Factual Recall

**IOI**. In the same setting of Figure 5 (left), we plot the prediction distributions of Pythia-410M and 1B on the 100 IOI inputs in Figure 15. The model checkpoints are the final ones after training. LASER turns out to decrease the probability of "the" while keeping that of the correct [IO] high.

**More examples of Factual Recall**. In additional to the factual query "Madrid is located in" in Figure 5 (right), we consider more such examples in Table 5. We plot the prediction distributions of Pythia-1B, 1.4B and 2.8B on these inputs in Figure 16, where LASER significantly lowers the probability of predicting "the" vesus the correct outputs.

## D Proof of Theorem 1

In this section, we will present the expectations and variances of $\nabla_{\mathbf{W}_V}\hat{L}$ and $\nabla_{\mathbf{W}_F}\hat{L}$ with $\mathbf{W}_V = \mathbf{W}_F = 0$ at initialization. The targets are to show:

1. a gap between $\lim_{m\to\infty}\nabla_{\mathbf{W}_V}\hat{L}$ and $\lim_{m\to\infty}\nabla_{\mathbf{W}_F}\hat{L}$ so that a step of GD with large learning rates is enough to learn the noise in $\mathbf{W}_F$, and

2. sample complexity of $\nabla_{\mathbf{W}_V}\hat{L}$ and $\nabla_{\mathbf{W}_F}\hat{L}$ based on expectations and variances.

**Assumption D.1** (Orthonormal embeddings). *The embeddings $u_k \in \mathbb{R}^d$ are assumed to be orthonormal, i.e., $u_i^\top u_j = \mathbb{1}\{i = j\}$. Meanwhile, if a matrix $\mathbf{W} \in \mathbb{R}^{d\times d}$ is random initialized, it holds $u_i^\top \mathbf{W} u_j = 0$.*

### D.1 Gradient for the Feed-forward Matrix $\mathbf{W}_F$

**Lemma D.1.** *Consider zero initialization, $\mathbf{W}_V = \mathbf{W}_F = \mathbf{W}_{QK} = 0$ and $N \gg 1$. Then with probability $1 - \delta$, for any $j, k \in [N + 1]$, it holds*

$$
\left| \mathbf{W}_U(k)^\top (\nabla_{\mathbf{W}_F}\hat{L})\mathbf{W}_E(q) - \mu(k) \right|
$$
$$
\leq \sqrt{\frac{4\sigma^2(k)\left(\ln(N+1) + \ln(\frac{2}{\delta})\right)}{m}} + \frac{4R(k)\left(\ln(N+1) + \ln(\frac{2}{\delta})\right)}{m}, \tag{8}
$$

*where $\mu(k), \sigma^2(k), R(k)$ are expectation, variance and range for different choices of $k \in [N]$ as follows:*

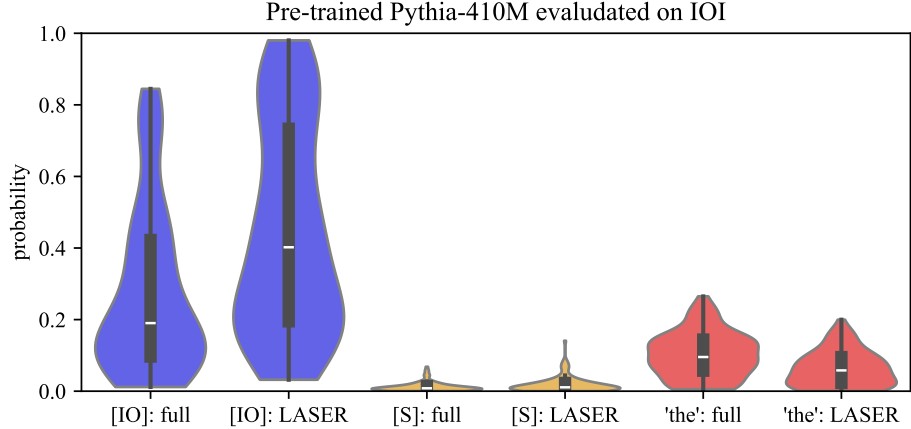

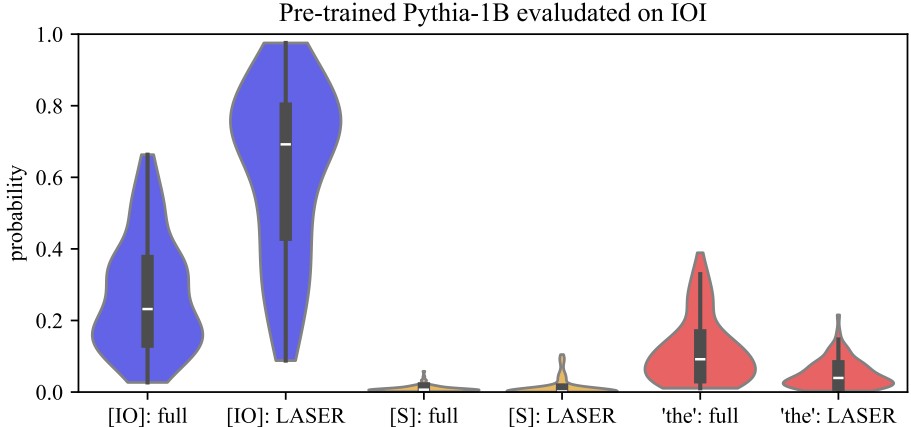

Figure 15: The prediction distributions of Pythia-410M and 1B on the IOI task. The setting is the same as in Fgure 5 (left). The evaluated models are the final checkpoints after training. LASER turns out to decrease the probability of "the" while keeping that of the correct [IO] high.

$$\forall\, k \le N: \quad \begin{array}{lll} \mu(N+1) = -\alpha, & \sigma^2(N+1) = \alpha(1-\alpha), & R(N+1) = \max\{\alpha, 1-\alpha\}, \\ \mu(k) = \frac{1}{N+1} - \frac{1-\alpha}{N}, & \sigma^2(k) = \frac{1-\alpha}{N}, & R(k) = 1. \end{array}$$

*Proof.* Due to zero initialization, *i.e.*, $\mathbf{W}_V = \mathbf{W}_F = 0$, the current predicted probability is $\hat{p}_{\mathbf{W}}(k|x_i) \equiv \frac{1}{N+1}$ for all $i \in [m]$ and $k \in [N+1]$. Therefore, from Lemma H.1, we have

$$\nabla_{\mathbf{W}_F}\hat{L} = \frac{1}{m}\sum_{i=1}^{m}\left[\sum_{k=1}^{N+1}\left(\frac{1}{N+1} - \mathbb{1}\{y_i = k\}\right)\mathbf{W}_U(k)x_{i,T}^\top\right],$$

where $x_{i,T} \in \mathbb{R}^d = \mathbf{W}_E(z_{i,T}) + p_T$ is the input embedding with input token $z_{i,T}$ at position $T$ in sequence $i$, together with positional encoding $p_T$ for position $T$. Since $z_{i,T}$ is set to be the trigger $q$ in the data generation process and $p_T$ is assumed to orthogonal to any other vector in $\mathbf{W}_E$ in Assumption D.1, we have the following projections for $\nabla_{\mathbf{W}_F}\hat{L}$: $\forall\, k \in [N+1]$,

$$\mathbf{W}_U(k)^\top(\nabla_{\mathbf{W}_F}\hat{L})\mathbf{W}_E(q) = \frac{1}{m}\sum_{i=1}^{m}\left(\frac{1}{N+1} - \mathbb{1}\{y_i = k\}\right).$$

From the data generation process, it is obvious to get

$$\mathbb{E}_{(x,y)}\left[\frac{1}{N+1} - \mathbb{1}\{y = k\}\right] = \frac{1}{N+1} - \alpha \cdot \mathbb{1}\{k = N+1\} - \frac{1-\alpha}{N} \cdot \mathbb{1}\{k \le N\}. \quad (9)$$

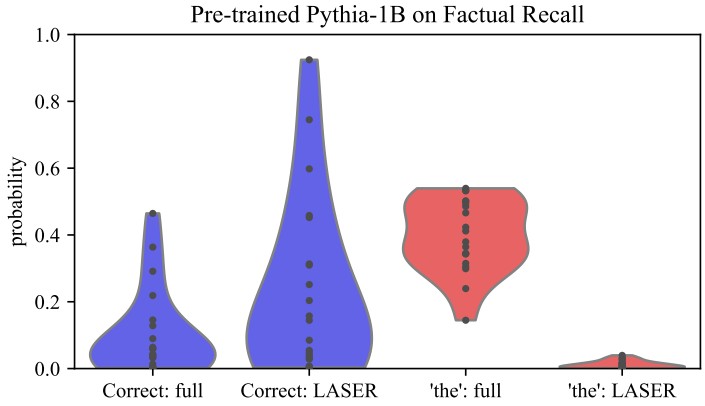

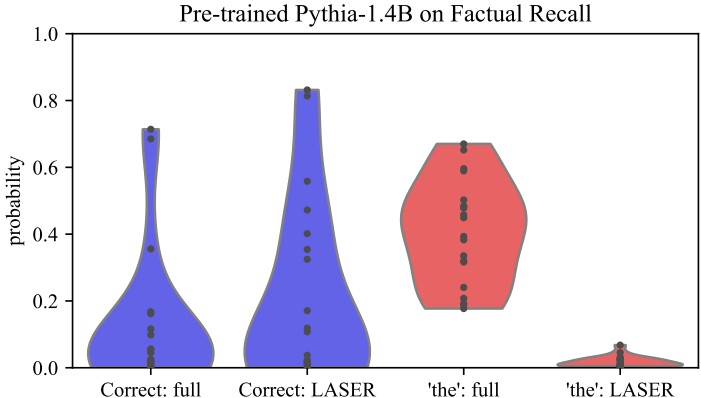

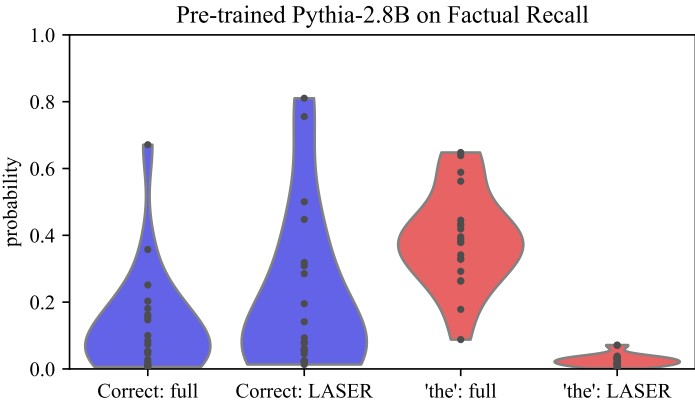

Figure 16: The prediction distributions of Pythia-1B, 1.4B and 2.8B on more examples of factual recall. Compared with the setting in Figure 5 (right), here we use 20 examples in Table 5. LASER turns out to significantly decrease the probability of "the" against the correct tokens.

Since $\alpha = \Theta(1)$ is much larger than $\frac{1}{N+1}$ when $N \gg 1$, due to law of large numbers, we have the population gradient $\nabla_{\mathbf{W}_F} L$ satisfying

$$\mathbf{W}_U(N+1)^\top (-\nabla_{\mathbf{W}_F} L) \mathbf{W}_E(q) \approx \alpha = \Theta(1),$$

$$\forall\, k \leq N: \quad \mathbf{W}_U(k)^\top (-\nabla_{\mathbf{W}_F} L) \mathbf{W}_E(q) < 0, \text{ with absolute value in } O(1/N).$$

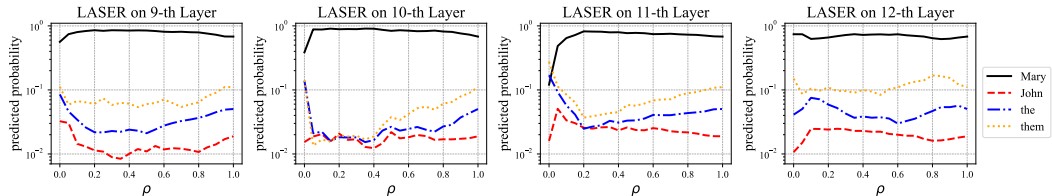

Figure 17: Predicted probability for $c \in \{\text{"Mary", "them", "the", "John"}\}$. LASER is conducted on input matrices of MLP layers on the layer $l = 9, 10, 11, 12$ of GPT-2 Small. The input is "When Mary and John went to a store, John gave a drink to". The horizontal is the fraction of perserved rank, $\rho \in [0, 1]$, where $\rho = 1$ stands for the full model. It turns out LASER clearly decreases probability of "the" and "them" when $\rho \in [0.1, 0.8]$ for layer $l = 9, 10, 11$, compared with the full model.

The variance of the gradient projection onto $\mathbf{W}_U(N+1)\mathbf{W}_E(q)^\top$ of a single data point follows that of Bernoulli distribution with parameter $\alpha$, which means

$$\text{Var}\left[\frac{1}{N+1} - \mathbb{1}\{y = N+1\}\right] = \alpha(1 - \alpha). \tag{10}$$

Similarly, for any $k \leq N$, the variance of the gradient projection onto $\mathbf{W}_U(N+1)\mathbf{W}_E(q)^\top$ of a single data point follows that of Bernoulli distribution with parameter $\frac{1-\alpha}{N}$, which means

$$\text{Var}\left[\frac{1}{N+1} - \mathbb{1}\{y = k\}\right] = \frac{1-\alpha}{N}\left(1 - \frac{1-\alpha}{N}\right) = \Theta(1/N). \tag{11}$$

The ranges of the gradient projections' deviation from the expectation are

$$\left|\frac{1}{N+1} - \mathbb{1}\{y = N+1\} - \left(\frac{1}{N+1} - \alpha\right)\right| \leq \max\{\alpha, 1-\alpha\},$$

$$\forall\, k \leq N: \quad \left|\frac{1}{N+1} - \mathbb{1}\{y = k\} - \left(\frac{1}{N+1} - \frac{1-\alpha}{N}\right)\right| \lesssim 1. \tag{12}$$

For each choice of $k \in [N+1]$ *individually*, after having the expectation $\mu(k)$, variance $\sigma^2(k)$ and range $R(k)$, by applying Bernstein's inequality, then: for each $k \in [N+1]$, with probability $1 - \delta$, it holds

$$\left|\mathbf{W}_U(k)^\top (\nabla_{\mathbf{w}_F}\hat{L})\mathbf{W}_E(q) - \mu(k)\right| \leq \sqrt{\frac{4\sigma^2(k)\ln(\frac{2}{\delta})}{m}} + \frac{4R(k)\ln(\frac{2}{\delta})}{m}.$$

Then by the union bound in probability, we need $(N+1)$ events above to hold at the same time, so we can substitute $\delta$ with $\frac{\delta}{N+1}$ to have: with probability $1 - \delta$, for any $k \in [N+1]$, it holds

$$\left|\mathbf{W}_U(k)^\top (\nabla_{\mathbf{w}_F}\hat{L})\mathbf{W}_E(q) - \mu(k)\right| \leq \sqrt{\frac{4\sigma^2(k)\left(\ln(N+1) + \ln(\frac{2}{\delta})\right)}{m}} + \frac{4R(k)\left(\ln(N+1) + \ln(\frac{2}{\delta})\right)}{m}. \tag{13}$$

$\square$

## D.2 GRADIENT FOR THE VALUE MATRIX $\mathbf{W}_V$

**Lemma D.2.** *Consider zero initialization, $\mathbf{W}_V = \mathbf{W}_F = \mathbf{W}_{QK} = 0$. Then with probability $1 - \delta$, for any $j, k \in [N+1]$, it holds*

$$\left|\mathbf{W}_U(j)^\top (\nabla_{\mathbf{w}_V}\hat{L})\mathbf{W}_E(k) - \mu(j, k)\right|$$

$$\leq \sqrt{\frac{4\sigma^2(j, k)\left(2\ln(N+1) + \ln(\frac{2}{\delta})\right)}{m}} + \frac{4R(j, k)\left(2\ln(N+1) + \ln(\frac{2}{\delta})\right)}{m}, \tag{14}$$

Table 3: $\mu(j,k), \sigma^2(j,k), R(j,k)$ for different choices of $(j,k)$ in Lemma D.2.

| $j$ | $k$ | $\mu$ | $\sigma^2$ | $R$ |
|---|---|---|---|---|
| $N+1$ | $N+1$ | $-\frac{\alpha^2}{N}$ | $\frac{\alpha^2}{TN} + \frac{\alpha^3-\alpha^4}{N^2}$ | $\frac{1}{2}$ |
| $N+1$ | $q$ | $-\frac{\alpha}{N}$ | $\frac{\alpha}{TN} + \frac{\alpha-\alpha^2}{N^2}$ | $1$ |
| $N+1$ | $[N]\setminus\{q\}$ | $-\frac{\alpha}{N}$ | $\frac{\alpha}{TN} + \frac{\alpha-\alpha^2}{N^2}$ | $1$ |
| $q$ | $N+1$ | $\frac{2\alpha-1}{N^2}$ | $\frac{1}{TN^2} + \frac{\alpha^2-\alpha+1}{N^3}$ | $\frac{1}{2}$ |
| $q$ | $q$ | $\frac{2\alpha-1}{\alpha N^2}$ | $\frac{\alpha^3-\alpha^2-\alpha+2}{\alpha^3 TN^2} + \frac{\alpha^2-\alpha+1}{\alpha^2 N^3}$ | $1$ |
| $q$ | $[N]\setminus\{q\}$ | $\frac{\alpha}{N^2}$ | $(2-\alpha)\cdot\left(\frac{1}{TN^2} + \frac{1}{N^3}\right)$ | $1$ |
| $[N]\setminus\{q\}$ | $N+1$ | $\frac{\alpha^2}{N^2}$ | $(2-\alpha)\left(\frac{\alpha}{TN^2} + \frac{\alpha^2}{N^3}\right)$ | $\frac{1}{3}$ |
| $[N]\setminus\{q\}$ | $q$ | $\frac{\alpha}{N^2}$ | $(2-\alpha)\left(\frac{1}{TN^2} + \frac{1}{N^3}\right)$ | $\frac{1}{2}$ |
| $[N]\setminus\{q\}$ | $j$ | $\frac{-\alpha^2+3\alpha-1}{N^2}$ | $\frac{1+(1-\alpha)(2-\alpha)}{TN^2} + \frac{1+(1-\alpha)(2-\alpha)^2}{N^3}$ | $1$ |
| $[N]\setminus\{q\}$ | $[N]\setminus\{q,j\}$ | $\frac{\alpha}{N^2}$ | $(2-\alpha)\left(\frac{1}{TN^2} + \frac{1}{N^3}\right)$ | $1$ |

*where $\mu(j,k), \sigma^2(j,k), R(j,k)$ are expectation, variance and range for different choices of $(j,k)$ at listed in Table 3.*

*Proof.* Due to zero initialization, *i.e.*, $\mathbf{W}_V = \mathbf{W}_F = 0$, the current predicted probability is $\hat{p}_\mathbf{W}(k|x_i) \equiv \frac{1}{N+1}$ for all $i \in [m]$ and $k \in [N+1]$. Meanwhile, the attention score is uniform as $\frac{1}{T}$ for all context positions due to $\mathbf{W}_K = 0$. Therefore, from Lemma H.1, we have

$$\nabla_{\mathbf{W}_F}\hat{L} = \frac{1}{m}\sum_{i=1}^m \left[\sum_{k=1}^{N+1}\left(\frac{1}{N+1} - \mathbb{1}\{y_i = k\}\right)\mathbf{W}_U(k)\left(\frac{1}{T}\sum_{t=1}^T x_{i,t}\right)^\top\right],$$

where $x_{i,t} \in \mathbb{R}^d = \mathbf{W}_E(z_{i,t}) + p_t$ is the input embedding with input token $z_{i,t}$ at position $t$ in sequence $i$, together with positional encoding $p_t$ for position $t$. With the assumption of orthonormality in Assumption D.1, we have the projection of $\nabla_{\mathbf{W}_F}\hat{L}$: $\forall\, j, k \in [N+1]$,

$$\mathbf{W}_U(j)^\top(\nabla_{\mathbf{W}_V}\hat{L})\mathbf{W}_E(k) = \frac{1}{m}\sum_{i=1}^m\left[\left(\frac{1}{N+1} - \mathbb{1}\{y_i = j\}\right)\left(\frac{1}{T}\sum_{t=1}^T \mathbb{1}\{z_{i,t} = k\}\right)\right].$$

Since each sample is drawn i.i.d., it suffices to discuss the expectation and variance of

$$\Gamma_i(j,k) \triangleq \left(\frac{1}{N+1} - \mathbb{1}\{z_{i,T+1} = j\}\right)\left(\frac{1}{T}\sum_{t=1}^T \mathbb{1}\{z_{i,t} = k\}\right),$$

$$\hat{\Gamma}(j,k) \triangleq \frac{1}{m}\sum_{i=1}^m \Gamma_i(j,k),$$

where we use the fact $y_i = z_{i,T+1}$.

Recall that, for each sample in the data generation process, the trigger $q$ is fixed while the correct next token $\bar{y} \sim \text{Uniform}([N])$. Hence, conditioning on $z_{i,T} = q$, it has probability $\alpha$ for $z_{i,T+1} = N+1$ and probability $1-\alpha$ for $z_{i,T+1} = \bar{y}$. This leads to the necessity of discussing whether or not $\bar{y} = k$. Meanwhile, a corner case of $\bar{y} = q$ is also necessary to consider, as this implies an event that increases the counting $\frac{1}{T}\sum_{t=1}^T \mathbb{1}\{z_{i,t} = q\}$ than the case of $\bar{y} \neq q$.

Therefore, generally there are 10 cases due to different choices of $(j,k)$ as follows:

1. $j = N+1, k = N+1$,

2. $j = N+1, k = q$,

3. $j = N+1, k \in [N]\setminus\{q\}$,

4. $j = q, k = N + 1$,

5. $j = q, k = q$,

6. $j = q, k \in [N] \setminus \{q\}$,

7. $j \in [N] \setminus \{q\}, k = N + 1$,

8. $j \in [N] \setminus \{q\}, k = q$,

9. $j \in [N] \setminus \{q\}, k = j$,

10. $j \in [N] \setminus \{q\}, k \in [N] \setminus \{q, j\}$.

For each $\Gamma_i(j, k)$ *individually*, if we have its expectation $\mu(j, k)$, variance $\sigma^2(j, k)$ and range $R(j, k)$, by applying Bernstein's inequality, then: for each $j, k \in [N + 1]$, with probability $1 - \delta$, it holds

$$\left| \hat{\Gamma}(j, k) - \mu(j, k) \right| \leq \sqrt{\frac{4\sigma^2(j, k) \ln(\frac{2}{\delta})}{m}} + \frac{4R(j, k) \ln(\frac{2}{\delta})}{m}.$$

Then by the union bound in probability, we need $(N + 1)^2$ events above to hold at the same time, so we can substitute $\delta$ with $\frac{\delta}{(N+1)^2}$ to have: with probability $1 - \delta$, for any $j, k \in [N + 1]$, it holds

$$\left| \hat{\Gamma}(j, k) - \mu(j, k) \right| \leq \sqrt{\frac{4\sigma^2(j, k) \left( 2 \ln(N + 1) + \ln(\frac{2}{\delta}) \right)}{m}} + \frac{4R(j, k) \left( 2 \ln(N + 1) + \ln(\frac{2}{\delta}) \right)}{m}.$$

(15)

As a final step of the proof, now we elaborate the expectation, variance and range of $\Gamma_i(j, k)$ for these 10 cases.

**Case 1:** $j = N + 1, k = N + 1$.

There is probability $\frac{1}{N}$ for $\bar{y} = q$ and probability $\frac{N-1}{N}$ for $\bar{y} \neq q$. Hence, we have

$$\mathbb{E}[\Gamma_i(j, k)] = \frac{1}{N} \mathbb{E}[\Gamma_i(j, k) | \bar{y} = q] + \frac{N - 1}{N} \mathbb{E}[\Gamma_i(j, k) | \bar{y} \neq q],$$
$$\mathbb{E}[\Gamma_i(j, k)^2] = \frac{1}{N} \mathbb{E}[\Gamma_i(j, k)^2 | \bar{y} = q] + \frac{N - 1}{N} \mathbb{E}[\Gamma_i(j, k)^2 | \bar{y} \neq q].$$

From Lemma E.2 and the independence between $\mathbb{1}\{z_{i,T+1} = N + 1\}$ and $\sum_{t \leq T} \mathbb{1}\{z_{i,t} = k\}$, we have

$$\mathbb{E}[\Gamma_i(j, k) | \bar{y} = q] \approx -\alpha \cdot \frac{1}{N},$$
$$\mathbb{E}[\Gamma_i(j, k)^2 | \bar{y} = q] \approx \alpha \cdot \left( \frac{1}{TN} + \frac{1}{N^2} \right),$$

where the second is from

$$\mathbb{E}\left[ \left( \frac{1}{N + 1} - \mathbb{1}\{z_{i,T+1} = N + 1\} \right)^2 \right] = (1 - \alpha) \cdot \left( \frac{1}{N + 1} \right)^2 + \alpha \cdot \left( \frac{1}{N + 1} - 1 \right)^2 \approx \alpha.$$

Similarly, from Lemma E.5, we have

$$\mathbb{E}[\Gamma_i(j, k) | \bar{y} \neq q] \approx -\alpha \cdot \frac{\alpha}{N},$$
$$\mathbb{E}[\Gamma_i(j, k)^2 | \bar{y} \neq q] \approx \alpha \cdot \left( \frac{\alpha}{TN} + \frac{\alpha^2}{N^2} \right).$$

Therefore, it holds

$$\mathbb{E}[\Gamma_i(j,k)] = \frac{1}{N}\frac{-\alpha}{N} + \frac{N-1}{N}\frac{-\alpha^2}{N} \approx -\frac{\alpha^2}{N},$$

$$\mathbb{E}[\Gamma_i(j,k)^2] = \frac{1}{N}\mathbb{E}[\Gamma_i(j,k)^2|\bar{y}=q] + \frac{N-1}{N}\mathbb{E}[\Gamma_i(j,k)^2|\bar{y}\neq q]$$

$$\approx \frac{1}{N}\alpha\cdot\left(\frac{1}{TN} + \frac{1}{N^2}\right) + \frac{N-1}{N}\alpha\cdot\left(\frac{\alpha}{TN} + \frac{\alpha^2}{N^2}\right) \approx \frac{\alpha^2}{TN} + \frac{\alpha^3}{N^2},$$

$$\mathrm{Var}[\Gamma_i(j,k)] = \mathbb{E}[\Gamma_i(j,k)^2] - \mathbb{E}[\Gamma_i(j,k)]^2 \approx \frac{\alpha^2}{TN} + \frac{\alpha^3-\alpha^4}{N^2}.$$

The range of $\Gamma_i(j,k)$ is

$$|\Gamma_i(j,k) - \mathbb{E}[\Gamma_i(j,k)]| \leq \frac{1}{2},$$

and the extreme case is when half of the sequence is $N+1$ with the rest all being $q$.

**Case 2:** $j = N+1, k = q$.

Similar to Case 1, we have $\mathbb{1}\{z_{i,T+1} = N+1\}$ is independent of $\sum_{t\leq T}\mathbb{1}\{z_{i,t} = k\}$.

From Lemma E.1, we have

$$\mathbb{E}[\Gamma_i(j,k)|\bar{y}=q] \approx -\alpha\cdot\frac{1}{\alpha N},$$

$$\mathbb{E}[\Gamma_i(j,k)^2|\bar{y}=q] \approx \alpha\cdot\left(\frac{1}{\alpha TN}\left(-1 + \frac{2}{\alpha^2}\right) + \frac{1}{\alpha^2 N^2}\right).$$

From Lemma E.4, we have

$$\mathbb{E}[\Gamma_i(j,k)|\bar{y}\neq q] \approx -\alpha\cdot\frac{1}{N},$$

$$\mathbb{E}[\Gamma_i(j,k)^2|\bar{y}\neq q] \approx \alpha\cdot\left(\frac{1}{TN} + \frac{1}{N^2}\right).$$

Therefore, we have

$$\mathbb{E}[\Gamma_i(j,k)] = \frac{1}{N}\mathbb{E}[\Gamma_i(j,k)|\bar{y}=q] + \frac{N-1}{N}\mathbb{E}[\Gamma_i(j,k)|\bar{y}\neq q] \approx -\frac{\alpha}{N},$$

$$\mathbb{E}[\Gamma_i(j,k)^2] = \frac{1}{N}\mathbb{E}[\Gamma_i(j,k)^2|\bar{y}=q] + \frac{N-1}{N}\mathbb{E}[\Gamma_i(j,k)^2|\bar{y}\neq q] \approx \frac{\alpha}{TN} + \frac{\alpha}{N^2},$$

$$\mathrm{Var}[\Gamma_i(j,k)] = \mathbb{E}[\Gamma_i(j,k)^2] - \mathbb{E}[\Gamma_i(j,k)]^2 \approx \frac{\alpha}{TN} + \frac{\alpha-\alpha^2}{N^2}.$$

The range of $\Gamma_i(j,k)$ is

$$|\Gamma_i(j,k) - \mathbb{E}[\Gamma_i(j,k)]| \lessapprox 1,$$

and the extreme case is when $\bar{y} = q$ and the sequence is all $q$'s.

**Case 3:** $j = N+1, k \in [N]\setminus\{q\}$.

Similar to Case 1, we have $\mathbb{1}\{z_{i,T+1} = N+1\}$ is independent of $\sum_{t\leq T}\mathbb{1}\{z_{i,t} = k\}$.

From Lemma E.3, we have

$$\mathbb{E}[\Gamma_i(j,k)|\bar{y}=q] \approx -\alpha\cdot\frac{1}{N},$$

$$\mathbb{E}[\Gamma_i(j,k)^2|\bar{y}=q] \approx \alpha\cdot\left(\frac{1}{TN} + \frac{1}{N^2}\right).$$

From Lemma E.7, we have

$$\mathbb{E}[\Gamma_i(j,k)|\bar{y} \neq q] \approx -\alpha \cdot \frac{1}{N},$$
$$\mathbb{E}[\Gamma_i(j,k)^2|\bar{y} \neq q] \approx \alpha \cdot \left(\frac{1}{TN} + \frac{1}{N^2}\right).$$

Therefore, we have

$$\mathbb{E}[\Gamma_i(j,k)] \approx -\alpha \cdot \frac{1}{N},$$
$$\mathbb{E}[\Gamma_i(j,k)^2] \approx \alpha \cdot \left(\frac{1}{TN} + \frac{1}{N^2}\right),$$
$$\mathrm{Var}[\Gamma_i(j,k)] = \mathbb{E}[\Gamma_i(j,k)^2] - \mathbb{E}[\Gamma_i(j,k)]^2 \approx \frac{\alpha}{TN} + \frac{\alpha - \alpha^2}{N^2}.$$

The range of $\Gamma_i(j,k)$ is

$$|\Gamma_i(j,k) - \mathbb{E}[\Gamma_i(j,k)]| \lesssim 1,$$

and the extreme case is when all of the sequence except the last one is $k$.

**Case 4:** $j = q, k = N + 1$.

If $\bar{y} \neq q$, we always have $z_{i,T+1} \neq q$ because $z_{i,T+1} \in \{\bar{y}, N+1\}$. If conditioning on $\bar{y} = q$, it has probability $1 - \alpha$ for $z_{i,T+1} = q$, independent of $\sum_{t \leq T} \mathbb{1}\{z_{i,t} = N+1\}$.

From Lemma E.5, we have

$$\mathbb{E}[\Gamma_i(j,k)|\bar{y} \neq q] \approx \frac{1}{N+1} \cdot \frac{\alpha}{N},$$
$$\mathbb{E}[\Gamma_i(j,k)^2|\bar{y} \neq q] \approx \frac{1}{N+1} \cdot \left(\frac{\alpha}{TN} + \frac{\alpha^2}{N^2}\right).$$

From Lemma E.2, we have

$$\mathbb{E}[\Gamma_i(j,k)|\bar{y} = q] \approx -(1 - \alpha) \cdot \frac{1}{N},$$
$$\mathbb{E}[\Gamma_i(j,k)^2|\bar{y} = q] \approx (1 - \alpha) \cdot \left(\frac{1}{TN} + \frac{1}{N^2}\right).$$

Therefore, we have

$$\mathbb{E}[\Gamma_i(j,k)] = \frac{1}{N}\mathbb{E}[\Gamma_i(j,k)|\bar{y} = q] + \frac{N-1}{N}\mathbb{E}[\Gamma_i(j,k)|\bar{y} \neq q] \approx \frac{2\alpha - 1}{N^2},$$
$$\mathbb{E}[\Gamma_i(j,k)^2] = \frac{1}{N}\mathbb{E}[\Gamma_i(j,k)^2|\bar{y} = q] + \frac{N-1}{N}\mathbb{E}[\Gamma_i(j,k)^2|\bar{y} \neq q] \approx \frac{1}{TN^2} + \frac{\alpha^2 - \alpha + 1}{N^3},$$
$$\mathrm{Var}[\Gamma_i(j,k)] = \mathbb{E}[\Gamma_i(j,k)^2] - \mathbb{E}[\Gamma_i(j,k)]^2 \approx \frac{1}{TN^2} + \frac{\alpha^2 - \alpha + 1}{N^3}.$$

The range of $\Gamma_i(j,k)$ is

$$|\Gamma_i(j,k) - \mathbb{E}[\Gamma_i(j,k)]| \lesssim \frac{1}{2},$$

and the extreme case is when $\bar{y} = q$ and half of the sequence is $N + 1$ with the rest all being $q$.

**Case 5:** $j = q, k = q$.

Similar to Case 4, if $\bar{y} \neq q$, we always have $z_{i,T+1} \neq q$. If conditioning on $\bar{y} = q$, it has probability $1 - \alpha$ for $z_{i,T+1} = q$, independent of $\sum_{t \leq T} \mathbb{1}\{z_{i,t} = q\}$.

From Lemma E.4, we have

$$\mathbb{E}[\Gamma_i(j,k)|\bar{y} \neq q] \approx \frac{1}{N+1} \cdot \frac{1}{N},$$

$$\mathbb{E}[\Gamma_i(j,k)^2|\bar{y} \neq q] \approx \frac{1}{N+1} \cdot \left( \frac{1}{TN} + \frac{1}{N^2} \right).$$

From Lemma E.1, we have

$$\mathbb{E}[\Gamma_i(j,k)|\bar{y} = q] \approx -(1-\alpha) \cdot \frac{1}{\alpha N},$$

$$\mathbb{E}[\Gamma_i(j,k)^2|\bar{y} = q] \approx (1-\alpha) \cdot \left( \frac{1}{\alpha TN} \left( -1 + \frac{2}{\alpha^2} \right) + \frac{1}{\alpha^2 N^2} \right).$$

Therefore, we have

$$\mathbb{E}[\Gamma_i(j,k)] = \frac{1}{N}\mathbb{E}[\Gamma_i(j,k)|\bar{y}=q] + \frac{N-1}{N}\mathbb{E}[\Gamma_i(j,k)|\bar{y} \neq q] \approx \frac{2\alpha - 1}{\alpha N^2},$$

$$\mathbb{E}[\Gamma_i(j,k)^2] = \frac{1}{N}\mathbb{E}[\Gamma_i(j,k)^2|\bar{y}=q] + \frac{N-1}{N}\mathbb{E}[\Gamma_i(j,k)^2|\bar{y} \neq q]$$

$$\approx \frac{\alpha^3 - \alpha^2 - \alpha + 2}{\alpha^3 TN^2} + \frac{\alpha^2 - \alpha + 1}{\alpha^2 N^3},$$

$$\mathrm{Var}[\Gamma_i(j,k)] = \mathbb{E}[\Gamma_i(j,k)^2] - \mathbb{E}[\Gamma_i(j,k)]^2 \approx \frac{\alpha^3 - \alpha^2 - \alpha + 2}{\alpha^3 TN^2} + \frac{\alpha^2 - \alpha + 1}{\alpha^2 N^3}.$$

The range of $\Gamma_i(j,k)$ is

$$|\Gamma_i(j,k) - \mathbb{E}[\Gamma_i(j,k)]| \lessgtr 1,$$

and the extreme case is when $\bar{y} = q$ and all of the sequence are $q$.

**Case 6:** $j = q, k \in [N] \setminus \{q\}$.

Similar to Case 4, if $\bar{y} \neq q$, we always have $z_{i,T+1} \neq q$. If conditioning on $\bar{y} = q$, it has probability $1 - \alpha$ for $z_{i,T+1} = q$, independent of $\sum_{t \leq T} \mathbb{1}\{z_{i,t} = k\}$.

Moreover, we need to consider whether $\bar{y} = k$ or not.

From Lemma E.6, we have

$$\mathbb{E}[\Gamma_i(j,k)|\bar{y} \neq q, k = \bar{y}] \approx \frac{1}{N+1} \cdot \frac{2-\alpha}{N},$$

$$\mathbb{E}[\Gamma_i(j,k)^2|\bar{y} \neq q, k = \bar{y}] \approx \frac{1}{N+1} \cdot \left( \frac{2-\alpha}{TN} + \frac{(2-\alpha)^2}{N^2} \right).$$

From Lemma E.7, we have

$$\mathbb{E}[\Gamma_i(j,k)|\bar{y} \neq q, k \in [N] \setminus \{q, \bar{y}\}] \approx \frac{1}{N+1} \cdot \frac{1}{N},$$

$$\mathbb{E}[\Gamma_i(j,k)^2|\bar{y} \neq q, k \in [N] \setminus \{q, \bar{y}\}] \approx \frac{1}{N+1} \cdot \left( \frac{1}{TN} + \frac{1}{N^2} \right).$$

From Lemma E.3, we have

$$\mathbb{E}[\Gamma_i(j,k)|\bar{y} = q] \approx -(1-\alpha) \cdot \frac{1}{N},$$

$$\mathbb{E}[\Gamma_i(j,k)^2|\bar{y} = q] \approx (1-\alpha) \cdot \left( \frac{1}{TN} + \frac{1}{N^2} \right).$$

Therefore, we have

$$\mathbb{E}[\Gamma_i(j,k)] = \frac{1}{N}\mathbb{E}[\Gamma_i(j,k)|\bar{y} = q] + \frac{1}{N}\mathbb{E}[\Gamma_i(j,k)|\bar{y} \neq q, k = \bar{y}]$$
$$+ \frac{N-2}{N}\mathbb{E}[\Gamma_i(j,k)|\bar{y} \neq q, k \in [N] \setminus \{q, \bar{y}\}]$$
$$\approx \frac{\alpha}{N^2},$$

$$\mathbb{E}[\Gamma_i(j,k)^2] = \frac{1}{N}\mathbb{E}[\Gamma_i(j,k)^2|\bar{y} = q] + \frac{1}{N}\mathbb{E}[\Gamma_i(j,k)^2|\bar{y} \neq q, k = \bar{y}]$$
$$+ \frac{N-2}{N}\mathbb{E}[\Gamma_i(j,k)^2|\bar{y} \neq q, k \in [N] \setminus \{q, \bar{y}\}]$$
$$\approx (2-\alpha) \cdot \left(\frac{1}{TN^2} + \frac{1}{N^3}\right),$$

$$\mathrm{Var}[\Gamma_i(j,k)] = \mathbb{E}[\Gamma_i(j,k)^2] - \mathbb{E}[\Gamma_i(j,k)]^2 \approx (2-\alpha) \cdot \left(\frac{1}{TN^2} + \frac{1}{N^3}\right).$$

The range of $\Gamma_i(j,k)$ is

$$|\Gamma_i(j,k) - \mathbb{E}[\Gamma_i(j,k)]| \lesssim 1,$$

and the extreme case is when all of the sequence except the last one are $k$.

**Case 7:** $j \in [N] \setminus \{q\}, k = N + 1$.

If $\bar{y} \neq j$, we always have $z_{i,T+1} \neq j$ because $z_{i,T+1} \in \{\bar{y}, N+1\}$. If conditioning on $\bar{y} = j$, it has probability $1 - \alpha$ for $z_{i,T+1} = j$, independent of $\sum_{t \leq T} \mathbb{1}\{z_{i,t} = N+1\}$.

Moreover, in the case of $\bar{y} \neq j$, we need to discuss whether or not $\bar{y} = q$.

From Lemma E.2, we have

$$\mathbb{E}[\Gamma_i(j,k)|\bar{y} = q] \approx \frac{1}{N+1} \cdot \frac{1}{N},$$
$$\mathbb{E}[\Gamma_i(j,k)^2|\bar{y} = q] \approx \frac{1}{N+1} \cdot \left(\frac{1}{TN} + \frac{1}{N^2}\right).$$

From Lemma E.5, we have

$$\mathbb{E}[\Gamma_i(j,k)|\bar{y} \neq q, \bar{y} \neq j] \approx \frac{1}{N+1} \cdot \frac{\alpha}{N},$$
$$\mathbb{E}[\Gamma_i(j,k)^2|\bar{y} \neq q, \bar{y} \neq j] \approx \frac{1}{N+1} \cdot \left(\frac{\alpha}{TN} + \frac{\alpha^2}{N^2}\right).$$

From Lemma E.5, we have

$$\mathbb{E}[\Gamma_i(j,k)|\bar{y} = j] \approx -(1-\alpha) \cdot \frac{\alpha}{N},$$
$$\mathbb{E}[\Gamma_i(j,k)^2|\bar{y} = j] \approx (1-\alpha) \cdot \left(\frac{\alpha}{TN} + \frac{\alpha^2}{N^2}\right).$$

Therefore, we have

$$\mathbb{E}[\Gamma_i(j,k)] = \frac{1}{N}\mathbb{E}[\Gamma_i(j,k)|\bar{y} = q] + \frac{1}{N}\mathbb{E}[\Gamma_i(j,k)|\bar{y} = j] + \frac{N-2}{N}\mathbb{E}[\Gamma_i(j,k)|y \neq q, \bar{y} \neq j]$$
$$\approx \frac{\alpha^2}{N^2},$$

$$\mathbb{E}[\Gamma_i(j,k)^2] = \frac{1}{N}\mathbb{E}[\Gamma_i(j,k)^2|\bar{y} = q] + \frac{1}{N}\mathbb{E}[\Gamma_i(j,k)^2|\bar{y} = j] + \frac{N-2}{N}\mathbb{E}[\Gamma_i(j,k)^2|y \neq q, \bar{y} \neq j]$$
$$\approx (2-\alpha)\left(\frac{\alpha}{TN^2} + \frac{\alpha^2}{N^3}\right),$$

$$\mathrm{Var}[\Gamma_i(j,k)] = \mathbb{E}[\Gamma_i(j,k)^2] - \mathbb{E}[\Gamma_i(j,k)]^2 \approx (2-\alpha)\left(\frac{\alpha}{TN^2} + \frac{\alpha^2}{N^3}\right).$$

The range of $\Gamma_i(j, k)$ is

$$|\Gamma_i(j, k) - \mathbb{E}[\Gamma_i(j, k)]| \lesssim \frac{1}{3},$$

and the extreme case is when $\bar{y} = j$ and one-third of the sequence are $k$, where the sequence has a repeated pattern like $[q, j, N + 1, q, j, N + 1, \dots]$.

**Case 8:** $j \in [N] \setminus \{q\}, k = q$.

Similar to Case 7, if $\bar{y} \neq j$, we always have $z_{i,T+1} \neq j$. If conditioning on $\bar{y} = j$, it has probability $1 - \alpha$ for $z_{i,T+1} = j$, independent of $\sum_{t \leq T} \mathbb{1}\{z_{i,t} = N + 1\}$.

Moreover, in the case of $\bar{y} \neq j$, we need to discuss whether or not $\bar{y} = q$.

From Lemma E.1, we have

$$\mathbb{E}[\Gamma_i(j, k)|\bar{y} = q] \approx \frac{1}{N + 1} \cdot \frac{1}{\alpha N},$$

$$\mathbb{E}[\Gamma_i(j, k)^2|\bar{y} = q] \approx \frac{1}{N + 1} \cdot \left( \frac{T}{\alpha N} \left( -1 + \frac{2}{\alpha^2} \right) + \frac{T^2}{\alpha^2 N^2} \right).$$

From Lemma E.4, we have

$$\mathbb{E}[\Gamma_i(j, k)|\bar{y} \neq q, \bar{y} \neq j] \approx \frac{1}{N + 1} \cdot \frac{1}{N},$$

$$\mathbb{E}[\Gamma_i(j, k)^2|\bar{y} \neq q, \bar{y} \neq j] \approx \frac{1}{N + 1} \cdot \left( \frac{1}{TN} + \frac{1}{N^2} \right).$$

From Lemma E.4, we have

$$\mathbb{E}[\Gamma_i(j, k)|\bar{y} = j] \approx -(1 - \alpha) \cdot \frac{1}{N},$$

$$\mathbb{E}[\Gamma_i(j, k)^2|\bar{y} = j] \approx (1 - \alpha) \cdot \left( \frac{1}{TN} + \frac{1}{N^2} \right).$$

Therefore, we have

$$\mathbb{E}[\Gamma_i(j, k)] = \frac{1}{N}\mathbb{E}[\Gamma_i(j, k)|\bar{y} = q] + \frac{1}{N}\mathbb{E}[\Gamma_i(j, k)|\bar{y} = j] + \frac{N - 2}{N}\mathbb{E}[\Gamma_i(j, k)|y \neq q, \bar{y} \neq j]$$

$$\approx \frac{\alpha}{N^2},$$

$$\mathbb{E}[\Gamma_i(j, k)^2] = \frac{1}{N}\mathbb{E}[\Gamma_i(j, k)^2|\bar{y} = q] + \frac{1}{N}\mathbb{E}[\Gamma_i(j, k)^2|\bar{y} = j] + \frac{N - 2}{N}\mathbb{E}[\Gamma_i(j, k)^2|y \neq q, \bar{y} \neq j]$$

$$\approx (2 - \alpha) \left( \frac{1}{TN^2} + \frac{1}{N^3} \right),$$

$$\mathrm{Var}[\Gamma_i(j, k)] = \mathbb{E}[\Gamma_i(j, k)^2] - \mathbb{E}[\Gamma_i(j, k)]^2 \approx (2 - \alpha) \left( \frac{1}{TN^2} + \frac{1}{N^3} \right).$$

The range of $\Gamma_i(j, k)$ is

$$|\Gamma_i(j, k) - \mathbb{E}[\Gamma_i(j, k)]| \lesssim \frac{1}{2},$$

and the extreme case is when $\bar{y} = j$ and half of the sequence are $q$.

**Case 9:** $j \in [N] \setminus \{q\}, k = j$.

Similar to Case 7, if $\bar{y} \neq j$, we always have $z_{i,T+1} \neq j$. If conditioning on $\bar{y} = j$, it has probability $1 - \alpha$ for $z_{i,T+1} = j$, independent of $\sum_{t \leq T} \mathbb{1}\{z_{i,t} = N + 1\}$.

Moreover, in the case of $\bar{y} \neq j$, we need to discuss whether or not $\bar{y} = q$.

From Lemma E.3, we have

$$\mathbb{E}[\Gamma_i(j,k)|\bar{y}=q] \approx \frac{1}{N+1} \cdot \frac{1}{N},$$

$$\mathbb{E}[\Gamma_i(j,k)^2|\bar{y}=q] \approx \frac{1}{N+1} \cdot \left(\frac{1}{TN} + \frac{1}{N^2}\right).$$

From Lemma E.7, we have

$$\mathbb{E}[\Gamma_i(j,k)|\bar{y}\neq q, \bar{y}\neq j] \approx \frac{1}{N+1} \cdot \frac{1}{N},$$

$$\mathbb{E}[\Gamma_i(j,k)^2|\bar{y}\neq q, \bar{y}\neq j] \approx \frac{1}{N+1} \cdot \left(\frac{1}{TN} + \frac{1}{N^2}\right).$$

From Lemma E.6, we have

$$\mathbb{E}[\Gamma_i(j,k)|\bar{y}=j] \approx -(1-\alpha) \cdot \frac{2-\alpha}{N},$$

$$\mathbb{E}[\Gamma_i(j,k)^2|\bar{y}=j] \approx (1-\alpha) \cdot \left(\frac{2-\alpha}{TN} + \frac{(2-\alpha)^2}{N^2}\right).$$

Therefore, we have

$$\mathbb{E}[\Gamma_i(j,k)] = \frac{1}{N}\mathbb{E}[\Gamma_i(j,k)|\bar{y}=q] + \frac{1}{N}\mathbb{E}[\Gamma_i(j,k)|\bar{y}=j] + \frac{N-2}{N}\mathbb{E}[\Gamma_i(j,k)|y\neq q, \bar{y}\neq j]$$

$$\approx \frac{-\alpha^2 + 3\alpha - 1}{N^2},$$

$$\mathbb{E}[\Gamma_i(j,k)^2] = \frac{1}{N}\mathbb{E}[\Gamma_i(j,k)^2|\bar{y}=q] + \frac{1}{N}\mathbb{E}[\Gamma_i(j,k)^2|\bar{y}=j] + \frac{N-2}{N}\mathbb{E}[\Gamma_i(j,k)^2|y\neq q, \bar{y}\neq j]$$

$$\approx \frac{1+(1-\alpha)(2-\alpha)}{TN^2} + \frac{1+(1-\alpha)(2-\alpha)^2}{N^3},$$

$$\mathrm{Var}[\Gamma_i(j,k)] = \mathbb{E}[\Gamma_i(j,k)^2] - \mathbb{E}[\Gamma_i(j,k)]^2 \approx \frac{1+(1-\alpha)(2-\alpha)}{TN^2} + \frac{1+(1-\alpha)(2-\alpha)^2}{N^3}.$$

The range of $\Gamma_i(j,k)$ is

$$|\Gamma_i(j,k) - \mathbb{E}[\Gamma_i(j,k)]| \lesssim 1,$$

and the extreme case is when $\bar{y}=j$ and all of the sequence are $j=k$.

**Case 10:** $j \in [N] \setminus \{q\}, k \in [N] \setminus \{q,j\}$.

Similar to Case 7, if $\bar{y}\neq j$, we always have $z_{i,T+1}\neq j$. If conditioning on $\bar{y}=j$, it has probability $1-\alpha$ for $z_{i,T+1}=j$, independent of $\sum_{t\leq T}\mathbb{1}\{z_{i,t}=N+1\}$.

Moreover, in the case of $\bar{y}\neq j$, we need to discuss whether or not $\bar{y}=q$.

From Lemma E.3, we have

$$\mathbb{E}[\Gamma_i(j,k)|\bar{y}=q] \approx \frac{1}{N+1} \cdot \frac{1}{N},$$

$$\mathbb{E}[\Gamma_i(j,k)^2|\bar{y}=q] \approx \frac{1}{N+1} \cdot \left(\frac{1}{TN} + \frac{1}{N^2}\right).$$

From Lemma E.7, we have

$$\mathbb{E}[\Gamma_i(j,k)|\bar{y}=j] \approx -(1-\alpha) \cdot \frac{1}{N},$$

$$\mathbb{E}[\Gamma_i(j,k)^2|\bar{y}=j] \approx (1-\alpha) \cdot \left(\frac{1}{TN} + \frac{1}{N^2}\right).$$

From Lemma E.6, we have

$$\mathbb{E}[\Gamma_i(j,k)|\bar{y}=k] \approx \frac{1}{N+1} \cdot \frac{2-\alpha}{N},$$

$$\mathbb{E}[\Gamma_i(j,k)^2|\bar{y}=k] \approx \frac{1}{N+1} \cdot \left(\frac{2-\alpha}{TN} + \frac{(2-\alpha)^2}{N^2}\right).$$

From Lemma E.7, we have

$$\mathbb{E}[\Gamma_i(j,k)|\bar{y} \neq q, \bar{y} \neq j, \bar{y} \neq k] \approx \frac{1}{N+1} \cdot \frac{1}{N},$$

$$\mathbb{E}[\Gamma_i(j,k)^2|\bar{y} \neq q, \bar{y} \neq j, \bar{y} \neq k] \approx \frac{1}{N+1} \cdot \left(\frac{1}{TN} + \frac{1}{N^2}\right).$$

Therefore, we have

$$\mathbb{E}[\Gamma_i(j,k)] = \frac{1}{N}\mathbb{E}[\Gamma_i(j,k)|\bar{y}=q] + \frac{1}{N}\mathbb{E}[\Gamma_i(j,k)|\bar{y}=j] + \frac{1}{N}\mathbb{E}[\Gamma_i(j,k)|\bar{y}=k]$$

$$+ \frac{N-3}{N}\mathbb{E}[\Gamma_i(j,k)|y \neq q, \bar{y} \neq j]$$

$$\approx \frac{\alpha}{N^2},$$

$$\mathbb{E}[\Gamma_i(j,k)^2] = \frac{1}{N}\mathbb{E}[\Gamma_i(j,k)^2|\bar{y}=q] + \frac{1}{N}\mathbb{E}[\Gamma_i(j,k)^2|\bar{y}=j] + \frac{1}{N}\mathbb{E}[\Gamma_i(j,k)^2|\bar{y}=k]$$

$$+ \frac{N-3}{N}\mathbb{E}[\Gamma_i(j,k)^2|y \neq q, \bar{y} \neq j]$$

$$\approx (2-\alpha)\left(\frac{1}{TN} + \frac{1}{N^2}\right),$$

$$\mathrm{Var}[\Gamma_i(j,k)] = \mathbb{E}[\Gamma_i(j,k)^2] - \mathbb{E}[\Gamma_i(j,k)]^2 \approx (2-\alpha)\left(\frac{1}{TN^2} + \frac{1}{N^3}\right).$$

The range of $\Gamma_i(j,k)$ is

$$|\Gamma_i(j,k) - \mathbb{E}[\Gamma_i(j,k)]| \lessgtr 1,$$

and the extreme case is when $\bar{y} = j$ and all of the sequence except the last are $k$.

$\square$

## D.3 COMPLETING THE PROOF OF THEOREM 1

**Theorem 4** (Restatement of Theorem 1). *Assume $N, T \gg 1, \alpha = \Theta(1)$. Consider a one gradient step update from zero-initialization on $m$ i.i.d. samples of $z_{1:T}$ with separate learning rates $\eta_f$ for $\mathbf{W}_F$ and $\eta_v$ for $\mathbf{W}_V$ (note that the gradient on $\mathbf{W}_{QK}$ is zero). For a test sequence $z_{1:T}$, the resulting logits for the feed-forward and attention blocks satisfy, with probability $1 - \delta$*

$$|\Delta(\xi_{ff}(x_{1:T})) - \eta_f \cdot \alpha| \leq \eta_f \cdot O\left(\sqrt{\frac{\ln \frac{2(N+1)}{\delta}}{m}}\right),$$

$$\left|\Delta(\xi_{attn}(x_{1:T})) - \frac{\eta_v}{N} \cdot (\alpha^2 \hat{q} + \alpha(1 - \hat{q}))\right| \leq \eta_v \cdot O\left(\sqrt{\frac{(\frac{1}{TN} + \frac{1}{N^2})\ln \frac{2(N+1)}{\delta}}{m}} + \frac{\ln \frac{2(N+1)}{\delta}}{m}\right),$$

*where $\Delta(\xi) = \xi_{N+1} - \max_{j \in [N]} \xi_j$ is the margin of predicting the noise token and $\hat{q} = \frac{1}{T}\sum_{t \leq T} \mathbb{1}\{z_t = N + 1\}$.*

*Proof.* For $\mathbf{W}_F$, since the input is always $z_T = q$, the logits will be $[\xi_{\mathrm{ff}}]_k = \mathbf{W}_U(k)^\top \mathbf{W}_F \mathbf{W}_E(q)$, $\forall\, k \in [N+1]$. As $\mathbf{W}_F$ is initialized from 0 and updated by GD with learning rate $\eta_f$, after one-step update, we have

$$\xi_{\mathrm{ff}} = \mathbf{W}_U(k)^\top \left( -\eta_f \nabla_{\mathbf{W}_F} \hat{L} \Big|_{\mathbf{W}_F = 0} \right) \mathbf{W}_E(q) \in \mathbb{R}^{N+1}.$$

By Lemma D.1, with probability $1 - \frac{1}{2}\delta$, we have

$$\left| [\xi_{\mathrm{ff}}]_{N+1} - \eta_f \cdot \alpha \right| \leq \eta_f \cdot O\left( \sqrt{\frac{\ln \frac{2(N+1)}{\delta}}{m}} \right),$$

$$\forall\, k \leq N, \quad \left| [\xi_{\mathrm{ff}}]_k - \eta_f \cdot \left( \frac{1-\alpha}{N} - \frac{1}{N+1} \right) \right| \leq \eta_f \cdot O\left( \sqrt{\frac{\ln \frac{2(N+1)}{\delta}}{Nm}} + \frac{\ln \frac{2(N+1)}{\delta}}{m} \right),$$

and then triangle inequality finishes the proof for $\xi_{\mathrm{ff}}$.

For $\mathbf{W}_V$, since the gradient on $\mathbf{W}_{QK}$ at initialization is zero, $\mathbf{W}_{QK}$ being zero after the first step induces a uniform attention over the input sequence. Consider the input sequence $\{z_i\}_{i=1}^T$, then the logits will be $[\xi_{\mathrm{attn}}]_j = \mathbf{W}_U(j)^\top \mathbf{W}_V \frac{1}{T} \sum_{t=1}^T \mathbf{W}_E(z_t)$, $\forall\, j \in [N+1]$.

Then considering the concentration bound of $\mathbf{W}_V$ after one-step update in Lemma D.2, denoting $\Gamma(j, k) = \mathbf{W}_U(j)^\top \mathbf{W}_V \mathbf{W}_E(k)$, we have

$$[\xi_{\mathrm{attn}}]_j = \frac{1}{T} \sum_{t \leq T} \Gamma(j, z_t) = \frac{1}{T} \sum_{k \leq N+1} n_k \cdot \Gamma(j, k),$$

with concentration bound for each $\Gamma(\cdot, \cdot)$ in Lemma D.2. From Table 3, note that for all $j = N+1, k \leq N$, the expectation and variances are the same, while $k = N+1$ has slightly different expectation and variance (but still in the same order of the others). Hence, denoting $\hat{q} = \frac{1}{T} \sum_{t \leq T} \mathbb{1}\{z_t = N+1\}$ dependent of the test sample $z_{1:T}$, we have

$$\left| [\xi_{\mathrm{attn}}(x_{1:T})]_{N+1} - \frac{\eta_v}{N} \cdot (\alpha^2 \hat{q} + \alpha(1 - \hat{q})) \right| \leq \eta_v \cdot O\left( \sqrt{\frac{(\frac{1}{TN} + \frac{1}{N^2}) \ln \frac{2(N+1)}{\delta}}{m}} + \frac{\ln \frac{2(N+1)}{\delta}}{m} \right).$$

Meanwhile, as the terms in Table 3 for $j \neq N+1$ always have much smaller mean and variance by a factor $1/N$, using the Bernstein's inequalites for these terms in Lemma D.2 finishes the proof for $\mathbf{W}_V$.

$\square$

# E   PROOF FOR FIRST AND SECOND MOMENTS IN LEMMA D.2

In this section, we will show the proof of the first and second moments of $\left[ \sum_{1 \leq t \leq T} \mathbb{1}\{z_t = k\} | \cdot \right]$ for all cases. Note that we do not consider $z_T = q$, but including it will not change the results, as $T \gg 1$ and $z_T$ is explicitly fixed as $q$ during data generation in Section 3. Generally, there are three factors to classify the cases as follows:

1. The i.i.d. uniformly sampled correct token $\bar{y} \in [N]$:
   (a) $\bar{y} = q$,
   (b) $\bar{y} \neq q$.
2. The target token $k \in [N+1]$:
   (a) $k = q$,
   (b) $k = N+1$.
   (c) $k \leq N, k \neq q, k \neq \bar{y}$,
   (d) (if $\bar{y} \neq q$) $k \leq N, k \neq q, k = \bar{y}$,

3. A condition about the token $z_0$ before the sequence $\{z_t\}_{t \geq 1}$:

   (a) $z_0 = q$,
   (b) $z_0 \in [N+1] \setminus \{q\}$.

Note that when $z_0$ will be implicitly or explicitly considered. When there is no condition on the first token, which means $z_1 \sim \text{Uniform}([N])$, this belongs to Case (3b), *i.e.*, $z_0 \in [N+1] \setminus \{q\}$, following the data generation process.

Table 4 summarizes all lemmas about the seven cases classified by the first two factors. The third factor about $z_0$ is explicitly presented in the proof of each corresponding lemma.

Table 4: All lemmas about the seven cases classified by $\bar{y}$ and $k$.

|       | (2a) | (2b) | (2c) | (2d) |
|-------|------|------|------|------|
| (1a)  | E.1  | E.2  | E.3  | N/A  |
| (1b)  | E.4  | E.5  | E.7  | E.6  |

## E.1  WHEN $\bar{y} = q$

**Lemma E.1** ($\bar{y} = q, k = q$)**.** *Following the data generation process, assuming $N, T \gg 1$ and $\alpha = \Theta(1)$, if $\bar{y} = q$ and $k = q$, it holds*

$$
\mathbb{E}\left[\sum_{t \leq T} \mathbb{1}\{z_t = k\} \middle| \bar{y} = q, k = q\right] \approx \frac{T}{\alpha N},
$$

$$
\mathbb{E}\left[\left(\sum_{t \leq T} \mathbb{1}\{z_t = k\}\right)^2 \middle| \bar{y} = q, k = q\right] \approx \frac{T}{\alpha N}\left(-1 + \frac{2}{\alpha^2}\right) + \frac{T^2}{\alpha^2 N^2}.
$$

(16)

*Proof.* For simplicity, we omit the condition of $\bar{y} = q, k = q$ in this proof. Denote

$$
Y(T) \triangleq \mathbb{E}\left[\sum_{t \leq T} \mathbb{1}\{z_t = k\} \middle| z_0 = q\right],
$$

$$
\hat{Y}(T) \triangleq \mathbb{E}\left[\sum_{t \leq T} \mathbb{1}\{z_t = k\} \middle| z_0 \in [N+1], z_0 \neq q\right].
$$

Then the data generation process implies, $\forall\, T \geq 1$,

$$
Y(T) = p(z_1 = q | z_0 = q) \cdot (1 + Y(T-1)) + p(z_1 = N+1 | z_0 = q) \cdot \hat{Y}(T-1),
$$

$$
\hat{Y}(T) = p(z_1 = q | z_0 \neq q) \cdot (1 + Y(T-1)) + p(z_1 \in [N] \setminus \{q\} | z_0 \neq q) \cdot \hat{Y}(T-1).
$$

The iteration becomes

$$
Y(T) = (1 - \alpha) \cdot Y(T-1) + \alpha \cdot \hat{Y}(T-1) + 1 - \alpha,
$$

$$
\hat{Y}(T) = \frac{1}{N} \cdot Y(T-1) + \frac{N-1}{N} \cdot \hat{Y}(T-1) + \frac{1}{N}.
$$

This gives

$$
Y(T) - \hat{Y}(T) = (1 - \alpha - \frac{1}{N})(Y(T-1) - \hat{Y}(T-1)) + 1 - \alpha - \frac{1}{N},
$$

$$
\frac{1}{N}Y(T) + \alpha\hat{Y}(T) = \frac{1}{N}Y(T-1) + \alpha\hat{Y}(T-1) + \frac{1}{N}.
$$

Consider the initialization $Y(0) = \hat{Y}(0) = 0$. This implies

$$Y(T) - \hat{Y}(T) = \frac{1 - \alpha - \frac{1}{N}}{\alpha + \frac{1}{N}}\left(1 - \left(1 - \alpha - \frac{1}{N}\right)^T\right),$$

$$\frac{1}{N}Y(T) + \alpha\hat{Y}(T) = \frac{1}{N}T.$$

Then we obtain

$$Y(T) \approx \frac{1}{\alpha N + 1}(T - \alpha N) + \frac{\alpha}{(\alpha + \frac{1}{N})^2} = \frac{1}{\alpha N + 1}\left(T - \alpha N + \frac{N^2}{\alpha N + 1}\right)$$

$$\approx \frac{T}{\alpha N} - 1 + \frac{1}{\alpha^2},$$

$$\hat{Y}(T) \approx \frac{1}{\alpha N + 1}T - \frac{N}{(\alpha N + 1)^2} + \frac{1}{\alpha N + 1}$$

$$\approx \frac{T}{\alpha N}.$$

Since the data generation process implicitly assumes $z_0 \neq q$, we have the desired expectation as

$$\mathbb{E}\left[\sum_{t \leq T} \mathbb{1}\{z_t = k\}\bigg|\bar{y} = q, k = q\right] = \hat{Y}(T) \approx \frac{T}{\alpha N}.$$

To obtain the expectation of the quadratic term, we similarly denote the following terms with different $z_0$:

$$Z(T) \triangleq \mathbb{E}\left[\left(\sum_{t \leq T} \mathbb{1}\{z_t = k\}\right)^2\bigg|z_0 = q\right],$$

$$\hat{Z}(T) \triangleq \mathbb{E}\left[\left(\sum_{t \leq T} \mathbb{1}\{z_t = k\}\right)^2\bigg|z_0 \in [N+1], z_0 \neq q\right].$$

Then the data generation process implies, $\forall\, T \geq 1$,

$$Z(T) = p(z_1 = q|z_0 = q) \cdot (1 + 2Y(T-1) + Z(T-1)) + p(z_1 = N+1|z_0 = q) \cdot Z(T-1),$$

$$\hat{Z}(T) = p(z_1 = q|z_0 \neq q) \cdot (1 + 2Y(T-1) + Z(T-1)) + p(z_1 \neq q|z_0 \neq q) \cdot \hat{Z}(T-1),$$

where $2Y(T-1)$ is due to $\mathbb{E}[(1 + \sum_{2 \leq t \leq T} \cdot)^2] = 1 + 2\mathbb{E}[\sum_{2 \leq t \leq T} \cdot] + \mathbb{E}[(\sum_{2 \leq t \leq T} \cdot)^2]$.

Then the iteration becomes

$$Z(T) = (1 - \alpha) \cdot (1 + 2Y(T-1) + Z(T-1)) + \alpha \cdot \hat{Z}(T-1)$$

$$= (1 - \alpha)Z(T-1) + \alpha\hat{Z}(T-1) + (1 - \alpha)(1 + 2Y(T-1)),$$

$$\hat{Z}(T) = \frac{1}{N} \cdot (1 + 2Y(T-1) + Z(T-1)) + \frac{N-1}{N} \cdot \hat{Z}(T-1)$$

$$= \frac{1}{N}Z(T-1) + \frac{N-1}{N}\hat{Z}(T-1) + \frac{1}{N}(1 + 2Y(T-1)).$$

This gives

$$Z(T) - \hat{Z}(T) = (1 - \alpha - \frac{1}{N})(Z(T-1) - \hat{Z}(T-1)) + (1 - \alpha - \frac{1}{N})(1 + 2Y(T-1)),$$

$$\frac{1}{N}Z(T) + \alpha\hat{Z}(T) = \frac{1}{N}Z(T-1) + \alpha\hat{Z}(T-1) + \frac{1}{N}(1 + 2Y(T-1)).$$

Considering the initialization $Z(0) = \hat{Z}(0) = 0$, we have

$$Z(T) - \hat{Z}(T) = \sum_{t \leq T-1} (1 - \alpha - \frac{1}{N})^{T-t}(1 + 2Y(t))$$

$$\approx \sum_{t \leq T-1} (1 - \alpha - \frac{1}{N})^{T-t} \left( 1 + \frac{2t}{\alpha N} - 2 + \frac{2}{\alpha^2} \right)$$

$$\approx \left( -1 + \frac{2}{\alpha^2} \right) \frac{1 - \alpha}{\alpha} + \frac{2(1 - \alpha)}{\alpha^2} \cdot \frac{T}{N}.$$

$$\frac{1}{N} Z(T) + \alpha \hat{Z}(T) = \frac{T}{N} + \frac{2}{N} \sum_{1 \leq t \leq T-1} Y(t)$$

$$\approx \frac{T}{N} + \frac{2}{N} \sum_{1 \leq t \leq T-1} \left( \frac{t}{\alpha N} - 1 + \frac{1}{\alpha^2} \right)$$

$$\approx \frac{T}{N} \left( -1 + \frac{2}{\alpha^2} \right) + \frac{T^2}{\alpha N^2}.$$

Then we obtain

$$Z(T) \approx \frac{T}{N} \left( -\frac{3}{\alpha} + \frac{2}{\alpha^2} + \frac{2}{\alpha^3} \right) + \frac{T^2}{\alpha^2 N^2} + \frac{1 - \alpha}{\alpha}(\frac{2}{\alpha^2} - 1),$$

$$\hat{Z}(T) \approx \frac{T}{\alpha N} \left( -1 + \frac{2}{\alpha^2} \right) + \frac{T^2}{\alpha^2 N^2}.$$

Since the data generation process implicitly assumes $z_0 \neq q$, we have the desired expectation as

$$\mathbb{E} \left[ \left( \sum_{t \leq T} \mathbb{1}\{z_t = k\} \right)^2 \Big| \bar{y} = q, k = q \right] = \hat{Z}(T) \approx \frac{T}{\alpha N} \left( -1 + \frac{2}{\alpha^2} \right) + \frac{T^2}{\alpha^2 N^2}.$$

$\square$

**Lemma E.2** ($\bar{y} = q, k = N + 1$). *Following the data generation process, assuming $N, T \gg 1$ and $\alpha = \Theta(1)$, if $\bar{y} = q$ and $k = N + 1$, it holds*

$$\mathbb{E} \left[ \sum_{t \leq T} \mathbb{1}\{z_t = k\} \Big| \bar{y} = q, k = N + 1 \right] \approx \frac{T}{N},$$

$$\mathbb{E} \left[ \left( \sum_{t \leq T} \mathbb{1}\{z_t = k\} \right)^2 \Big| \bar{y} = q, k = N + 1 \right] \approx \frac{T}{N} + \frac{T^2}{N^2}.$$

(17)

*Proof.* For simplicity, we omit the condition of $\bar{y} = q, k = N + 1$ in this proof. Denote

$$Y(T) \triangleq \mathbb{E} \left[ \sum_{t \leq T} \mathbb{1}\{z_t = k\} \Big| z_0 = q \right],$$

$$\hat{Y}(T) \triangleq \mathbb{E} \left[ \sum_{t \leq T} \mathbb{1}\{z_t = k\} \Big| z_0 \in [N + 1], z_0 \neq q \right].$$

Then the data generation process implies, $\forall T \geq 1$,

$$Y(T) = p(z_1 = q | z_0 = q) \cdot Y(T - 1) + p(z_1 = N + 1 | z_0 = q) \cdot (1 + \hat{Y}(T - 1)),$$

$$\hat{Y}(T) = p(z_1 = q | z_0 \neq q) \cdot Y(T - 1) + p(z_1 \in [N] \setminus \{q\} | z_0 \neq q) \cdot \hat{Y}(T - 1).$$

The iteration becomes

$$Y(T) = (1 - \alpha) \cdot Y(T - 1) + \alpha \cdot \hat{Y}(T - 1) + \alpha,$$
$$\hat{Y}(T) = \frac{1}{N} \cdot Y(T - 1) + \frac{N - 1}{N} \cdot \hat{Y}(T - 1).$$

This gives

$$Y(T) - \hat{Y}(T) = (1 - \alpha - \frac{1}{N})(Y(T - 1) - \hat{Y}(T - 1)) + \alpha,$$
$$\frac{1}{N}Y(T) + \alpha\hat{Y}(T) = \frac{1}{N}Y(T - 1) + \alpha\hat{Y}(T - 1) + \frac{\alpha}{N}.$$

Consider the initialization $Y(0) = \hat{Y}(0) = 0$. This implies

$$Y(T) - \hat{Y}(T) = \frac{\alpha}{\alpha + \frac{1}{N}} \left( 1 - \left( 1 - \alpha - \frac{1}{N} \right)^T \right),$$
$$\frac{1}{N}Y(T) + \alpha\hat{Y}(T) = \frac{\alpha}{N}T.$$

Then we obtain

$$Y(T) \approx \frac{T}{N} + 1,$$
$$\hat{Y}(T) \approx \frac{T}{N}.$$

Since the data generation process implicitly assumes $z_0 \neq q$, we have the desired expectation as

$$\mathbb{E}\left[ \sum_{t \leq T} \mathbb{1}\{z_t = k\} \Big| \bar{y} = q, k = N + 1 \right] = \hat{Y}(T) \approx \frac{T}{N}.$$

To obtain the expectation of the quadratic term, we similarly denote the following terms with different $z_0$:

$$Z(T) \triangleq \mathbb{E}\left[ \left( \sum_{t \leq T} \mathbb{1}\{z_t = k\} \right)^2 \Big| z_0 = q \right],$$
$$\hat{Z}(T) \triangleq \mathbb{E}\left[ \left( \sum_{t \leq T} \mathbb{1}\{z_t = k\} \right)^2 \Big| z_0 \in [N + 1], z_0 \neq q \right].$$

Then the data generation process implies, $\forall\, T \geq 1$,

$$Z(T) = p(z_1 = q | z_0 = q) \cdot Z(T - 1) + p(z_1 = N + 1 | z_0 = q) \cdot (1 + 2\hat{Y}(T - 1) + \hat{Z}(T - 1)),$$
$$\hat{Z}(T) = p(z_1 = q | z_0 \neq q) \cdot Z(T - 1) + p(z_1 \neq q | z_0 \neq q) \cdot \hat{Z}(T - 1),$$

where $2\hat{Y}(T - 1)$ is due to $\mathbb{E}[(1 + \sum_{2 \leq t \leq T} \cdot)^2] = 1 + 2\mathbb{E}[\sum_{2 \leq t \leq T} \cdot] + \mathbb{E}[(\sum_{2 \leq t \leq T} \cdot)^2]$.

Then the iteration becomes

$$Z(T) = (1 - \alpha) \cdot Z(T - 1) + \alpha \cdot (1 + 2\hat{Y}(T - 1) + \hat{Z}(T - 1))$$
$$= (1 - \alpha)Z(T - 1) + \alpha\hat{Z}(T - 1) + \alpha(1 + 2\hat{Y}(T - 1)),$$
$$\hat{Z}(T) = \frac{1}{N} \cdot Z(T - 1) + \frac{N - 1}{N} \cdot \hat{Z}(T - 1).$$

This gives

$$Z(T) - \hat{Z}(T) = (1 - \alpha - \frac{1}{N})(Z(T - 1) - \hat{Z}(T - 1)) + \alpha(1 + 2\hat{Y}(T - 1)),$$
$$\frac{1}{N}Z(T) + \alpha\hat{Z}(T) = \frac{1}{N}Z(T - 1) + \alpha\hat{Z}(T - 1) + \frac{\alpha}{N}(1 + 2\hat{Y}(T - 1)).$$

Considering the initialization $Z(0) = \hat{Z}(0) = 0$, we have

$$
\begin{aligned}
Z(T) - \hat{Z}(T) &= \sum_{t \leq T-1} \alpha (1 - \alpha - \frac{1}{N})^{T-1-t}(1 + 2\hat{Y}(t)) \\
&\approx \sum_{t \leq T-1} \alpha (1 - \alpha - \frac{1}{N})^{T-1-t}\left(1 + \frac{2t}{N}\right) \\
&\approx \frac{2T}{N} + 1,
\end{aligned}
$$

$$
\begin{aligned}
\frac{1}{N}Z(T) + \alpha \hat{Z}(T) &= \frac{\alpha T}{N} + \frac{2\alpha}{N} \sum_{1 \leq t \leq T-1} \hat{Y}(t) \\
&\approx \frac{\alpha T}{N} + \frac{2\alpha}{N} \sum_{1 \leq t \leq T-1} \frac{t}{N} \\
&\approx \frac{\alpha T}{N} + \frac{\alpha T^2}{N^2}.
\end{aligned}
$$

Then we obtain

$$
Z(T) \approx 3\alpha \frac{T}{N} + \alpha \frac{T^2}{N^2} + \alpha,
$$

$$
\hat{Z}(T) \approx \frac{T}{N} + \frac{T^2}{N^2}.
$$

Since the data generation process implicitly assumes $z_0 \neq q$, we have the desired expectation as

$$
\mathbb{E}\left[\left(\sum_{t \leq T} \mathbb{1}\{z_t = k\}\right)^2 \middle| \bar{y} = q, k = N + 1\right] = \hat{Z}(T) \approx \frac{T}{N} + \frac{T^2}{N^2}.
$$

$\square$

**Lemma E.3** ($\bar{y} = q, k \leq N, k \neq q$). *Following the data generation process, assuming $N, T \gg 1$ and $\alpha = \Theta(1)$, if $\bar{y} = q$ and $k \in [N] \setminus \{q\}$, it holds*

$$
\begin{aligned}
\mathbb{E}\left[\sum_{t \leq T} \mathbb{1}\{z_t = k\} \middle| \bar{y} = q, k \in [N] \setminus \{q\}\right] &\approx \frac{T}{N}, \\
\mathbb{E}\left[\left(\sum_{t \leq T} \mathbb{1}\{z_t = k\}\right)^2 \middle| \bar{y} = q, k \in [N] \setminus \{q\}\right] &\approx \frac{T}{N} + \frac{T^2}{N^2}.
\end{aligned} \tag{18}
$$

*Proof.* For simplicity, we omit the condition of $\bar{y} = q, k \in [N] \setminus \{q\}$ in this proof. Denote

$$
Y(T) \triangleq \mathbb{E}\left[\sum_{t \leq T} \mathbb{1}\{z_t = k\} \middle| z_0 = q\right],
$$

$$
\hat{Y}(T) \triangleq \mathbb{E}\left[\sum_{t \leq T} \mathbb{1}\{z_t = k\} \middle| z_0 \in [N + 1], z_0 \neq q\right].
$$

Then the data generation process implies, $\forall\, T \geq 1$,

$$
\begin{aligned}
Y(T) &= p(z_1 = q | z_0 = q) \cdot Y(T - 1) + p(z_1 = N + 1 | z_0 = q) \cdot \hat{Y}(T - 1), \\
\hat{Y}(T) &= p(z_1 = q | z_0 \neq q) \cdot Y(T - 1) \\
&\quad + p(z_1 \in [N] \setminus \{q\} | z_0 \neq q) \cdot (p(z_1 = k | z_1 \sim \text{Uniform}([N] \setminus \{q\}) + \hat{Y}(T - 1)).
\end{aligned}
$$

The iteration becomes

$$Y(T) = (1 - \alpha) \cdot Y(T - 1) + \alpha \cdot \hat{Y}(T - 1),$$
$$\hat{Y}(T) = \frac{1}{N} \cdot Y(T - 1) + \frac{N - 1}{N} \cdot (\hat{Y}(T - 1) + \frac{1}{N - 1}).$$

This gives

$$Y(T) - \hat{Y}(T) = (1 - \alpha - \frac{1}{N})(Y(T - 1) - \hat{Y}(T - 1)) - \frac{1}{N},$$
$$\frac{1}{N}Y(T) + \alpha\hat{Y}(T) = \frac{1}{N}Y(T - 1) + \alpha\hat{Y}(T - 1) + \frac{\alpha}{N}.$$

Consider the initialization $Y(0) = \hat{Y}(0) = 0$. This implies

$$Y(T) - \hat{Y}(T) = \frac{-\frac{1}{N}}{\alpha + \frac{1}{N}} \left(1 - \left(1 - \alpha - \frac{1}{N}\right)^T\right),$$
$$\frac{1}{N}Y(T) + \alpha\hat{Y}(T) = \frac{\alpha}{N}T.$$

Then we obtain

$$Y(T) \approx \frac{T}{N},$$
$$\hat{Y}(T) \approx \frac{T}{N}.$$

Since the data generation process implicitly assumes $z_0 \neq q$, we have the desired expectation as

$$\mathbb{E}\left[\sum_{t \leq T} \mathbb{1}\{z_t = k\} \Big| \bar{y} = q, k = N + 1\right] = \hat{Y}(T) \approx \frac{T}{N}.$$

To obtain the expectation of the quadratic term, we similarly denote the following terms with different $z_0$:

$$Z(T) \triangleq \mathbb{E}\left[\left(\sum_{t \leq T} \mathbb{1}\{z_t = k\}\right)^2 \Big| z_0 = q\right],$$
$$\hat{Z}(T) \triangleq \mathbb{E}\left[\left(\sum_{t \leq T} \mathbb{1}\{z_t = k\}\right)^2 \Big| z_0 \in [N + 1], z_0 \neq q\right].$$

Then the data generation process implies, $\forall \, T \geq 1$,

$$Z(T) = p(z_1 = q|z_0 = q) \cdot Z(T - 1) + p(z_1 = N + 1|z_0 = q) \cdot \hat{Z}(T - 1),$$
$$\hat{Z}(T) = p(z_1 = q|z_0 \neq q) \cdot Z(T - 1) + p(z_1 \neq q|z_0 \neq q) \cdot \hat{Z}(T - 1)$$
$$+ p(z_1 = k|z_0 \neq q) \cdot (1 + 2\hat{Y}(T - 1)),$$

where $2\hat{Y}(T - 1)$ is due to $\mathbb{E}[(1 + \sum_{2 \leq t \leq T} \cdot)^2] = 1 + 2\mathbb{E}[\sum_{2 \leq t \leq T} \cdot] + \mathbb{E}[(\sum_{2 \leq t \leq T} \cdot)^2]$.

Then the iteration becomes

$$Z(T) = (1 - \alpha) \cdot Z(T - 1) + \alpha \cdot \hat{Z}(T - 1),$$
$$\hat{Z}(T) = \frac{1}{N} \cdot Z(T - 1) + \frac{N - 1}{N} \cdot \hat{Z}(T - 1) + \frac{1}{N}(1 + 2\hat{Y}(T - 1)).$$

This gives

$$Z(T) - \hat{Z}(T) = (1 - \alpha - \frac{1}{N})(Z(T - 1) - \hat{Z}(T - 1)) - \frac{1}{N}(1 + 2\hat{Y}(T - 1)),$$
$$\frac{1}{N}Z(T) + \alpha\hat{Z}(T) = \frac{1}{N}Z(T - 1) + \alpha\hat{Z}(T - 1) + \frac{\alpha}{N}(1 + 2\hat{Y}(T - 1)).$$

Considering the initialization $Z(0) = \hat{Z}(0) = 0$, we have

$$
\begin{aligned}
Z(T) - \hat{Z}(T) &= -\frac{1}{N} \sum_{t \leq T-1} (1 - \alpha - \frac{1}{N})^{T-1-t}(1 + 2\hat{Y}(t)) \\
&\approx -\frac{1}{N} \sum_{t \leq T-1} (1 - \alpha - \frac{1}{N})^{T-1-t} \left(1 + \frac{2t}{N}\right) \\
&\approx -\frac{1}{\alpha N} \left(\frac{2T}{N} + 1\right),
\end{aligned}
$$

$$
\begin{aligned}
\frac{1}{N} Z(T) + \alpha \hat{Z}(T) &= \frac{\alpha T}{N} + \frac{2\alpha}{N} \sum_{1 \leq t \leq T-1} \hat{Y}(t) \\
&\approx \frac{\alpha T}{N} + \frac{2\alpha}{N} \sum_{1 \leq t \leq T-1} \frac{t}{N} \\
&\approx \frac{\alpha T}{N} + \frac{\alpha T^2}{N^2}.
\end{aligned}
$$

Then we obtain

$$
Z(T) \approx \frac{T}{N} + \frac{T^2}{N^2},
$$

$$
\hat{Z}(T) \approx \frac{T}{N} + \frac{T^2}{N^2}.
$$

Since the data generation process implicitly assumes $z_0 \neq q$, we have the desired expectation as

$$
\mathbb{E} \left[ \left( \sum_{t \leq T} \mathbb{1}\{z_t = k\} \right)^2 \middle| \bar{y} = q, k \in [N] \setminus \{q\} \right] = \hat{Z}(T) \approx \frac{T}{N} + \frac{T^2}{N^2}.
$$

$\square$

### E.2 WHEN $\bar{y} \neq q$

**Lemma E.4** ($\bar{y} \neq q, k = q$). *Following the data generation process, assuming $N, T \gg 1$ and $\alpha = \Theta(1)$, if $\bar{y} \neq q$ and $k = q$, it holds*

$$
\begin{aligned}
\mathbb{E} \left[ \sum_{t \leq T} \mathbb{1}\{z_t = k\} \middle| \bar{y} \neq q, k = q \right] &\approx \frac{T}{N}, \\
\mathbb{E} \left[ \left( \sum_{t \leq T} \mathbb{1}\{z_t = k\} \right)^2 \middle| \bar{y} \neq q, k = q \right] &\approx \frac{T}{N} + \frac{T^2}{N^2}.
\end{aligned}
\tag{19}
$$

*Proof.* For simplicity, we omit the condition of $\bar{y} \neq q, k = q$ in this proof. Denote

$$
Y(T) \triangleq \mathbb{E} \left[ \sum_{t \leq T} \mathbb{1}\{z_t = k\} \middle| z_0 = q \right],
$$

$$
\hat{Y}(T) \triangleq \mathbb{E} \left[ \sum_{t \leq T} \mathbb{1}\{z_t = k\} \middle| z_0 \in [N+1], z_0 \neq q \right].
$$

Then the data generation process implies, $\forall T \geq 1$,

$$Y(T) = \hat{Y}(T-1),$$
$$\hat{Y}(T) = p(z_1 = q | z_0 \neq q) \cdot (1 + Y(T-1)) + p(z_1 \in [N] \setminus \{q\} | z_0 \neq q) \cdot \hat{Y}(T-1).$$

The iteration becomes

$$Y(T) = \hat{Y}(T-1),$$
$$\hat{Y}(T) = \frac{1}{N} \cdot Y(T-1) + \frac{N-1}{N} \cdot \hat{Y}(T-1) + \frac{1}{N}.$$

This gives

$$Y(T) - \hat{Y}(T) = -\frac{1}{N}(Y(T-1) - \hat{Y}(T-1)) - \frac{1}{N},$$
$$\frac{1}{N}Y(T) + \hat{Y}(T) = \frac{1}{N}Y(T-1) + \hat{Y}(T-1) + \frac{1}{N}.$$

Consider the initialization $Y(0) = \hat{Y}(0) = 0$. This implies

$$Y(T) - \hat{Y}(T) = \frac{-\frac{1}{N}}{1 + \frac{1}{N}}\left(1 - \left(-\frac{1}{N}\right)^T\right),$$
$$\frac{1}{N}Y(T) + \hat{Y}(T) = \frac{1}{N}T.$$

Then we obtain

$$Y(T) \approx \frac{T}{N},$$
$$\hat{Y}(T) \approx \frac{T}{N}.$$

Since the data generation process implicitly assumes $z_0 \neq q$, we have the desired expectation as

$$\mathbb{E}\left[\sum_{t \leq T} \mathbb{1}\{z_t = k\} \Big| \bar{y} \neq q, k = q\right] = \hat{Y}(T) \approx \frac{T}{N}.$$

To obtain the expectation of the quadratic term, we similarly denote the following terms with different $z_0$:

$$Z(T) \triangleq \mathbb{E}\left[\left(\sum_{t \leq T} \mathbb{1}\{z_t = k\}\right)^2 \Big| z_0 = q\right],$$
$$\hat{Z}(T) \triangleq \mathbb{E}\left[\left(\sum_{t \leq T} \mathbb{1}\{z_t = k\}\right)^2 \Big| z_0 \in [N+1], z_0 \neq q\right].$$

Then the data generation process implies, $\forall\, T \geq 1$,

$Z(T) = \hat{Z}(T-1),$

$\hat{Z}(T) = p(z_1 = q | z_0 \neq q) \cdot (1 + 2Y(T-1) + Z(T-1)) + p(z_1 \in [N] \setminus \{q\} | z_0 \neq q) \cdot \hat{Z}(T-1),$

where $2Y(T-1)$ is due to $\mathbb{E}[(1 + \sum_{2 \leq t \leq T} \cdot)^2] = 1 + 2\mathbb{E}[\sum_{2 \leq t \leq T} \cdot] + \mathbb{E}[(\sum_{2 \leq t \leq T} \cdot)^2].$

Then the iteration becomes

$$Z(T) = \hat{Z}(T-1),$$
$$\hat{Z}(T) = \frac{1}{N}Z(T-1) + \frac{N-1}{N}\hat{Z}(T-1) + \frac{1}{N}(1 + 2Y(T-1)).$$

This gives

$$Z(T) - \hat{Z}(T) = -\frac{1}{N}(Z(T-1) - \hat{Z}(T-1)) - \frac{1}{N}(1 + 2Y(T-1)),$$
$$\frac{1}{N}Z(T) + \hat{Z}(T) = \frac{1}{N}Z(T-1) + \hat{Z}(T-1) + \frac{1}{N}(1 + 2Y(T-1)).$$

Considering the initialization $Z(0) = \hat{Z}(0) = 0$, we have

$$Z(T) - \hat{Z}(T) = -\frac{1}{N} \sum_{t \leq T-1} (-\frac{1}{N})^{T-1-t} (1 + 2Y(t))$$

$$\approx -\frac{1}{N} \sum_{t \leq T-1} (-\frac{1}{N})^{T-1-t} \left(1 + \frac{2t}{N}\right)$$

$$\approx -\frac{1}{N} - \frac{2T}{N^2},$$

$$\frac{1}{N}Z(T) + \hat{Z}(T) = \frac{T}{N} + \frac{2}{N} \sum_{1 \leq t \leq T-1} Y(t)$$

$$\approx \frac{T}{N} + \frac{2}{N} \sum_{1 \leq t \leq T-1} \frac{t}{N}$$

$$\approx \frac{T}{N} + \frac{T^2}{N^2}.$$

Then we obtain

$$Z(T) \approx \frac{T}{N} + \frac{T^2}{N^2},$$

$$\hat{Z}(T) \approx \frac{T}{N} + \frac{T^2}{N^2}.$$

Since the data generation process implicitly assumes $z_0 \neq q$, we have the desired expectation as

$$\mathbb{E}\left[\left(\sum_{t \leq T} \mathbb{1}\{z_t = k\}\right)^2 \middle| \bar{y} = q, k \in [N] \setminus \{q\}\right] = \hat{Z}(T) \approx \frac{T}{N} + \frac{T^2}{N^2}.$$

$\square$

**Lemma E.5** ($\bar{y} \neq q, k = N + 1$). *Following the data generation process, assuming $N, T \gg 1$ and $\alpha = \Theta(1)$, if $\bar{y} \neq q$ and $k = N + 1$, it holds*

$$\mathbb{E}\left[\sum_{t \leq T} \mathbb{1}\{z_t = k\} \middle| \bar{y} \neq q, k = N + 1\right] \approx \frac{\alpha T}{N},$$

$$\mathbb{E}\left[\left(\sum_{t \leq T} \mathbb{1}\{z_t = k\}\right)^2 \middle| \bar{y} \neq q, k = N + 1\right] \approx \frac{\alpha T}{N} + \frac{\alpha^2 T^2}{N^2}. \tag{20}$$

*Proof.* For simplicity, we omit the condition of $\bar{y} \neq q, k = N + 1$ in this proof. Denote

$$Y(T) \triangleq \mathbb{E}\left[\sum_{t \leq T} \mathbb{1}\{z_t = k\} \middle| z_0 = q\right],$$

$$\hat{Y}(T) \triangleq \mathbb{E}\left[\sum_{t \leq T} \mathbb{1}\{z_t = k\} \middle| z_0 \in [N+1], z_0 \neq q\right].$$

Then the data generation process implies, $\forall\, T \geq 1$,

$$Y(T) = \hat{Y}(T-1) + p(z_1 = N+1 | z_0 = q),$$

$$\hat{Y}(T) = p(z_1 = q | z_0 \neq q) \cdot Y(T-1) + p(z_1 \in [N] \setminus \{q\} | z_0 \neq q) \cdot \hat{Y}(T-1).$$

The iteration becomes

$$Y(T) = \hat{Y}(T-1) + \alpha,$$

$$\hat{Y}(T) = \frac{1}{N} \cdot Y(T-1) + \frac{N-1}{N} \cdot \hat{Y}(T-1).$$

This gives

$$Y(T) - \hat{Y}(T) = -\frac{1}{N}(Y(T-1) - \hat{Y}(T-1)) + \alpha,$$

$$\frac{1}{N}Y(T) + \hat{Y}(T) = \frac{1}{N}Y(T-1) + \hat{Y}(T-1) + \frac{\alpha}{N}.$$

Consider the initialization $Y(0) = \hat{Y}(0) = 0$. This implies

$$Y(T) - \hat{Y}(T) = \frac{\alpha}{1 + \frac{1}{N}}\left(1 - \left(-\frac{1}{N}\right)^T\right),$$

$$\frac{1}{N}Y(T) + \hat{Y}(T) = \frac{\alpha}{N}T.$$

Then we obtain

$$Y(T) \approx \frac{\alpha T}{N} + \alpha,$$

$$\hat{Y}(T) \approx \frac{\alpha T}{N}.$$

Since the data generation process implicitly assumes $z_0 \neq q$, we have the desired expectation as

$$\mathbb{E}\left[\sum_{t \leq T} \mathbb{1}\{z_t = k\} \bigg| \bar{y} \neq q, k = q\right] = \hat{Y}(T) \approx \frac{\alpha T}{N}.$$

To obtain the expectation of the quadratic term, we similarly denote the following terms with different $z_0$:

$$Z(T) \triangleq \mathbb{E}\left[\left(\sum_{t \leq T} \mathbb{1}\{z_t = k\}\right)^2 \bigg| z_0 = q\right],$$

$$\hat{Z}(T) \triangleq \mathbb{E}\left[\left(\sum_{t \leq T} \mathbb{1}\{z_t = k\}\right)^2 \bigg| z_0 \in [N+1], z_0 \neq q\right].$$

Then the data generation process implies, $\forall\, T \geq 1$,

$$Z(T) = \hat{Z}(T-1) + p(z_1 = N+1|z_0 = q) \cdot (1 + 2\hat{Y}(T-1)),$$
$$\hat{Z}(T) = p(z_1 = q|z_0 \neq q) \cdot Z(T-1) + p(z_1 \in [N] \setminus \{q\}|z_0 \neq q) \cdot \hat{Z}(T-1),$$

where $2\hat{Y}(T-1)$ is due to $\mathbb{E}[(1 + \sum_{2 \leq t \leq T} \cdot)^2] = 1 + 2\mathbb{E}[\sum_{2 \leq t \leq T} \cdot] + \mathbb{E}[(\sum_{2 \leq t \leq T} \cdot)^2]$.

Then the iteration becomes

$$Z(T) = \hat{Z}(T-1) + \alpha(1 + 2\hat{Y}(T-1)),$$
$$\hat{Z}(T) = \frac{1}{N}Z(T-1) + \frac{N-1}{N}\hat{Z}(T-1).$$

This gives

$$Z(T) - \hat{Z}(T) = -\frac{1}{N}(Z(T-1) - \hat{Z}(T-1)) + \alpha(1 + 2\hat{Y}(T-1)),$$

$$\frac{1}{N}Z(T) + \hat{Z}(T) = \frac{1}{N}Z(T-1) + \hat{Z}(T-1) + \frac{\alpha}{N}(1 + 2\hat{Y}(T-1)).$$

Considering the initialization $Z(0) = \hat{Z}(0) = 0$, we have

$$Z(T) - \hat{Z}(T) = \alpha \sum_{t \leq T-1} (-\frac{1}{N})^{T-1-t}(1 + 2\hat{Y}(t))$$

$$\approx \alpha \sum_{t \leq T-1} (-\frac{1}{N})^{T-1-t} \left(1 + \frac{2\alpha t}{N}\right)$$

$$\approx \frac{2\alpha^2 T}{N} + \alpha,$$

$$\frac{1}{N}Z(T) + \hat{Z}(T) = \frac{\alpha T}{N} + \frac{2\alpha}{N} \sum_{1 \leq t \leq T-1} \hat{Y}(t)$$

$$\approx \frac{\alpha T}{N} + \frac{2\alpha}{N} \sum_{1 \leq t \leq T-1} \frac{\alpha t}{N}$$

$$\approx \frac{\alpha T}{N} + \frac{\alpha^2 T^2}{N^2}.$$

Then we obtain

$$Z(T) \approx \frac{T}{N}(2\alpha^2 + \alpha) + \frac{\alpha^2 T^2}{N^2} + \alpha,$$

$$\hat{Z}(T) \approx \frac{\alpha T}{N} + \frac{\alpha^2 T^2}{N^2}.$$

Since the data generation process implicitly assumes $z_0 \neq q$, we have the desired expectation as

$$\mathbb{E}\left[\left(\sum_{t \leq T} \mathbb{1}\{z_t = k\}\right)^2 \middle| \bar{y} = q, k \in [N] \setminus \{q\}\right] = \hat{Z}(T) \approx \frac{\alpha T}{N} + \frac{\alpha^2 T^2}{N^2}.$$

$\square$

**Lemma E.6** ($\bar{y} \neq q, k = \bar{y}$)**.** *Following the data generation process, assuming $N, T \gg 1$ and $\alpha = \Theta(1)$, if $\bar{y} \neq q$ and $k = \bar{y}$, it holds*

$$\mathbb{E}\left[\sum_{t \leq T} \mathbb{1}\{z_t = k\} \middle| \bar{y} \neq q, k = \bar{y}\right] \approx (2 - \alpha)\frac{T}{N},$$

$$\mathbb{E}\left[\left(\sum_{t \leq T} \mathbb{1}\{z_t = k\}\right)^2 \middle| \bar{y} \neq q, k = \bar{y}\right] \approx \frac{(2 - \alpha)T}{N} + \frac{(2 - \alpha)^2 T^2}{N^2}.$$

(21)

*Proof.* For simplicity, we omit the condition of $\bar{y} \neq q, k = \bar{y}$ in this proof. Denote

$$Y(T) \triangleq \mathbb{E}\left[\sum_{t \leq T} \mathbb{1}\{z_t = k\} \middle| z_0 = q\right],$$

$$\hat{Y}(T) \triangleq \mathbb{E}\left[\sum_{t \leq T} \mathbb{1}\{z_t = k\} \middle| z_0 \in [N + 1], z_0 \neq q\right].$$

Then the data generation process implies, $\forall T \geq 1$,

$Y(T) = \hat{Y}(T - 1) + p(z_1 = \bar{y}|z_0 = q)$,

$\hat{Y}(T) = p(z_1 = q|z_0 \neq q) \cdot Y(T - 1) + p(z_1 \in [N] \setminus \{q\}|z_0 \neq q) \cdot \hat{Y}(T - 1) + p(z_1 = \bar{y}|z_0 \neq q)$.

The iteration becomes

$$Y(T) = \hat{Y}(T - 1) + (1 - \alpha),$$

$$\hat{Y}(T) = \frac{1}{N} \cdot Y(T - 1) + \frac{N - 1}{N} \cdot \hat{Y}(T - 1) + \frac{1}{N}.$$

This gives

$$Y(T) - \hat{Y}(T) = -\frac{1}{N}(Y(T-1) - \hat{Y}(T-1)) + (1 - \alpha - \frac{1}{N}),$$

$$\frac{1}{N}Y(T) + \hat{Y}(T) = \frac{1}{N}Y(T-1) + \hat{Y}(T-1) + \frac{2-\alpha}{N}.$$

Consider the initialization $Y(0) = \hat{Y}(0) = 0$. This implies

$$Y(T) - \hat{Y}(T) = \frac{1 - \alpha - \frac{1}{N}}{1 + \frac{1}{N}} \left( 1 - \left( -\frac{1}{N} \right)^T \right),$$

$$\frac{1}{N}Y(T) + \hat{Y}(T) = \frac{2-\alpha}{N}T.$$

Then we obtain

$$Y(T) \approx (1 - \alpha) + (2 - \alpha)\frac{T}{N},$$

$$\hat{Y}(T) \approx (2 - \alpha)\frac{T}{N}.$$

Since the data generation process implicitly assumes $z_0 \neq q$, we have the desired expectation as

$$\mathbb{E}\left[ \sum_{t \leq T} \mathbb{1}\{z_t = k\} \middle| \bar{y} \neq q, k = q \right] = \hat{Y}(T) \approx (2 - \alpha)\frac{T}{N}.$$

To obtain the expectation of the quadratic term, we similarly denote the following terms with different $z_0$:

$$Z(T) \triangleq \mathbb{E}\left[ \left( \sum_{t \leq T} \mathbb{1}\{z_t = k\} \right)^2 \middle| z_0 = q \right],$$

$$\hat{Z}(T) \triangleq \mathbb{E}\left[ \left( \sum_{t \leq T} \mathbb{1}\{z_t = k\} \right)^2 \middle| z_0 \in [N+1], z_0 \neq q \right].$$

Then the data generation process implies, $\forall T \geq 1$,

$$Z(T) = \hat{Z}(T-1) + p(z_1 = \bar{y}|z_0 = q) \cdot (1 + 2\hat{Y}(T-1)),$$

$$\hat{Z}(T) = p(z_1 = q|z_0 \neq q) \cdot Z(T-1) + p(z_1 \in [N] \setminus \{q\}|z_0 \neq q) \cdot \hat{Z}(T-1)$$
$$+ p(z_1 = \bar{y}|z_0 \neq q) \cdot (1 + 2\hat{Y}(T-1)),$$

where $2\hat{Y}(T-1)$ is due to $\mathbb{E}[(1 + \sum_{2 \leq t \leq T} \cdot)^2] = 1 + 2\mathbb{E}[\sum_{2 \leq t \leq T} \cdot] + \mathbb{E}[(\sum_{2 \leq t \leq T} \cdot)^2]$.

Then the iteration becomes

$$Z(T) = \hat{Z}(T-1) + (1 - \alpha)(1 + 2\hat{Y}(T-1)),$$

$$\hat{Z}(T) = \frac{1}{N}Z(T-1) + \frac{N-1}{N}\hat{Z}(T-1) + \frac{1}{N}(1 + 2\hat{Y}(T-1)).$$

This gives

$$Z(T) - \hat{Z}(T) = -\frac{1}{N}(Z(T-1) - \hat{Z}(T-1)) + (1 - \alpha - \frac{1}{N})(1 + 2\hat{Y}(T-1)),$$

$$\frac{1}{N}Z(T) + \hat{Z}(T) = \frac{1}{N}Z(T-1) + \hat{Z}(T-1) + \frac{2-\alpha}{N}(1 + 2\hat{Y}(T-1)).$$

Considering the initialization $Z(0) = \hat{Z}(0) = 0$, we have

$$
\begin{aligned}
Z(T) - \hat{Z}(T) &= (1 - \alpha - \frac{1}{N}) \sum_{t \leq T-1} (-\frac{1}{N})^{T-1-t}(1 + 2\hat{Y}(t)) \\
&\approx (1 - \alpha - \frac{1}{N}) \sum_{t \leq T-1} (-\frac{1}{N})^{T-1-t} \left(1 + \frac{2(2-\alpha)t}{N}\right) \\
&\approx (1 - \alpha) \left(1 + \frac{2(2-\alpha)T}{N}\right),
\end{aligned}
$$

$$
\begin{aligned}
\frac{1}{N} Z(T) + \hat{Z}(T) &= \frac{(2-\alpha)T}{N} + \frac{2(2-\alpha)}{N} \sum_{1 \leq t \leq T-1} \hat{Y}(t) \\
&\approx \frac{(2-\alpha)T}{N} + \frac{2(2-\alpha)}{N} \sum_{1 \leq t \leq T-1} \frac{(2-\alpha)t}{N} \\
&\approx \frac{(2-\alpha)T}{N} + \frac{(2-\alpha)^2 T^2}{N^2}.
\end{aligned}
$$

Then we obtain

$$
Z(T) \approx \frac{T}{N}(2-\alpha)(3-2\alpha) + \frac{(2-\alpha)^2 T^2}{N^2} + (1-\alpha),
$$
$$
\hat{Z}(T) \approx \frac{(2-\alpha)T}{N} + \frac{(2-\alpha)^2 T^2}{N^2}.
$$

Since the data generation process implicitly assumes $z_0 \neq q$, we have the desired expectation as

$$
\mathbb{E}\left[\left(\sum_{t \leq T} \mathbb{1}\{z_t = k\}\right)^2 \bigg| \bar{y} = q, k \in [N] \setminus \{q\}\right] = \hat{Z}(T) \approx \frac{(2-\alpha)T}{N} + \frac{(2-\alpha)^2 T^2}{N^2}.
$$

$\square$

**Lemma E.7** ($\bar{y} \neq q, k \leq N, k \neq q, k \neq \bar{y}$)**.** *Following the data generation process, assuming $N, T \gg 1$ and $\alpha = \Theta(1)$, if $\bar{y} \neq q$ and $k \in [N] \setminus \{\bar{y}, q\}$, it holds*

$$
\begin{aligned}
\mathbb{E}\left[\sum_{t \leq T} \mathbb{1}\{z_t = k\} \bigg| \bar{y} \neq q, k \in [N] \setminus \{\bar{y}, q\}\right] &\approx \frac{T}{N}, \\
\mathbb{E}\left[\left(\sum_{t \leq T} \mathbb{1}\{z_t = k\}\right)^2 \bigg| \bar{y} \neq q, k \in [N] \setminus \{\bar{y}, q\}\right] &\approx \frac{T}{N} + \frac{T^2}{N^2}.
\end{aligned}
\tag{22}
$$

*Proof.* For simplicity, we omit the condition of $\bar{y} \neq q, k \in [N] \setminus \{\bar{y}, q\}$ in this proof. Denote

$$
Y(T) \triangleq \mathbb{E}\left[\sum_{t \leq T} \mathbb{1}\{z_t = k\} \bigg| z_0 = q\right],
$$

$$
\hat{Y}(T) \triangleq \mathbb{E}\left[\sum_{t \leq T} \mathbb{1}\{z_t = k\} \bigg| z_0 \in [N+1], z_0 \neq q\right].
$$

Then the data generation process implies, $\forall\, T \geq 1$,

$Y(T) = \hat{Y}(T-1)$,
$\hat{Y}(T) = p(z_1 = q | z_0 \neq q) \cdot Y(T-1) + p(z_1 \in [N] \setminus \{q\} | z_0 \neq q) \cdot \hat{Y}(T-1) + p(z_1 = k | z_0 \neq q)$.

The iteration becomes

$$Y(T) = \hat{Y}(T-1) + (1-\alpha),$$
$$\hat{Y}(T) = \frac{1}{N} \cdot Y(T-1) + \frac{N-1}{N} \cdot \hat{Y}(T-1) + \frac{1}{N}.$$

Note that these two equations are exactly the same as those in Lemma E.4 with same initialization as $Y(0) = \hat{Y}(0) = 0$. Therefore, we have

$$Y(T) \approx \frac{T}{N},$$
$$\hat{Y}(T) \approx \frac{T}{N}.$$

Since the data generation process implicitly assumes $z_0 \neq q$, we have the desired expectation as

$$\mathbb{E}\left[\sum_{t \leq T} \mathbb{1}\{z_t = k\} \middle| \bar{y} \neq q, k = q\right] = \hat{Y}(T) \approx \frac{T}{N}.$$

To obtain the expectation of the quadratic term, we similarly denote the following terms with different $z_0$:

$$Z(T) \triangleq \mathbb{E}\left[\left(\sum_{t \leq T} \mathbb{1}\{z_t = k\}\right)^2 \middle| z_0 = q\right],$$

$$\hat{Z}(T) \triangleq \mathbb{E}\left[\left(\sum_{t \leq T} \mathbb{1}\{z_t = k\}\right)^2 \middle| z_0 \in [N+1], z_0 \neq q\right].$$

Then the data generation process implies, $\forall\, T \geq 1$,

$$Z(T) = \hat{Z}(T-1),$$
$$\hat{Z}(T) = p(z_1 = q|z_0 \neq q) \cdot Z(T-1) + p(z_1 \in [N] \setminus \{q\}|z_0 \neq q) \cdot \hat{Z}(T-1)$$
$$+ p(z_1 = \bar{k}|z_0 \neq q) \cdot (1 + 2\hat{Y}(T-1)),$$

where $2\hat{Y}(T-1)$ is due to $\mathbb{E}[(1 + \sum_{2 \leq t \leq T} \cdot)^2] = 1 + 2\mathbb{E}[\sum_{2 \leq t \leq T} \cdot] + \mathbb{E}[(\sum_{2 \leq t \leq T} \cdot)^2]$.

Then the iteration becomes

$$Z(T) = \hat{Z}(T-1),$$
$$\hat{Z}(T) = \frac{1}{N} Z(T-1) + \frac{N-1}{N} \hat{Z}(T-1) + \frac{1}{N}(1 + 2\hat{Y}(T-1)).$$

Again note that, since $Y(T) \approx \hat{Y}(T)$, these two equations are the same as those in Lemma E.4. Therefore, we have

$$Z(T) \approx \frac{T}{N} + \frac{T^2}{N^2},$$
$$\hat{Z}(T) \approx \frac{T}{N} + \frac{T^2}{N^2}.$$

Since the data generation process implicitly assumes $z_0 \neq q$, we have the desired expectation as

$$\mathbb{E}\left[\left(\sum_{t \leq T} \mathbb{1}\{z_t = k\}\right)^2 \middle| \bar{y} = q, k \in [N] \setminus \{q\}\right] = \hat{Z}(T) \approx \frac{T}{N} + \frac{T^2}{N^2}.$$

$\square$

# F  PROOF OF THEOREM 2: TRAINING DYNAMICS OF THE ATTENTION LAYER

We consider the following simplified 1-layer model for the noisy in-context recall task.

$$x_t \triangleq \mathbf{W}_E(z_t) + \widetilde{\mathbf{W}}_E(z_{t-1}) \in \mathbb{R}^d,$$

$$\phi(x_T, x_{1:T}) \triangleq \sum_{t \leq T} \left[ \sigma\left( x_T^\top \mathbf{W}_{QK} x_{1:T} \right) \right]_t \cdot \mathbf{W}_V x_t \in \mathbb{R}^d,$$

$$\xi_{\text{attn}}(x_{1:T}) \triangleq \mathbf{W}_U \phi(x_T, x_{1:T}) \in \mathbb{R}^{N+1}, \tag{23}$$

$$\xi_{\text{ff}}(x_{1:T}) \triangleq \mathbf{W}_U F(x_T) = \mathbf{W}_U \mathbf{W}_F x_T \in \mathbb{R}^{N+1},$$

With zero initialization of $\mathbf{W}_{QK}, \mathbf{W}_V, \mathbf{W}_F$, we analyze the training dynamics of these three matrices in three phases:

1. $\mathbf{W}_F$ learns the noise association in $O(\frac{1}{\eta})$ time,

2. $\mathbf{W}_V$ learns to be identity for all tokens $k \in [N+1]$,

3. $\mathbf{W}_{QK}$ attends to any position $t$ such that $z_{t-1} = q$ and $z_t = \bar{y}$.

**Assumption F.1.** *In this section, we make the following assumptions*

1. *(orthonormal embedding)* $\mathbf{W}_E(i)^\top \mathbf{W}_E(j) = \widetilde{\mathbf{W}}_E(i)^\top \widetilde{\mathbf{W}}_E(j) = \mathbb{1}\{i = j\}$ *and* $\mathbf{W}_E(i)^\top \widetilde{\mathbf{W}}_E(j) = 0$ *for any* $i, j \in [N+1]$.

2. *(Feed-forward learns noise association) After phase 1, the prediction for noise always satisfies* $\hat{p}(N+1|z_{1:T}) = \alpha$ *for any* $z_{1:T} \in [N+1]^{\otimes T}$. *If* $\hat{p}$ *deviates from* $\alpha$, $\mathbf{W}_F$ *will learn the noise association in a more quick speed than the other weights, so that it is fair to assume* $\hat{p} = \alpha$ *for computing gradients of these weights.*

3. *(Infinite samples)* $m \to \infty$ *so the training loss $L$ is population loss.*

4. $\alpha \leq 1.5 - \sqrt{5}/2 \approx 0.38$. *This is to ensure the sign* $\mathbf{W}_U(j)^\top (-\nabla_{\mathbf{W}_V} L) \mathbf{W}_E(k) > 0$ *for any* $j = k \leq N$ *in* (25).

**Phase 1**: In this phase, the impact of $\widetilde{\mathbf{W}}_E(z_{T-1})$ on $\mathbf{W}_F$ and $\mathbf{W}_V$ is negligible compared with that of $\mathbf{W}_E(z_T)$ because $Z_{T-1}$ is close to uniform in $[N+1]$ while $z_T = q$ is fixed.

Lemma D.1 gives

$$\mathbf{W}_U(k)^\top (-\nabla_{\mathbf{W}_F} L) \mathbf{W}_E(q) = \begin{cases} \Theta(1), & \text{if } k = N+1, \\ \Theta(\frac{1}{N}), & \text{if } k \leq N. \end{cases}$$

Lemma D.2 gives

$$\mathbf{W}_U(j)^\top (-\nabla_{\mathbf{W}_V} L) \mathbf{W}_E(k) = \begin{cases} \Theta(\frac{1}{N}), & \text{if } j = N+1, \forall\, k, \\ \Theta(\frac{1}{N^2}), & \text{if } j \leq N, \forall\, k. \end{cases} \tag{24}$$

Note that the entries of the above projection have the following signs, with details as $-\mu$ in Table 3,

$$\mathbf{W}_U(j)^\top (-\nabla_{\mathbf{W}_V} L) \mathbf{W}_E(k) \begin{cases} > 0, & \text{if } (j = N+1) \text{ or } (j = k) \text{ or } (j = q, k = N+1), \\ < 0, & \text{otherwise.} \end{cases} \tag{25}$$

The arguments in Appendix B.3 show

$$\mathbf{W}_E(j)^\top (-\nabla_{\mathbf{W}_{QK}} L) \mathbf{W}_E(q) = \begin{cases} -\Theta(\frac{1}{N^2}), & \text{if } j = N+1, \\ \Theta(\frac{1}{N^3}), & \text{if } j \leq N. \end{cases} \tag{26}$$

Therefore, during this phase, $\mathbf{W}_F$ learns the noise association with effective graident norm of $\Theta(1)$ as $\mathbf{W}_U(N+1)^\top (-\nabla_{\mathbf{W}_F} L) \mathbf{W}_E(q) = \Theta(1)$. Meanwhile, $\mathbf{W}_F$ moves in the other directions

uniformly in $\Theta(\frac{1}{N})$ as $\mathbf{W}_U(k)^\top(-\nabla_{\mathbf{W}_F}L)\mathbf{W}_E(q) = \Theta(\frac{1}{N})$ for any $k \le N$, which in fact ensures $\hat{p}(k|z_{1:T}) = \frac{1-\hat{p}(N+1|z_{1:T})}{N}$ for any $k \le N$ and $z_{1:T} \in [N+1]^{\otimes T}$.

After $O(\eta^{-1})$ steps in this phase, we have $\hat{p}(N+1|z_{1:T}) = \alpha$ and $\hat{p}(k|z_{1:T}) = \frac{1-\alpha}{N}$ for any $k \le N$ and $z_{1:T}$.

**Phase 2**: Assume $\hat{p}(N+1|\cdot) = \alpha$ starting from the beginning of this phase as discussed above. Due to symmetry for the rest $k$ channels, we have $\hat{p}(k|\cdot) = \frac{1-\alpha}{N}$. Note that the attention scores in $\phi(\cdot,\cdot)$ are still close to uniform, *i.e.*, $\left[\sigma\left(x_T^\top \mathbf{W}_{QK} x_{1:T}\right)\right]_t \approx \frac{1}{T}$, since the update of $\mathbf{W}_{QK}$ is in $O(N^{-2})$ whose impact on attention scores is also in $O(N^{-2})$ through $\exp(x) \approx 1 + x$ for $x \approx 0$. Then we track the movement of $\mathbf{W}_V$ under these conditions.

Since $m \to \infty$, taking $\bar{x} \triangleq \frac{1}{T}\sum_{i=1}^T x_i$, $\mu_k \triangleq \mathbb{E}[\bar{x}|y=k]$ and $\hat{\mu}_k \triangleq \mathbb{E}[\frac{\hat{p}(k|x)}{p(y=k)}\bar{x}] = \mathbb{E}[\bar{x}]$ since $\hat{p}(k|x) = \alpha\mathbb{1}\{k=N+1\} + \frac{1-\alpha}{N}\mathbb{1}\{k \le N\} = p(y|k)$, Lemma H.1 gives

$$
\begin{aligned}
\nabla_{\mathbf{W}_V}L &= \sum_{k=1}^{N+1} p(y=k)\mathbf{W}_U(k)(\mathbb{E}[\bar{x}] - \mathbb{E}[\bar{x}|y=k])^\top \\
&= \sum_{k=1}^{N} p(y=k)\mathbf{W}_U(k)(\mathbb{E}[\bar{x}] - \mathbb{E}[\bar{x}|y=k])^\top \\
&= \sum_{k=1}^{N} \frac{1-\alpha}{N}\mathbf{W}_U(k)(\mathbb{E}[\bar{x}] - \mathbb{E}[\bar{x}|y=k])^\top \\
&= -\frac{1-\alpha}{N^2}\sum_{k=1}^{N}\mathbf{W}_U(k)(\mathbf{W}_E(k) - \overline{\mathbf{W}}_E + \widetilde{\mathbf{W}}_E(k) - \overline{\widetilde{\mathbf{W}}}_E)^\top,
\end{aligned}
$$

where the second equality is due to $\mathbb{E}[\bar{x}] = \mathbb{E}[\bar{x}|y=N+1]$ due to $y = N+1$ is uniform for any correct token $\bar{y} \le N$, and the last equality is from

$$
\mathbb{E}[\bar{x}] - \mathbb{E}[\bar{x}|y=k] \approx -\frac{1}{N}(\mathbf{W}_E(k) - \overline{\mathbf{W}}_E) - \frac{1}{N}(\widetilde{\mathbf{W}}_E(k) - \overline{\widetilde{\mathbf{W}}}_E)
$$

with $\overline{\mathbf{W}}_E = N^{-1}\sum_{i=1}^N \mathbf{W}_E(i)$, $\overline{\widetilde{\mathbf{W}}}_E = N^{-1}\sum_{i=1}^N \widetilde{\mathbf{W}}_E(i)$ because $\mathbb{E}[\bar{x}] = \mathbb{E}_y[\mathbb{E}_x[\bar{x}|y]]$, and the expected number of the tuple $(q, \hat{y})$ in a context length $T$ is $\Theta(\frac{T}{N})$ by comparing Lemma E.6 and E.7.

Therefore, the gradient for $\mathbf{W}_V$ has the following structure

$$
\begin{aligned}
\mathbf{W}_U(j)^\top(-\nabla_{\mathbf{W}_V}L)\mathbf{W}_E(k) &\approx \frac{1}{N^2}\mathbb{1}\{j=k\} + O\left(\frac{1}{N^3}\right), \forall j,k \le N, \\
\mathbf{W}_U(j)^\top(-\nabla_{\mathbf{W}_V}L)\widetilde{\mathbf{W}}_E(k) &\approx \frac{1}{N^2}\mathbb{1}\{j=k\} + O\left(\frac{1}{N^3}\right), \forall j,k \le N.
\end{aligned}
\tag{27}
$$

Denote steps of phase 1 and phase 2 as $t_1$ and $t_2$. Combined with the structure of $\mathbf{W}_V$ in phase 1 as in Eq.(24,25), ignoring projections that are $O(N^{-3})$ or negative, $\mathbf{W}_V$ has the following structure after phase 2

$$
\mathbf{W}_U(j)^\top\mathbf{W}_V\mathbf{W}_E(k) = \begin{cases} \Theta(\eta t_1 N^{-1}), & \text{if } j = N+1, \forall k, \\ \Theta(\eta t_1 N^{-2} + \eta t_2 N^{-2}), & \text{if } j = k \le N, \\ \Theta(\eta t_1 N^{-2}), & \text{if } j = q, k = N+1, \end{cases}
\tag{28}
$$

$$
\mathbf{W}_U(j)^\top\mathbf{W}_V\widetilde{\mathbf{W}}_E(k) = \Theta(\eta t_2 N^{-2}), \text{if } j = k \le N.
$$

**Phase 3**: now assume $\mathbf{W}_V$ has the structure in Eq(28). The model still predicts $\hat{p}_{\mathbf{W}}(k|z) = \alpha\mathbb{1}\{k = N+1\} + \frac{1-\alpha}{N}\mathbb{1}\{k \le N\}$ because the above projections of $\mathbf{W}_V$ onto $\mathbf{W}_U(j : j \le N)$ is $o(\frac{1}{N})$.

Meanwhile, the attention scores are uniform as $\frac{1}{T}$ as $\mathbf{W}_{QK} \approx 0$. Therefore, the gradient of $\mathbf{W}_{QK}$ is

$$\nabla_{\mathbf{W}_{QK}} L = \frac{1}{T} \sum_{k=1}^{N+1} \sum_{t \leq T} p(y = k)(\mathbb{E}[(\mathbf{W}_U(k)^\top \mathbf{W}_V x_t) \cdot x_T (x_t - \bar{x})^\top]$$

$$- \mathbb{E}[(\mathbf{W}_U(k)^\top \mathbf{W}_V x_t) \cdot x_T (x_t - \bar{x})^\top | y = k])$$

$$= \frac{1-\alpha}{TN} \sum_{k=1}^{N} \sum_{t \leq T} (\mathbb{E}[(\mathbf{W}_U(k)^\top \mathbf{W}_V x_t) \cdot x_T (x_t - \bar{x})^\top]$$

$$- \mathbb{E}[(\mathbf{W}_U(k)^\top \mathbf{W}_V x_t) \cdot x_T (x_t - \bar{x})^\top | y = k]),$$

where $\bar{x} = T^{-1} \sum_{t \leq T} x_t$ and the last equality holds due to the condition of $y = N + 1$ uniform for any correct token $\hat{y} \leq N$. Then, considering the above structure of $\mathbf{W}_V$, we notice that $\mathbf{W}_U(j)^\top \mathbf{W}_V x_t \approx \beta_1 \mathbb{1}\{z_t = j\} + \beta_2 \mathbb{1}\{z_{t-1} = j\}$ with $\beta_1 = \eta t_1 N^{-2} + \eta t_2 N^{-2}$ and $\beta_2 = \eta t_2 N^{-2}$ for any $j, k \leq N$. Here note that we ignore the projection of $j = q, k = N + 1$ in Eq(28) because $\hat{y} = q$ is with probability $1/N = o(1)$ so that it will not influence much the following derivation.

Plug-in $\mathbf{W}_U(j)^\top \mathbf{W}_V x_t$ and we get

$$\mathbf{W}_E(q)^\top(-\nabla_{\mathbf{W}_{QK}} L)(\mathbf{W}_E(b_1) + \widetilde{\mathbf{W}}_E(b_2)) = \frac{1-\alpha}{TN} \sum_{k \leq N} \sum_{t \leq T} \mathbb{E}[A_{k,b_1,b_2}^{(t)} | y = k] - \mathbb{E}[A_{k,b_1,b_2}^{(t)}]$$

(29)

where

$$A_{k,b_1,b_2}^{(t)} = (\beta_1 \mathbb{1}\{z_t = k\} + \beta_2 \mathbb{1}\{z_{t-1} = k\})$$
$$\cdot \left( \mathbb{1}\{z_t = b_1\} - \frac{\sum_{s \leq T} \mathbb{1}\{z_s = b_1\}}{T} + \mathbb{1}\{z_{t-1} = b_2\} - \frac{\sum_{s \leq T} \mathbb{1}\{z_{s-1} = b_2\}}{T} \right).$$

Now we are to control $\Delta_{k,b_1,b_2} \triangleq \sum_{t \leq T} \mathbb{E}[A_{k,b_1,b_2}^{(t)} | y = k] - \mathbb{E}[A_{k,b_1,b_2}^{(t)}]$ for different choices of $b_1, b_2$. Note that $b_1$ and $b_2$ co-exist by sum in $A_{k,b_1,b_2}^{(t)}$, so the additivity of expectation allows us to discuss choices of $b_1, b_2$ separately and then combine the results. Denote

$$B_{k,b_1}^{(t)} = (\beta_1 \mathbb{1}\{z_t = k\} + \beta_2 \mathbb{1}\{z_{t-1} = k\}) \left( \mathbb{1}\{z_t = b_1\} - \frac{\sum_{s \leq T} \mathbb{1}\{z_s = b_1\}}{T} \right),$$
$$C_{k,b_2}^{(t)} = (\beta_1 \mathbb{1}\{z_t = k\} + \beta_2 \mathbb{1}\{z_{t-1} = k\}) \left( \mathbb{1}\{z_{t-1} = b_2\} - \frac{\sum_{s \leq T} \mathbb{1}\{z_{s-1} = b_2\}}{T} \right).$$

(30)

Controlling $\sum_{t \leq T} \mathbb{E}[B_{k,b_1}^{(t)} | y = k] - \mathbb{E}[B_{k,b_1}^{(t)}]$:

- If $b_1 = k$, from Lemma E.6 and E.7, we have

$$\mathbb{E} \left[ \sum_{t \leq T} \beta_1 \mathbb{1}\{z_t = k\} \mathbb{1}\{z_t = k\} \middle| y = k \right] - \mathbb{E} \left[ \sum_{t \leq T} \beta_1 \mathbb{1}\{z_t = k\} \mathbb{1}\{z_t = k\} \right] = \beta_1 (1-\alpha) \frac{T}{N}.$$

$$\mathbb{E} \left[ -\sum_{t \leq T} \beta_1 \mathbb{1}\{z_t = k\} \frac{\sum_{s \leq T} \mathbb{1}\{z_s = k\}}{T} \middle| y = k \right] - \mathbb{E} \left[ -\sum_{t \leq T} \beta_1 \mathbb{1}\{z_t = k\} \frac{\sum_{s \leq T} \mathbb{1}\{z_s = k\}}{T} \right]$$

$$= -\mathbb{E} \left[ \beta_1 T^{-1} (\sum_{s \leq T} \mathbb{1}\{z_s = k\})^2 | y = k \right] + \mathbb{E} \left[ \beta_1 T^{-1} (\sum_{s \leq T} \mathbb{1}\{z_s = k\})^2 \right]$$

$$= \beta_1 T^{-1} \left( \frac{T}{N} + \frac{T^2}{N^2} - \frac{(2-\alpha)T}{N} - \frac{(2-\alpha)^2 T^2}{N^2} \right) = o \left( \beta_1 \frac{T}{N} \right).$$

The terms involving $\mathbb{1}\{z_{t-1} = k\}$ are negligible as $O(T/N^2)$. Therefore, we have

$$\sum_{t \leq T} \mathbb{E}[B_{k,k}^{(t)} | y = k] - \mathbb{E}[B_{k,k}^{(t)}] = \beta_1 (1-\alpha) \frac{T}{N}.$$

(31)

- If $b_1 \neq k$, all terms are $O(T/N^2)$ because
  - If $b_1 \leq N$, it holds $p(z_t = b_1 | z_{t-1} = k) = 1/N$ with the expected number of $k$ in context of length $L$ being $\Theta(T/N)$ from lemmas in Appendix E.
  - If $b_1 = N + 1$, it holds $p(z_t = N + 1 | z_{t-1} = k) = O(1/N) \cdot \mathbb{1}\{k = q\}$ and the expected number of $q$ in context of length $T$ is $\Theta(T/N)$ from Lemma E.1 and E.4.
  - $\mathbb{E}[\sum_t \mathbb{1}\{z_{t-1} = k\} \#b_1/T | \cdot] = \mathbb{E}[\#k \cdot \#b_1/T] = O(T/N^2)$ no matter it is with condition $y = k$ or not.

  Therefore, for any $b_1 \neq k$, we have

  $$\sum_{t \leq T} \mathbb{E}[B_{k,b_1}^{(t)} | y = k] - \mathbb{E}[B_{k,b_1}^{(t)}] = o(T/N). \tag{32}$$

Controlling $\sum_{t \leq T} \mathbb{E}[C_{k,b_2}^{(t)} | y = k] - \mathbb{E}[C_{k,b_2}^{(t)}]$:

- If $b_2 = q$, Lemma E.4 gives

$$\mathbb{E}\left[\sum_{t \leq T} \beta_1 \mathbb{1}\{z_t = k\} \left(\mathbb{1}\{z_{t-1} = q\} - \frac{\#q}{T}\right) \Big| y = k\right]$$

$$- \mathbb{E}\left[\sum_{t \leq T} \beta_1 \mathbb{1}\{z_t = k\} \left(\mathbb{1}\{z_{t-1} = q\} - \frac{\#q}{T}\right)\right]$$

$$= (1 - p(\hat{y} = k)) \cdot \mathbb{E}\left[\sum_{t \leq T} \beta_1 \mathbb{1}\{z_t = k\} \left(\mathbb{1}\{z_{t-1} = q\} - \frac{\#q}{T}\right) \Big| y = k\right] + o\left(\beta_1 \frac{T}{N}\right)$$

$$\approx \beta_1(1 - \alpha)\frac{T}{N},$$

where the last equality is from $p(z_t = k | \bar{y} = k, z_{t-1} = q) = 1 - \alpha$.
All the other terms are negligible with the same reason as above.
Therefore, we have

$$\sum_{t \leq T} \mathbb{E}[C_{k,q}^{(t)} | y = k] - \mathbb{E}[C_{k,q}^{(t)}] = \beta_1(1 - \alpha)\frac{T}{N}. \tag{33}$$

- If $b_2 = k$, similar to the above discussion about $B_{k,k}$, we have

$$\sum_{t \leq T} \mathbb{E}[C_{k,k}^{(t)} | y = k] - \mathbb{E}[C_{k,k}^{(t)}] = \beta_2(1 - \alpha)\frac{T}{N}. \tag{34}$$

  Note that the key difference is that here we use $\beta_2$ instead of $\beta_1$, and $\beta_2 < \beta_1$.

- If $b_2 \neq q$ and $b_2 \neq k$, similar to the discussion for Eq(32), we have

$$\sum_{t \leq T} \mathbb{E}[C_{k,b_2}^{(t)} | y = k] - \mathbb{E}[C_{k,b_2}^{(t)}] = o(T/N). \tag{35}$$

Therefore, combining the above results in Eq(31, 32, 33, 34, 35), taking sums of the corresponding $B$ and $C$ from Eq(30) gives

$$\Delta_{k,b_1,b_2} = \begin{cases} \beta_1(1-\alpha)TN^{-1} + \beta_1(1-\alpha)TN^{-1}, & \text{if } b_1 = k, b_2 = q, \\ \beta_1(1-\alpha)TN^{-1} + \beta_2(1-\alpha)TN^{-1}, & \text{if } b_1 = k, b_2 = k, \\ \beta_1(1-\alpha)TN^{-1}, & \text{if } b_1 = k, \text{other } b_2, \\ \beta_1(1-\alpha)TN^{-1}, & \text{if } b_1 \neq k, b_2 = q, \\ \beta_2(1-\alpha)TN^{-1}, & \text{if } b_1 \neq k, b_2 = k, \\ O(TN^{-1}), & \text{otherwise.} \end{cases}$$

To take the summation over all $k \leq N$ in Eq(29), we discuss the following cases of $b_1$ and $b_2$ for $\mathbf{W}_E(q)^\top(-\nabla_{\mathbf{W}_{QK}}L)(\mathbf{W}_E(b_1) + \widetilde{\mathbf{W}}_E(b_2))$.

- If $b_1 \leq N, b_1 \neq b_2, b_2 = q$:
  - when $k = b_1$, we take $\Delta_{k,b_1,b_2}$ under the condition of $b_1 = k, b_2 = q$.
  - when $k \neq b_1$, we take $\Delta_{k,b_1,b_2}$ under the condition of $b_1 \neq k, b_2 = q$. Note that there are $(N-1)$ such $k$.

  Therefore, it holds

  $$\mathbf{W}_E(q)^\top(-\nabla_{\mathbf{W}_{QK}}L)(\mathbf{W}_E(b_1) + \widetilde{\mathbf{W}}_E(q)) = \frac{1-\alpha}{TN}\beta_1(1-\alpha)T(1+N^{-1}). \quad (36)$$

- If $b_1 = b_2 = q$:
  - when $k = b_1$, we take $\Delta_{k,b_1,b_2}$ under the condition of $b_1 = k, b_2 = k$ to achieve a lower bound of the gap later.
  - when $k \neq b_1$, we take $\Delta_{k,b_1,b_2}$ under the condition of $b_1 \neq k, b_2 = q$. Note that there are $(N-1)$ such $k$.

  Therefore, it holds

  $$\mathbf{W}_E(q)^\top(-\nabla_{\mathbf{W}_{QK}}L)(\mathbf{W}_E(b_1) + \widetilde{\mathbf{W}}_E(q)) \geq \frac{1-\alpha}{TN}\left(\beta_1(1-\alpha)T + \beta_2(1-\alpha)TN^{-1}\right). \quad (37)$$

- If $b_1 = N+1, b_2 = q$: for any $k \leq N$, it holds $k \neq b_1$, so we take $\Delta_{k,b_1,b_2}$ under the condition of $b_1 \neq k, b_2 = q$. Therefore, it holds

  $$\mathbf{W}_E(q)^\top(-\nabla_{\mathbf{W}_{QK}}L)(\mathbf{W}_E(N+1) + \widetilde{\mathbf{W}}_E(q)) = \frac{1-\alpha}{TN}\beta_1(1-\alpha)T. \quad (38)$$

- If $b_2 \neq q, \forall b_1$: To get an upper bound of the projection length, we take $\Delta_{k,b_1,b_2}$ under the condition of $b_= k, b_2 = k$ or $b_1 \neq k, b_2 = k$. Therefore, it holds

  $$\mathbf{W}_E(q)^\top(-\nabla_{\mathbf{W}_{QK}}L)(\mathbf{W}_E(b_1) + \widetilde{\mathbf{W}}_E(b_2)) \leq \frac{1-\alpha}{TN}(\beta_1 + 2\beta_2)(1-\alpha)TN^{-1}. \quad (39)$$

Comparing the above four cases, for any $\bar{y} \leq N$, the attention weight $\mathbf{W}_{QK}$ to attend more to $x_t = \mathbf{W}_E(\bar{y}) + \widetilde{\mathbf{W}}_E(q)$ than to $x_t = \mathbf{W}_E(N+1) + \widetilde{\mathbf{W}}_E(q)$, with

$$\mathbf{W}_E(q)^\top(-\nabla_{\mathbf{W}_{QK}}L)(\mathbf{W}_E(\bar{y}) + \widetilde{\mathbf{W}}_E(q)) - \mathbf{W}_E(q)^\top(-\nabla_{\mathbf{W}_{QK}}L)(\mathbf{W}_E(N+1) + \widetilde{\mathbf{W}}_E(q))$$
$$\geq \frac{(1-\alpha)^2}{N^2}\beta_2.$$

Meanwhile, any other setting of $b_1, b_2$ has smaller projection in $(-\nabla_{\mathbf{W}_{QK}}L)$.

In summary, $\mathbf{W}_{QK}$ has the following patterns

1. it learns to attend to indices $t$ such that $z_{t-1} = q$ is the trigger word,

2. when there are multiple $t_i$'s such that $z_{t_i-1=q}$, it learns to attend to those with $z_t = \bar{y}$ more than $z_t = N+1$.

## G    LINEAR ASSOCIATIVE MEMORY

### G.1    EXPERIMENTS AND DISCUSSIONS

In Section 3, we showed that *fully* truncating a feed-forward layer can be helpful for reasoning. We now present a setting where noisy associations are stored in a rank-one subspace of a layer, so that *intermediate* levels of truncation are more useful to remove noise.

**Model and data.** We consider a simple associative memory setting where the goal is learn an fixed permutation from input tokens to output tokens (w.l.o.g. taken to be the identity), with a linear model similar to Cabannes et al. (2024). Consider a learnable weight matrix $\mathbf{W} \in \mathbb{R}^{d \times d}$. Consider embeddings for $n$ input tokens as $\{e_i\}_{i=1}^n \subset \mathbb{R}^d$ and embeddings for $c$ output tokens as $\{u_i\}_{i=1}^c \subset \mathbb{R}^d$. In contrast to Cabannes et al. (2024), we consider an additional "common noise"

output token $c = n + 1$, which is chosen for any input with probability $\alpha \in (0, 1)$. For any input $x \in [n]$, the target distribution $p_\alpha(\cdot|x)$ is defined by

$$p_\alpha(y|x) = (1 - \alpha) \cdot \mathbb{1}\{y = x\} + \alpha \cdot \mathbb{1}\{y = c\}. \tag{40}$$

In other words, the last channel ($c$) for output is the **common noise** with probability $\alpha$ for any input. The training dataset $\mathcal{D}_\alpha$ consists of uniformly distributed inputs $x \in [n]$, and outputs conditionally sampled as $y|x \sim p_\alpha(\cdot|x)$.

Given any pair of input and output tokens, the associative memory model takes the form

$$f(i, j; \mathbf{W}) \triangleq \langle u_j, \mathbf{W}e_i \rangle, \quad \forall\, i, j \in [n] \times [c], \tag{41}$$

When $k \leq d$, we denote the rank-$k$ approximation of $f$ as $f^{(k)}$ by replacing $\mathbf{W}$ with $\mathbf{W}^{(k)}$, where $\mathbf{W}^{(k)}$ is the rank-$k$ approximation of $\mathbf{W}$.

**Training.** During training, the dataset $\mathcal{D}_\alpha$ is generated with non-zero noise probability $\alpha > 0$. At test time, the dataset $\mathcal{D}_0$ is without noise as $\alpha = 0$, so the computed loss is called **pure-label** loss. The model is trained with Gradient Descent (GD) subjected to cross-entropy loss.

**Experiments with randomness.** Assume both $\{e_i\}_{i=1}^n$ and $\{u_i\}_{i=1}^c$ are i.i.d. uniformly drawn from sphere $\mathbb{S}^{d-1}$. Also assume the model is initialized as $\mathbf{W}_{i,j} \sim \mathcal{N}(0, \frac{1}{d})$. Due to randomness from embeddings and model initialization, let's first conduct 20 runs of experiments to obtain significant factors before moving the theoretical argument.

Note that *only full models are trained*, and we track loss for low-rank models by conducting SVD in each step without manipulating training. In Figure 5, we illustrate the pure-label loss *v.s.* training steps for models of different ranks, where $n = 3$, $\alpha = 0.03$ and $d = 8$ or 12. It turns out, while the full model (rank$\geq 3$) has a constant pure-label loss ($\sim 0.03$, dependent on $\alpha$), the rank-2 model is very likely to have a significant loss than the full model. Meanwhile, the larger $d$ has more stable results than small $d$.

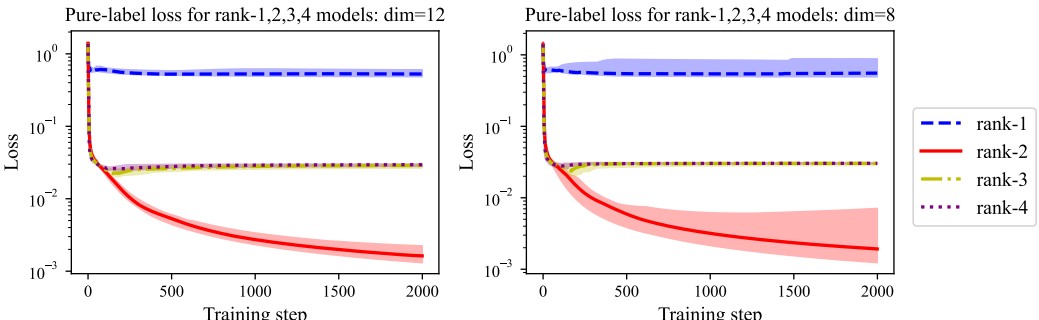

Figure 18: Pure-label loss for rank-1,2,3,4 models with $n = 3, \alpha = 0.03$ and $d = 12$ (left) or 8 (right). *Only full models are trained*, and we report low-rank results by conducting SVD in each step without manipulating the training. In both figures, the experiments are run for 20 times to examine the randomness. For each rank, we plot curves of the median, $25\%$ and $75\%$ out of 20 runs. It turns out: i) rank-2 models are very likely to have significantly lower pure-label loss thant full models (rank$\geq 3$), and ii) the larger dimension $d$ has more stable results.

Therefore, we can qualify the following important factors for this model:

i. $d$ *v.s.* $n, c$: when $d \gg n, c$, random drawn embeddings tend to be orthogonal to each other, with inner product in $O(1/\sqrt{d})$. If $n, c = \Omega(d)$, embeddings will be in strong correlations, making the problem extremely difficult to understand. Cabannes et al. (2024) also discussed about such particle interaction in associative memory.

ii. Low-rank subspace storing the noise. In Figure 18, the rank-1 subspace between the full and rank-2 models is responsible to store the noise, removing which will induce a model ideally predicting the ground-truth without noise. This is understandable if the embeddings are orthogonal, as shown in Theorem 3.

iii. $\alpha$ *v.s.* $n$. When $n$ is large, orthogonal embeddings still induces a low-rank subspace storing the noise, but $\alpha$ decides whether the low-rank subspace corresponds to the smallest singular values of $\mathbf{W}$. If not, it requires more careful manipulation of the spectrum instead of low-rank approximation of $\mathbf{W}$.

## G.2 Proof of Theorem 3

Now we present a theoretical analysis of this problem with some assumptions.

**Assumption G.1** (Orthonormality). *Embeddings of input and output tokens are orthonormal, i.e.,* $e_i^\top e_j = \mathbb{1}\{i = j\}, \forall\, i, j$ *and* $u_i^\top u_j = \mathbb{1}\{i = j\}, \forall\, i, j$.

**Assumption G.2** (Initialization). *The learnable matrix* $\mathbf{W}$ *is initialized from* $\mathbf{0}$ *when* $t = 0$.

**Theorem 5** (Restatement of Theorem 3). *Assume Assumptions G.1 and G.2 hold, considering* $n = 2, c = 3$ *and* $\alpha \in (0.2, 0.4)$, *we train the full model* $f(\cdot, \cdot; \mathbf{W})$ *with gradient flow. Denote* $P(i, j; \mathbf{W})$ *as the model's predicted probability for output* $j$ *conditioned on input* $i$. *Then, for* $t \to \infty$ *and* $i \in \{1, 2\}$, *we have*

$$P(i, j; \mathbf{W}) = (1 - \alpha) \cdot \mathbb{1}\{j = i\} + \alpha \cdot \mathbb{1}\{j = c\},$$

$$P(i, j; \mathbf{W}^{(1)}) = (1 - \Theta(t^{-1/2})) \cdot \mathbb{1}\{j = i\} + \Theta(t^{-1/2}) \cdot \mathbb{1}\{j = c\}.$$

*Remark* 1. Note that here the assumption $\alpha \in (0.2, 0.4)$ is a technical choice. In experiments, any value $\alpha \in (0, 0.4)$ still has the same result.

*Proof.* W.l.o.g., we assume the embeddings are standard basis in $\mathbb{R}^d$. For any $\mathbf{W}$, the gradient $\nabla_{\mathbf{W}} L$ can be decomposed as

$$\nabla_{\mathbf{W}} L = \gamma_1 \begin{bmatrix} 1 \\ -1 \\ 0 \end{bmatrix} \begin{bmatrix} 1 & -1 & 0 \end{bmatrix} + \gamma_2 \begin{bmatrix} 1 \\ 1 \\ -2 \end{bmatrix} \begin{bmatrix} 1 & 1 & 0 \end{bmatrix}. \tag{42}$$

Since $\mathbf{W}$ initializes from zero, this implies $\mathbf{W}$ can always be decomposed with the same basis

$$\mathbf{W} = \beta_1 \begin{bmatrix} 1 \\ -1 \\ 0 \end{bmatrix} \begin{bmatrix} 1 & -1 & 0 \end{bmatrix} + \beta_2 \begin{bmatrix} 1 \\ 1 \\ -2 \end{bmatrix} \begin{bmatrix} 1 & 1 & 0 \end{bmatrix}. \tag{43}$$

Then gradient flow gives the following ODE

$$
\begin{aligned}
\dot{\beta}_1 = -\gamma_1 &= \frac{\exp(-\beta_1 + \beta_2) - \exp(\beta_1 + \beta_2)}{\exp(-\beta_1 + \beta_2) + \exp(\beta_1 + \beta_2) + \exp(-2\beta_2)} + 1 - \alpha \\
&= \frac{\exp(-2\beta_1) - 1}{\exp(-2\beta_1) + \exp(-\beta_1 - 3\beta_2) + 1} + 1 - \alpha, \\
\dot{\beta}_2 = -\gamma_2 &= \frac{3\exp(-2\beta_2)}{\exp(-\beta_1 + \beta_2) + \exp(\beta_1 + \beta_2) + \exp(-2\beta_2)} - 3\alpha \\
&= \frac{3\exp(-\beta_1 - 3\beta_2)}{\exp(-2\beta_1) + \exp(-\beta_1 - 3\beta_2) + 1} - 3\alpha.
\end{aligned}
\tag{44}
$$

Denoting $a = -2\beta_1, b = -\beta_1 - 3\beta_2$, the ODE becomes

$$
\begin{aligned}
\dot{a} &= \frac{2 - 2\exp(a)}{\exp(a) + \exp(b) + 1} - 2 + 2\alpha, \\
\dot{b} &= \frac{2 - 8\exp(b)}{\exp(a) + \exp(b) + 1} - 2 + 10\alpha.
\end{aligned}
\tag{45}
$$

Lemma H.3 gives the solution as, when $t \to \infty$,

$$a \to -\log(t) - \log(1 - \alpha)(4 - 2\alpha), \quad b \to \log \frac{\alpha}{1 - \alpha}.$$

For the full model, taking the scores $\mathbf{W}_{1,:}$ of the first input token as an example, we have $\mathbf{W}_{11} = \beta_1 + \beta_2, \mathbf{W}_{12} = -\beta_1 + \beta_2, \mathbf{W}_{13} = -2\beta_2$, so the margins are

$$\mathbf{W}_{11} - \mathbf{W}_{12} = 2\beta_1 = -a, \mathbf{W}_{11} - \mathbf{W}_{13} = \beta_1 + 3\beta_2 = -b.$$

For the rank-1 model (assuming $\beta_1 > \beta_2$), the margins are

$$\mathbf{W}_{11}^{(1)} - \mathbf{W}_{12}^{(1)} = 2\beta_1, \mathbf{W}_{11}^{(1)} - \mathbf{W}_{13}^{(1)} = \beta_1.$$

The proof finishes by computing softmax on the margins. □

## H  USEFUL LEMMAS

**Lemma H.1.** *Let $p$ be a data distribution on $(x, y) \in \mathbb{R}^d \times [N]$. Consider training data as $m$ i.i.d. samples $\mathcal{D} \triangleq \{(x_i, y_i)\}_{i=1}^m \subset \mathbb{R}^d \times [N+1]$ from p. Consider the following classification problem, with fixed output embeddings $\mathbf{W}_U$:*

$$\hat{L}(\mathbf{W}) = \frac{1}{m} \sum_{i=1}^m [l(y_i, \mathbf{W}_U \mathbf{W} x_i)].$$

*The gradients take the following form: denoting $\hat{p}_{\mathbf{W}}(k|x_i)$ as the current predicted probability of class $k$ in $[N+1]$ classes for input $x_i$,*

$$\nabla_{\mathbf{W}} \hat{L}(\mathbf{W}) = \frac{1}{m} \sum_{i=1}^m \left[ \sum_{k=1}^{N+1} (\hat{p}_{\mathbf{W}}(k|x_i) - \mathbb{1}\{y_i = k\}) \mathbf{W}_U(k) x_i^\top \right].$$

*When $m \to \infty$, the above equation becomes*

$$\nabla_{\mathbf{W}} L(\mathbf{W}) = \sum_{k=1}^{N+1} p(y = k) \mathbf{W}_U(k) (\hat{\mu}_k - \mu_k)^\top,$$

*where $\mu_k \triangleq \mathbb{E}[x|y = k]$ and $\hat{\mu}_k \triangleq \mathbb{E}_x[\frac{\hat{p}_{\mathbf{W}}(k|x)}{p(y=k)} x]$.*

*Remark* 2.  This lemma is from Lemma 2 in Bietti et al. (2023).

*Proof.*  Recall the form of the cross-entropy loss for classification with $K$ classes:

$$l(y, \epsilon) = - \sum_{k=1}^K \mathbb{1}\{y = k\} \log \frac{e^{\xi_k}}{\sum_j e^{\xi_j}}.$$

Its derivatives take the form

$$\frac{\partial l}{\partial \xi_k}(y, \xi) = s(\xi)_k - \mathbb{1}\{y = k\},$$

where $s(\xi)_k = \frac{e^{\xi_k}}{\sum_j e^{\xi_j}}$.

The gradient of $L$ is then given by

$$\nabla_{\mathbf{W}} \hat{L}(\mathbf{W}) = \frac{1}{m} \sum_{i=1}^m \left[ \sum_{k=1}^{N+1} \frac{\partial l}{\partial \xi_k}(y_i, \mathbf{W}_U \mathbf{W} x_i) \nabla_{\mathbf{W}}(\mathbf{W}_U(k)^\top \mathbf{W} x_i) \right]$$

$$= \frac{1}{m} \sum_{i=1}^m \left[ \sum_{k=1}^{N+1} (\hat{p}_{\mathbf{W}}(k|x_i) - \mathbb{1}\{y_i = k\}) \mathbf{W}_U(k) x_i^\top \right].$$

When $m \to \infty$, the above equation becomes

$$\nabla_{\mathbf{W}} L(\mathbf{W}) = \sum_{k=1}^{N+1} \mathbf{W}_U(k) \mathbb{E}[\hat{p}_{\mathbf{W}}(k|x) x^\top] - \sum_{k=1}^{N+1} \mathbb{E}[\mathbb{1}\{y = k\} \mathbf{W}_U(k) \mathbb{E}[x|y]^\top]$$

$$= \sum_{k=1}^{N+1} \mathbf{W}_U(k) \mathbb{E}[\hat{p}_{\mathbf{W}}(k|x) x^\top] - \sum_{j,k} p(y = k) \mathbb{1}\{j = k\} \mathbf{W}_U(k) \mathbb{E}[x|y = j]^\top$$

$$= \sum_{k=1}^{N+1} p(y = k) \mathbf{W}_U(k) (\hat{\mu}_k - \mu_k)^\top.$$

$\square$

**Lemma H.2.** *Consider a sequence $\{S_t\}_{t \geq 1}$ with $S_t = a^t \cdot t$ where $a \neq 1$. Then $\sum_{1 \leq t \leq T} S_t = \frac{a(1-a^T)}{(a-1)^2} + \frac{a^{T+1} \cdot T}{a-1}$.*

*Proof.* Denote $X_t \triangleq \sum_{1 \leq t \leq T} S_t$. Then we have $a \cdot X_t = \sum_{2 \leq t \leq T+1} a^t \cdot (t-1)$. Hence, it holds $(a-1)X_t = -\sum_{2 \leq t \leq T} a^t - a + a^{T+1} \cdot T = -\frac{a(1-a^T)}{1-a} + a^{T+1} \cdot T$. Therefore, we have

$$X_t = \frac{a(1-a^T)}{(a-1)^2} + \frac{a^{T+1} \cdot T}{a-1}.$$

$\square$

**Lemma H.3.** *Consider the following ODE with with $a(0) = b(0) = 0$ and $\alpha \in (0.2, 0.4)$,*

$$\dot{a} = \frac{2 - 2\exp(a)}{\exp(a) + \exp(b) + 1} - 2 + 2\alpha,$$

$$\dot{b} = \frac{2 - 8\exp(b)}{\exp(a) + \exp(b) + 1} - 2 + 10\alpha.$$

*Then, when $t \to \infty$, we have*

$$a \to -\log(t) - \log(1-\alpha)(4-2\alpha), \quad b \to \log\frac{\alpha}{1-\alpha}.$$

*Proof.* The ODE can be re-written as

$$\dot{a} = 2 \cdot \frac{(\alpha - 2)\exp(a) + (\alpha - 1)\exp(b) + \alpha}{\exp(a) + \exp(b) + 1} \triangleq \frac{2D}{\exp(a) + \exp(b) + 1},$$

$$\dot{b} = 10 \cdot \frac{(\alpha - \frac{1}{5})\exp(a) + (\alpha - 1)\exp(b) + \alpha}{\exp(a) + \exp(b) + 1} \triangleq \frac{10E}{\exp(a) + \exp(b) + 1}.$$

At $t = 0$, it holds $\dot{a}(0) < 0, \dot{b}(0) < 0$ since $D = 3\alpha - 3 < 0, E = 3\alpha - \frac{6}{5} < 0$. Hence, $a$ and $b$ start to decrease from $t = 0$. The ending of the decreasing happens when one of $D$ and $E$ gets positive. Let's show $D$ and $E$ will never be positive when $\alpha \in (0.2, 0.4)$ by contradiction.

Assume time $T_1$ is when one of $E$ and $E$ equals to 0 for the first time. This means $E = 0$, because, for any time $t$, it always holds $D < E$ since $\exp(a) > 0$ for any $a \in \mathbb{R}$. Then at $T_1$, we have $\dot{a} < 0, \dot{b} = 0$, which means $\exp(a)$ will decrease for any small time window $\Delta t > 0$ and $\exp(b)$ stays unchanged. Together with $\alpha > 0.2$, this means it has $E < 0$ again at time $T_1 + \Delta t$. Therefore, it is possible for $E$ to be 0, but $E$ will never be positive. Meanwhile, this also guarantees $D$ will always be negative because $D < E$.

Then, we make an observation that when $D$ is always negative and $E$ is always non-positive, the decreasing nature of $a$ will have $D \approx E$ when $t \to \infty$ by $\exp(a) \approx 0$. This implies $b = \log\frac{\alpha}{1-\alpha}$. Then, by taking $\exp(a) = \beta \cdot t^{-\gamma}$, the ODE gives

$$-\gamma\frac{1}{t} = \frac{(2\alpha - 4)\beta \cdot t^{-\gamma}}{\beta \cdot t^{-\gamma} + \frac{1}{1-\alpha}},$$

which gives $\gamma = 1, \beta = \frac{1}{(1-\alpha)(4-2\alpha)}$.

Therefore, when $t \to \infty$, we have

$$a \to \log\left(\frac{1}{(1-\alpha)(4-2\alpha)}t^{-1}\right), \quad b \to \log\frac{\alpha}{1-\alpha}.$$

$\square$

# I  INPUT EXAMPLES FOR LLMS

## I.1  EXAMPLES FOR PREPOSITIONS

For experiments in Appendix C.1, we use two synthetic datasets: inputs are 30 prepositions, and inputs are 40 incomplete sentences ending with a preposition.

The 30 prepositions are:

"about", "above", "across", "after", "against", "along", "around", "at", "before", "behind", "below", "beneath", "beside", "between", "by", "during", "for", "from", "in", "inside", "into", "near", "of", "on", "over", "through", "to", "under", "with", "without".

Generated by Claude 3 (Anthropic, 2024), the 40 incomplete sentences are:

[ "Inspired painter gazed at pristine canvas, envisioning next creation about", "Children's delighted squeals filled yard as they frolicked, stumbling across", "Singer inhaled deeply, calming nerves before gracing stage before", "Ominous storm clouds amassed, promising downpour that would soon roll in", "Awestruck trekker admired breathtaking summit vista, looking over", "Rich aroma of freshly roasted beans permeated cozy cafe, enticing during", "With deft sleight of hand, illusionist made coin vanish, leaving spectators in awe without", "Majestic oak stood tall, branches reaching skyward above", "Gentle waves caressed shoreline, soothing rhythm lulling along", "Meticulous investigator scoured crime scene, searching for any evidence left behind", "Radiant sunbeams filtered through sheer curtains, warming hardwood floor beneath", "Concert pianist's nimble fingers glided across ivory keys, room resonating with melody around", "Crickets' evening chorus filled silent field from nearby meadow during", "Jubilant laughter resounded down corridor as jovial group headed towards celebration without", "Struggling poet tapped pen restlessly, seeking words to capture elusive emotion beneath", "Soothing patter of raindrops danced on windowpane, inviting serene relaxation with", "Mouthwatering scent of fresh bread beckoned passersby into cozy bakery without", "Mighty waves thundered against jagged cliffs, echoing roar along rugged shoreline around", "Seasoned trekker carefully navigated winding trail, cautiously avoiding exposed roots and rocks beneath", "Graceful ballerina flowed across stage, movements blending seamlessly with melody during", "Crackling campfire cast dancing shadows across gathered faces around", "Vibrant brush strokes danced across canvas, bold hues bursting into life before", "Photographer framed breathtaking sunset, capturing fleeting beauty over glistening ocean without", "Stern librarian hushed raucous group, reminding them to stay quiet inside", "Ink flowed from author's pen, words brimming with raw passion as page filled during", "Earthy aroma of freshly steeped tea perfumed air, inviting moment of serenity along", "Masterful guitarist's fingers danced nimbly across strings, room alive with haunting melody around", "Meticulous chef artfully garnished plate, adding delicate finishing touches over", "Indomitable marathoner pushed through punishing final stretch, fortitude driving every stride before", "Engrossed scientist examined specimen's intricate structures through microscope beneath", "Nervous thespian steadied breathing, striding into dazzling spotlight, delivering flawless performance with", "Skilled artist's pencil glided gracefully, deftly capturing subject's essence without", "Weary hiker paused to catch breath, marveling at sweeping panorama from lofty peak above", "Deep in thought, writer drummed fingers, seeking perfect phrasing to convey profound emotion without", "Lost in reverie, violinist swayed gently, fingers dancing across delicate strings during", "Painter's brushstrokes burst into radiant life, canvas ablaze with vivid sunset hues over", "Adept photographer framed picturesque scene, preserving landscape's beauty without", "World-renowned chef meticulously garnished plate, each component strategically placed around", "Dedicated researcher scrutinized specimen under microscope, documenting minute details beneath", "Seasoned actor inhaled deeply, embodying character as bright lights engulfed stage with", ].

## I.2  MORE EXAMPLES OF FACTUAL RECALL

We consider more examples of factual recall with pairs of input and output shown in Table 5.

Table 5: Inputs and Outputs of Factual Knowledge

| Input | Target output |
|---|---|
| The Great Wall is located in | China |
| Mount Kilimanjaro is located in | Tanzania |
| The Nobel Prize is awarded in | Sweden |
| The Statue of Liberty stands in | New York Harbor |
| Vatican City is enclosed within | Rome |
| The Acropolis is situated in | Athens |
| The Sydney Opera House is located on | Bennelong Point |
| The Galápagos Islands belong to | Ecuador |
| The Aurora Borealis can be seen in | Norway |
| The Amazon River flows through | Brazil |
| The Andes Mountains extend through | Chile |
| Machu Picchu is found in | Peru |
| The Kremlin is located in | Moscow |
| Uluru is a landmark found in | Australia |
| Petra is an archaeological city in | Jordan |
| Angkor Wat is located in | Cambodia |
| The city of Toronto is in | Canada |
| The city of Barcelona is in | Spain |
| The city of Mumbai is in | India |
| The Eiffel Tower is located in | Paris |

## J  SYNTHETIC IOI TASK

**Data and task.** Here we consider a synthetic data model similar to the IOI task Wang et al. (2022), with additional noise. Consider a vocabulary $\mathcal{V} = \{1, 2, \ldots, N, N+1\}$. The token $\tau \triangleq N+1$ is the generic noise token. We fix a *trigger* token $q \in [N]$, which governs in-context recall, and a context length $T$. Each sequence of tokens $z_{1:T} = [z_1, z_2, \ldots, z_T]$ is generated as follows:

i. Sample a correct *output* token $\bar{y}$ and a different *distractor* token $y^D$ uniformly in $[N]$.

ii. Sample three indices $i_1, i_2, i_3 \in [T-2]$ such that their distances are no smaller than 2. (This is for non-overlapping.)

iii. Set $z_{i_1} = z_{i_2} = z_{i_3} = q$. Among the three indices $i_1 + 1, i_2 + 1, i_3 + 1$, random select one of them with $z_{i_k+1} = \bar{y}$ with the other two as $z_{i_k+1} = y^D$.

iv. Set $z_T = q$ and sample $z_{T+1} \sim p_{\alpha, \bar{y}}(\cdot)$ with

$$
p_{\alpha, \bar{y}}(x) = \begin{cases} 1 - \alpha, & \text{if } x = \bar{y}, \\ \alpha, & \text{if } x = \tau, \\ 0, & \text{otherwise.} \end{cases}
$$

v. Random fill with tokens from $\mathcal{V} \setminus \{q\}$ into the remaining positions in $[T+1] \setminus \{i_1, i_1+1, i_2, i_2+1, i_3, i_3+1, T, T+1\}$.

The key difference between the above data and noisy in-context recall in Section 3 is that, in additional to detecting the tokens $\bar{y}$ and $y^D$ after the trigger $q$, this task also requires counting to decide which of $\bar{y}$ and $y^D$ appear more. This mechanism is exactly the definition of the correct IO token in Wang et al. (2022).

Most of the other settings are the same as that in Section 3, including the training procedure, the architecture of a transformer layer, dimensionality and the vocabulary size.

**Results.** Figure 19 shows the test performance for models with layers $L = 3, 4, 5, 6, 7$, where the models are trained with SGD. **Dropping the last-layer MLP** consistently improves the test performance across all models. Figure 20 shows the test performance for $L = 3, 4, 5$ trained with Adam (Kingma, 2014). **Truncating the last MLP's input weights** with $\rho = 0.01$ significantly

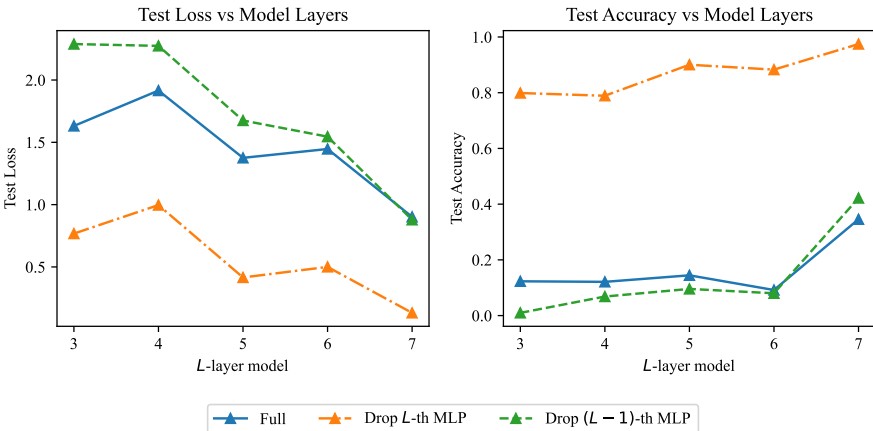

Figure 19: **Synthetic IOI trained with SGD**: test loss and accuracy for transformers with different layers. Dropping the last-layer MLP consistently improves the test accuracies across all models.

improves the performance for $L = 3, 4$. We also note that the model fails to converge for $L = 5$, possibly because we do not use any normalization technique in the architecture, so the Adam training is less stable for deep transformers.

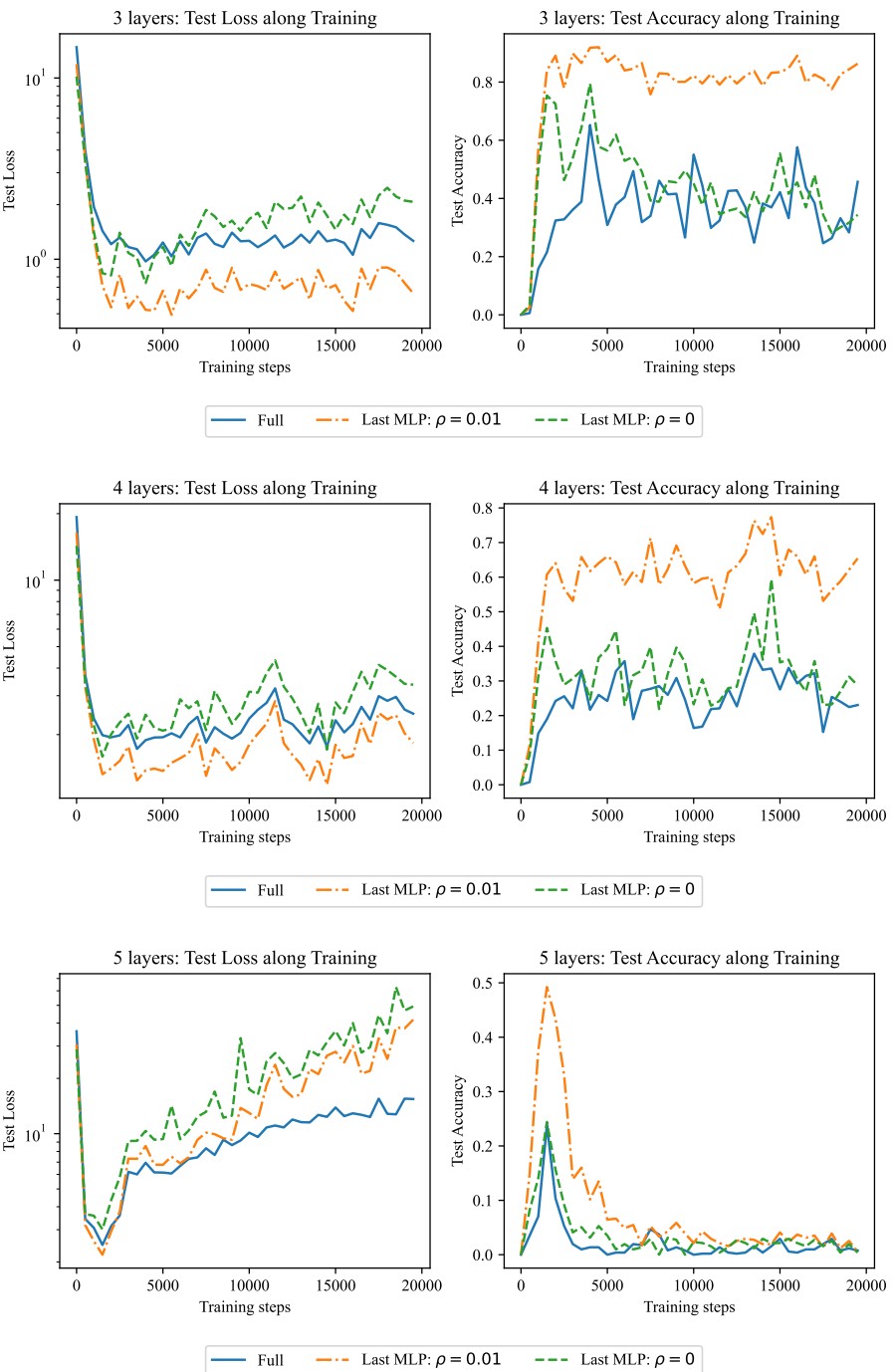

Figure 20: **Synthetic IOI trained with Adam**: test loss and accuracy for transformers with layers $L = 3, 4, 5$. Truncating the last-layer MLP's input weights with $\rho = 0.01$ improves the test performances for $L = 3, 4$, while the model fails to converge for $L = 5$.

