# OpenReview forum: "Distributional Associations vs In-Context Reasoning: A Study of Feed-forward and Attention Layers"
_ICLR.cc/2025/Conference — ICLR 2025 Poster_

### Official Review · Reviewer_P7Ta · 2024-10-31

**Soundness:** 3
**Presentation:** 3
**Contribution:** 2
**Rating:** 6
**Confidence:** 4

**Summary:**

The authors studied how attention mechanisms and feed-forward layers handle distributional associations and in-context inputs differently. They considered simple two-layer transformers on an in-context recall task and observed the distinct focuses of feed-forward layers and attention mechanisms through analyzing gradient.
They further employed LLMs like GPT-2 and Pythia, concentrating on IOI dataset . Additionally, the authors validate the effectiveness of the LASER technique.
Overall, the authors underscore the differing roles of attention layers and feed-forward layers in language models.

**Strengths:**

1. The authors conducts an in-depth study on how attention and feed-forward layers differently process distributional associations and contextual inputs, yielding clear and significant conclusions.
2. The flow of the paper is easy to follow.
3. The authors employ clear theoretical proofs and a logical structure, supporting the authors' arguments.

**Weaknesses:**

1. The experiments in this paper are insufficient to demonstrate authors’ claim, they only evaluate with limited number of datasets and model structures, which might not be representative of the general behavior.

2. The technique authors recommended, LASER[1], is not introduced by this paper, making this paper seems an extension or complement to LASER. Differentiation between this paper’s contributions and [1]’s should be claimed.

3. The methods and datasets this work employed is interesting but mostly appear a simple extension of that in [2]. This makes the research seem a supplementary work.

**Reference**

[1] The Truth is in There: Improving Reasoning in Language Models with Layer-Selective Rank Reduction

[2] Birth of a Transformer: A Memory Viewpoint

**Questions:**

1. Is there potential modifications or improvements for the authors to propose to LASER that could strengthen this paper’s contributions?
2. In Section 4.2, why LASER struggles with more shots?
3. Is it possible to make other modifications beyond those in [2], such as designing an entirely new dataset to better validate your claims?

---

> ### Author Response · Authors · 2024-11-25
> **Author response (1/2)**
>
> Thanks for your helpful review!
>
> ---
>
> **Q1**: The experiments in this paper are insufficient to demonstrate authors’ claim, they only evaluate with limited number of datasets and model structures, which might not be representative of the general behavior.
>
> **A1**: Thanks for expressing this concern. We set out to understand how and why feed-forward and attention layers learn different parts of the data distribution in **simple and controlled settings**, which allows us to have theory and experiments that closely match, and thus provide robust evidence that may generalize to larger systems. More complex data models and architectures are likely to be much more complicated to analyze, which would make our arguments less transparent. We also note that despite their simplicity, our toy models in Section 3 already capture the complexities of the task at hand. Finally, our findings extend to large language models pretrained on real-world datasets, as shown in Section 4, which highlights the robustness of our insights.
>
> **Q2**: The technique authors recommended, LASER, is not introduced by this paper, making this paper seems an extension or complement to LASER. Differentiation between this paper’s contributions and [1]’s should be claimed. Is there potential modifications or improvements for the authors to propose to LASER that could strengthen this paper’s contributions?
>
> **A2**: We would like to emphasize that we use LASER truncation merely as a tool to probe whether certain associations are "stored" in different layers. The original paper's finding that LASER may improve reasoning performance was in fact a motivation for us to understand the phenomenon more deeply. This lead us to define the notions of distributional vs in-context associations, and to show how the training process may disentangle them into feed-forward and attention layers.
>
>
> **Q3**: The methods and datasets this work employed is interesting but mostly appear a simple extension of that in [2]. This makes the research seem a supplementary work.
>
> **A3**: We agree that [2] is a significant previous work for Section 3 of our work, but we would like to highlight the following key differences between these two:
>
> 1. **Different motivations**. We consider more realistic scenarios where there is **competition** between in-context reasoning and distributional associations (e.g. predicting "in the" and "in Spain" are both possible after "Madrid is located"), while [2] has a clear separation between in-context predictions (only on trigger tokens) and global bigrams (only on non-trigger tokens).
> 2. **Different theoretical results**. Our theoretical results differ significantly from [2]: our focus is on comparing **feed-forward and attention** layers in storing distributional associations, and we provide **finite sample** results, which is crucial for understanding how the model disentangles distributional associations from in-context reasoning. In contrast, [2, Theorem 3] only considers gradients with infinite samples, and only studies attention layers for learning the induction head mechanism.
> 3. **Empirical verification of more tasks on pre-trained LLMs**. We extend our study to more realistic reasoning tasks with pre-trained LLMs (see Section 4), which shows that our findings about disentanglement of distributional associations and in-context reasoning extends to such real-world scenarios. This includes by the impact of truncating feed-forward layers on complex reasoning paradigms such as GSM8k with CoT. In contrast, [2] only considers two-layer transformers on the synthetic bigram dataset.
>
> ---
>
> References:
>
> [1] The Truth is in There: Improving Reasoning in Language Models with Layer-Selective Rank Reduction. Sharma et al. ICLR 2024.
>
> [2] Birth of a Transformer: A Memory Viewpoint. Bietti et al. NeurIPS 2023.

---

> > ### Author Response · Authors · 2024-11-25
> > **Author Response (2/2)**
> >
> > **Q4**: In Section 4.2, why LASER struggles with more shots?
> >
> > **A4**: Thanks for this question. One possible explanation is that truncating MLPs may remove too many distributional associations, including ones that are useful in intermediate steps of the reasoning chain, while few-shot prompting with many few-shot examples may succeed in only removing spurious distributional associations. Indeed, many-shot prompting has been found to promote in-context reasoning (to the point where it can [break safety guardrails](https://www.anthropic.com/research/many-shot-jailbreaking)), suggesting that spurious distributional associations will be automatically dropped.
> >
> > **Q5**: Is it possible to make other modifications beyond those in [2], such as designing an entirely new dataset to better validate your claims?
> >
> > **A5**: Yes, it is! Following your suggestion, we include a new synthetic IOI setting in Appendix J in the revision. This task requires counting the *correct* token $\bar{y}$ and the *distractor* token $y^D$ to predict the less frequent $\bar{y}$. Figure 19 shows that dropping the last-layer MLP consistently improves the test performance for $3,4,5,6,7$-layer models trained with SGD. Figure 20 shows that truncating the last-layer MLP weights also improves for $3,4$-layer models trained with Adam. We also note Adam training is not stable for more deep models like $L=5$, possibly because we did not use any normalization technique in the architecture.
> >
> > Generally, we position this new task as a more difficult one than the noisy in-context recall in Section 3, because it not only requires detecting tokens after trigger but also counting each token. We leave its theoretical analysis into future work.
> >
> >
> > ---
> >
> > We'd be happy to clarify any further questions.

---

> > > ### Comment · Reviewer_P7Ta · 2024-11-26
> > >
> > > Thank you for the response. I raised my rating to 6.

---

> > > > ### Author Response · Authors · 2024-11-26
> > > >
> > > > Thanks for your generous reconsideration!

---

### Official Review · Reviewer_d4XH · 2024-10-31

**Soundness:** 4
**Presentation:** 4
**Contribution:** 3
**Rating:** 8
**Confidence:** 3

**Summary:**

In this paper, the authors investigate the role of transformers' self-attention and feed-forward networks (FFNs) both experimentally, in a controlled environment with a simplified model, and theoretically, using an even simpler model that allows them to derive calculations. The authors thus highlight an important distinction: FFNs are biased towards learning distributional associations (i.e. co-occurrence cues), while self-attention focuses on in-context reasoning.

Concretely, as it is common to have the word “the” after the word “in”, a model using distributional cues only would be tempted to complete the sentence “Madrid is located in” with “the”, whereas a model using in-context reasoning would complete it with “Spain”.

So the authors set up an experiment in which the model is trained on a text containing with equal probability the “a + b” and “a + noise” patterns (where “a” varies from sentence to sentence, while the “noise” token remains the same) and then evaluated on a text containing only the “a + b” pattern. The idea is then to measure in-context recall in such a scenario: indeed, a purely distributional model will perform at 50% accuracy where a perfectly in-context reasoning model will achieve 100%. Using LASER to reduce the rank of the second-layer MLP matrix, until it is completely suppressed (rho = 0), they note that distributional behavior disappears with MLP rank, suggesting that FFNs are biased in this direction. They also note that self-attention manages to focus only on “a + b” patterns, ignoring “a + noise” patterns.

The authors then propose a theoretical explanation for such behavior, analyzing in particular the first gradient descent step. Finally, the authors propose to test their findings in a more realistic setting, i.e. on real models and real-world reasoning benchmarks.

**Strengths:**

- It is very well written and clear.
- Their experimental environment is very clearly established, from the creation of their dataset to the architecture of the model.
- I think that experimenting with LLMs in a controlled setting is an excellent methodology.
- The paper proposes a theoretical justification for their discovery that is more than welcome and very rare in the field.
- The findings are very interesting and help to better understand the behavior of LLMs, in particular FFNs, which are often less studied than self-attention.

**Weaknesses:**

- The main experiments and theoretical results are based on highly simplified models. However, the observed behavior may not scale with larger models.
- The theorems and their consequences are relatively complicated to understand for a non-initiated reader of this literature, perhaps try to explain the different terms a little more.
- Experimental figures have no error bars.

**Questions:**

1) Why did you use such a small model for the experiments? Would it be possible to perform the same kind of analysis with more layers?
2) What is your interpretation of the poorer performance of truncated MLPs after 8-shot on GSM8K (Table 2)?
3) Your results seem to indicate that reducing FFNs greatly improves in-context reasoning, which is often desirable. However, FFNs also play an important role. It would therefore be interesting to study what is lost when they are reduced/removed?

---

> ### Author Response · Authors · 2024-11-25
>
> Thanks for your encouraging feedback!
>
> ---
>
>
> **Q1**: The theorems and their consequences are relatively complicated to understand for a non-initiated reader of this literature, perhaps try to explain the different terms a little more.
>
> **A1**: We agree with that and will keep thinking about how to improve the presentation. This is also why we provide Figure 1 and 2 to help understanding our motivations and results even without reading the theorems.
>
> **Q2**: Experimental figures have no error bars.
>
> **A2**: Thanks for the reminder! For the settings of IOI and factual recall in Figure 5, we add new plots of the prediction distributions in Figure 15 and 16 in the revision. Both are evaluated on the last checkpoints of the models.
>
> 1. Figure 15 uses the same 100 IOI sentences on Pythia-410M and 1B.
> 2. Figure 16 uses 20 more examples of factual recall (in Table 3) to measure the prediction distribution, instead of the single factual query ''Madrid is located in'' in Figure 5.
>
> For other figures in Section 3, the curves are quite stable due to a synthetic setting, so we omit the error bars for better visualization.
>
>
>
> **Q3**: Why did you use such a small model for the experiments? Would it be possible to perform the same kind of analysis with more layers?
>
> **A3**: In Section 3, we use a two-layer transformer because it is the simplest model capable of solving the induction head task (the recent work [1] proves that one layer is not enough). This simplicity provides results and architectures that are more readable and interpretable. For deeper models with more than two layers, since the attention scores are almost uniform at initialization regardless of the layer, all layers should behave similarly at the first gradient step, and we expect a result similar to Theorem 1 to hold for each layer. However, subsequent steps would likely become more complex and tedious to analyze.
>
> In Section 4, small models are used in experiments for the following reasons:
>
> * For IOI, as a quick demo following the setting in [2], we study ignored top predictions about generic words from GPT-2 small, while [2] focuses on only subjects ([S]) and indirect objects ([IO]). Then we extend the study to larger pythia models.
> * As we are interested in the training dynamics from scratch, training checkpoints of pythia models are one of the very few open-source options that are accessible to us.
>
> [1] One-layer transformers fail to solve the induction heads task. Sanford et al. arXiv:2408.14332
>
> [2] Interpretability in the wild: a circuit for indirect object identification in gpt-2 small. Wang et al. ICLR 2023.
>
> **Q4**: What is your interpretation of the poorer performance of truncated MLPs after 8-shot on GSM8K (Table 2)?
>
> **A4**: Thanks for this question. One possibility is that truncating MLPs may remove too many distributional associations, including ones that are useful in the reasoning chain, while few-shot prompting with many few-shot examples may succeed in only removing spurious distributional associations. Indeed, many-shot prompting has been found to promote in-context reasoning (to the point where it can [break safety guardrails](https://www.anthropic.com/research/many-shot-jailbreaking)), suggesting that spurious distributional associations will be automatically dropped.
>
> **Q5**: Your results seem to indicate that reducing FFNs greatly improves in-context reasoning, which is often desirable. However, FFNs also play an important role. It would therefore be interesting to study what is lost when they are reduced/removed?
>
> **A5**: Thanks for this question, this is indeed an important point. One example where this might arise as a limitation is in GSM8K, as described in our answer above, where intermediate reasoning steps may be negatively affected by truncation, which leads to poor performance compared to using many few-shot examples. Understanding how to truncate layers more selectively, e.g., by erasing only certain well-chosen associations rather than entire subspaces, would be an important next step to overcome these limitations.
>
> ---
>
> We'd be happy to clarify any further questions.

---

> > ### Comment · Reviewer_d4XH · 2024-11-27
> >
> > Thank you for your detailed response to my questions. Your clarifications addressed my concerns, and I appreciate the effort you put into explaining these points.

---

### Official Review · Reviewer_Gq5Y · 2024-11-03

**Soundness:** 3
**Presentation:** 3
**Contribution:** 2
**Rating:** 6
**Confidence:** 3

**Summary:**

Large language models excel in in-context reasoning tasks like coherent language generation and knowledge storage. Their Transformer architecture relies on feed-forward and attention layers linked to knowledge and reasoning. In controlled experiments, the authors found feed-forward layers learn simple distributional associations, while attention layers focus on context reasoning. Gradient noise explains this difference. Pre-trained models, like Pythia, show similar disparities in simple reasoning tasks.

**Strengths:**

1. This paper uses a large number of experiments and theoretical proof to draw the following conclusions:
For the Transformer architecture of large language model LLM, feed-forward layers learn simple distributional associations, while attention layers focus on context reasoning.

2. The experimental results and conclusions are relatively solid, and the theoretical proof is quite convincing.

3. This work may be helpful for future researchers studying LLM.

**Weaknesses:**

1. This work is highly theoretical and lacks certain practical significance and applicability.

**Questions:**

1. In addition to feed-forward layers and attention layers, there are also wording embedding layers, residual connections, and LayNorm layers in the Transformer architecture. What special role do these layers play in helping the model understand the semantics of the natural language? Did the authors explore?

2. Because the attention module is the interaction between all tokens in the input sequence, it intuitively plays a role in understanding context. It seems obvious that the attention layer mainly focuses on in-context reasoning, right?

3. Are the given conclusions applicable to all models based on the Transformer architecture? Is this applicable to all LLM models?

4. In terms of format:
    Line 1935, Line 1938, Line 1952, Line 2001, Line 2069 exceed the margins and have incorrect formatting.

---

> ### Author Response · Authors · 2024-11-25
>
> Thanks for your helpful review!
>
> ---
>
> **Q1**: This work is highly theoretical and lacks certain practical significance and applicability.
>
> **A1**: We agree with the difficulty of making practical benefits in LLMs with theoretical understanding. Nevertheless, we position this work as a careful study of ignored empirical phenomenon to inspire better post-training. For instance, in [1], they studied a curcuit of attention heads in a trained LLM on IOI task, totally focusing on probabilities of predicting subjects (S) and indirect objects (IO) at the end of training. However, we carefully study the probabilities of generic ''the'' word along the training in Figure 5 (left), as well as LASER. Following many previous works on alignment, this work is another evidence of improving a pre-trained model during post-training.
>
> [1] Interpretability in the wild: a circuit for indirect object identification in gpt-2 small. Wang et al. ICLR 2023.
>
> **Q2**: In addition to feed-forward layers and attention layers, there are also wording embedding layers, residual connections, and LayNorm layers in the Transformer architecture. What special role do these layers play in helping the model understand the semantics of the natural language? Did the authors explore?
>
> **A2**: Thanks for raising this point! Our toy models *do* include residual connections and word embeddings, but they do not include LayerNorm, and the word embeddings are not trained. This is mostly to simplify our analysis, and we note that this simple model already suffices to capture the different behavior of FF and attention layers.
> While learned embeddings are crucial in real-world natural language problems, our toy task is mostly algorithmic and does not involve rich semantics. We expect that learning embeddings would be important in tasks with richer semantic structure (as in [2]).
> Layernorm is also crucial in practice for efficient and stable training, but we found it to have minor effects in our simple setup where data is well-balanced.
>
> [2] How Do Transformers Learn Topic Structure: Towards a Mechanistic Understanding. Li et al. ICML 2023.
>
>
> **Q3**: Because the attention module is the interaction between all tokens in the input sequence, it intuitively plays a role in understanding context. It seems obvious that the attention layer mainly focuses on in-context reasoning, right?
>
> **A3**: While this may seem obvious, the fact that distributional and in-context associations are **disentangled** into different blocks of a transformer is a nontrivial result! Indeed, in terms of expressivity alone, attention could easily capture distributional associations, and training an MLP on top of an untrained/frozen attention could capture in-context associations. The fact that different layers automatically specialize to disentangle between these two tasks is nontrivial, and follows from a careful study of training dynamics.
>
> **Q4**: Are the given conclusions applicable to all models based on the Transformer architecture? Is this applicable to all LLM models?
>
> **A4**: We do believe that our conclusions would apply to any Transformer LLMs. We have verified this empirically on GPT-2, Pythia models up to 2.8B, and Llama models up to 8B, and there is no reason to believe these behaviors would change at larger scale.
>
> We did not investigate other architectures beyond Transformers, but that would be an interesting avenue for future work.
>
> **Q5**: In terms of format: Line 1935, Line 1938, Line 1952, Line 2001, Line 2069 exceed the margins and have incorrect formatting.
>
> **A5**: Thanks for pointing out! We have fixed them in the revision.
>
> ---
>
> We'd be happy to clarify any further questions.

---

> ### Comment · Reviewer_Gq5Y · 2024-12-01
>
> Appreciate the reply.
> After reading the comments from other reviewers and the authors' responses, the authors answered most of my questions, and fixed the format in the revision version. I will keep my score.

---

### Official Review · Reviewer_U1qi · 2024-11-04

**Soundness:** 3
**Presentation:** 3
**Contribution:** 2
**Rating:** 6
**Confidence:** 3

**Summary:**

This work theoretically examines a phenomenon regarding the purpose of the feedforward and attention layers of a Transformer model, relative to the reasoning and knowledge learning tasks. It attempts to provide an explanation for the difference through a generic noise token, demonstrating that the feedforward layers store noise while the induction heads in the attention layers are insensitive to the noise tokens.

**Strengths:**

* The paper sets out to address a clear objective, and designs a 2-layer model that enables a theoretical analysis of the phenomenon of interest, then reduces to a 1-layer model for theoretical analysis, revealing important, mathematically-grounded supporting evidence for said phenomenon.
* I appreciate that the tasks of interest, and the distinguishment between the two abilities of interest (distributional associations and in-context reasoning) is made clear upfront.
* Figure 2 is especially helpful to follow the 2-layer setup for the in-context recall task with noise tokens — the finding that the induction head filters out the noisy token is an interesting insight, and reinforced by Theorem 2 and its proof in Appendix D.
* While the paper is dense, it is also organized quite well and in a manner that emphasizes the key takeaways.

**Weaknesses:**

* The novelty presented the results section (Section 4) is unclear — are there any architectural insights informed by the theoretical analyses that could be applied that have been considered, beyond an application of LASER? Aside from this, this section appears to be a more in-depth, investigative analysis on the behavior of the chosen pre-trained models for the selected tasks; it is also unclear how / whether these insights scale as the models considered are small.
* I have no major concerns about the theoretical results in the work, which reveal the means through which noise is stored both in the presence and absence of feedforward layers, under a simple 1-layer model. I must acknowledge that while I have largely studied and followed the proofs of Theorems 1 and 2 and the theory in Appendix A, I have not thoroughly verified the correctness of the theory in the rest of the appendix.

**Questions:**

* I would suggest moving the proofs of Lemmas C.1 and C.2 up, before they are invoked in the proof of Theorem 1 / 4. This would make it easier to follow the progression of claims made.
* In the abstract, the term “gradient noise” is used in line 21, but not referred to in the rest of the paper — please clarify (or remove, if not relevant) this term.
* The use of the term 'sequel' on line 125 is unclear; this should be clarified or rephrased accordingly.
* As noted in the weaknesses section, are there any architectural changes that may be proposed, conditioned on the results of the exploration? Is the key takeaway that one should perform low-rank truncation as in LASER on the feed-forward layers since they store noise? Including discussion on this would improve the actionability of the findings of this work and its impact.

---

> ### Author Response · Authors · 2024-11-25
>
> Thanks for your helpful review!
>
> ---
>
> **Q1:** The novelty presented the results section (Section 4) is unclear [...]
>
> **A1:** The novelty of our work is not any specific architectural change or the application of LASER. Rather, our contribution is an **understanding** of how FF and attention layers are able to **disentangle** different parts of the data distribution during the training process, namely by preferentially storing distributional associations in FF layers, while attention layers tend to pick up in-context reasoning mechanisms. Our use of weight truncation / LASER is simply as a tool for probing for this behavior in both our toy models and on pretrained LLMs, and Section 4 is meant to investigate the validity of our claims on actual LLMs beyond the synthetic settings of Section 3.
>
> **Q2**: Is the key takeaway that one should perform low-rank truncation as in LASER on the feed-forward layers since they store noise? Including discussion on this would improve the actionability of the findings of this work and its impact.
>
>
> **A2**: As mentioned above, our contribution focuses on understanding the different roles of attention and FF weights in disentangling distributional vs in-context associations, both empirically and theoretically. The application of low-rank truncation is simply a way to verify our claims, and is consistent with the findings in the LASER paper that truncating some FF layers may improve performance on some reasoning tasks.
>
> Nevertheless, our perspective based on distributional associations versus in-context reasoning may be helpful in thinking about how to allocate parameters to feed-forward versus attention layers: for instance, on our synthetic task, we found that for a fixed total parameter budget, models with fewer MLP parameters achieve higher loss on distributional predictions (e.g., non-contextual bigrams) compared to models with more MLP parameters (and fewer attention parameters). These notions may also provide a different way to reason about circuit discovery in mechanistic interpretability from the perspective of training dynamics and properties of the training data. Finally, this disentanglement may inform more effective ways to fine-tune models, e.g., by selectively choosing which layers to fine-tune.
>
> We've included these points and the results on varying MLP vs attention parameters in the revision, see Appendix A.
>
>
> **Q3**: I would suggest moving the proofs of Lemmas C.1 and C.2 up, before they are invoked in the proof of Theorem 1 / 4. This would make it easier to follow the progression of claims made.
>
> **A3**: We have reorganized the appendix in the revision.
>
>
> **Q4**: In the abstract, the term “gradient noise” is used in line 21, but not referred to in the rest of the paper — please clarify (or remove, if not relevant) this term.
>
> **A4**: Thanks for pointing this out. "Gradient noise" is referring to the fact that the attention gradients at initialization are much more noisy than the feed-forward gradients in storing the distributional association (see Theorem 1 and section 3.1 more generally). We have changed this to "noise in the gradients" in the abstract to avoid confusion.
>
> **Q5:** The use of the term 'sequel' on line 125 is unclear; this should be clarified or rephrased accordingly.
>
> **A5:** Thanks for pointing this out, we replaced "in the sequel" by "below": the *in-context recall* task is studied in Section 3 while the other two are explored empirically in Section 4.
>
> ---
>
> We'd be happy to clarify any further questions.

---

> > ### Comment · Reviewer_U1qi · 2024-11-27
> > **Official Comment by Reviewer U1qi**
> >
> > Thank you for your response and revisions. I appreciate the writing changes made, and think that the Proof of Theorem 1 follows more clearly now. I find the paper interesting and the theoretical contributions insightful, but still feel that a more concrete set of recommendations on the MLP vs attention parameters while *scaling* model size would improve the actionability of the work. As such, I will keep my score, but still think that the work merits inclusion in the conference.

---

### Author Response · Authors · 2024-11-25
**Updates in the Revision**

Dear reviwers and AC,

Thanks for your helpful feedback! We posted individual reponses to each reviewer, along with a revised submission.

The revision has the following major updates:

1. **A new synthetic IOI task** in Appendix J, following the question about additional tasks by Reviewer P7Ta. This task requires counting the *correct* token $\bar{y}$ and the *distractor* token $y^D$ to predict the less frequent $\bar{y}$. It turns out truncating the last-layer MLP weights consistently improve test performance for layers $L=3,4,5,6,7$.
2. **More examples of factual recall** in Table 5 with results presented in Appendix C.3, to show the robustness of our arguments.
3. Re-orgnized sections in Appendix, as suggested by Reviewer U1qi.
4. A discussion on our contributions and implications in Appendix A.

We are very glad to answer any further questions.

---

### Meta-Review · Area_Chair_Garu · 2024-12-20

**Metareview:**

The papers provide an analysis about feed-forward layers and attention layers. The main observation is that FFN is generally responsible for distributional association, whereas attention layers are responsible for in-context reasoning. The authors back up their claim by both empirical and theoretical analysis, although the setups are a little bit toy.

Reviewers generally recognize contributions of this paper. Concerns are raised regarding the simplified setups, but it is understandable that theoretical analysis on full-sized models could be extremely difficult. On the other hand, the implication of the finding could be further discussed.

**Additional Comments On Reviewer Discussion:**

Reviewers are unanimously on the positive side.

---

### Decision · Program_Chairs · 2025-01-22

Accept (Poster)